# FedOne: Query-Efficient Federated Learning for Black-box Discrete Prompt Learning

**Ganyu Wang** [* 1] **Jinjie Fang** [* 2] **Maxwell J. Yin** [1] **Bin Gu** [2] **Xi Chen** [3] **Boyu Wang** [1 4] **Yi Chang** [2] **Charles Ling** [1 4]

## Abstract

Black-Box Discrete Prompt Learning is a prompt-tuning method that optimizes discrete prompts without accessing model parameters or gradients, making the prompt tuning on a cloud-based Large Language Model (LLM) feasible. Adapting federated learning to BDPL could further enhance prompt tuning performance by leveraging data from diverse sources. However, all previous research on federated black-box prompt tuning had neglected the substantial query cost associated with the cloud-based LLM service. To address this gap, we conducted a theoretical analysis of query efficiency within the context of federated black-box prompt tuning. Our findings revealed that degrading FedAvg to activate only one client per round, a strategy we called *FedOne*, enabled optimal query efficiency in federated black-box prompt learning. Building on this insight, we proposed the FedOne framework, a federated black-box discrete prompt learning method designed to maximize query efficiency when interacting with cloud-based LLMs. We conducted numerical experiments on various aspects of our framework, demonstrating a significant improvement in query efficiency, which aligns with our theoretical results.

## 1. Introduction

Prompt tuning has emerged as a vital technique for adapting LLMs (Liu et al., 2019; Brown et al., 2020) to specific tasks without retraining the entire model. Traditionally, many tuning methods require access to the model's intermediate representations (Brown et al., 2020; Li & Liang, 2021; Liu et al., 2021; Lester et al., 2021), categorizing them as white-box approaches. However, when such access is unavailable, black-box prompt tuning becomes essential. This approach focuses on tuning the input prompts without access to the internal processes of the model (Sun et al., 2022; Diao et al., 2022; Deng et al., 2022; Xiao et al., 2023).

Federated Learning (FL) (McMahan et al., 2017; Li et al., 2020; 2021; Karimireddy et al., 2020; Gu et al., 2020; Mishchenko et al., 2022) has emerged as a promising approach for leveraging decentralized data from multiple clients while preserving privacy. Applying federated learning to prompt tuning offers a valuable opportunity to improve client capabilities and enhance model performance. Most federated prompt tuning methods to date have focused on white-box scenarios where clients have access to model parameters (Zhao et al., 2023; Zhang et al., 2023). In those approaches, only the trainable parameters (prompts) are trained by the client and shared with the server, significantly reducing both the number of trainable parameters and the communication costs compared to fine-tuning baselines.

However, several practical limitations hinder the applicability of federated prompt tuning that relies on white-box access. First, white-box prompt learning is not applicable to closed-source LLM that are accessed via APIs, as these models are not openly shared. In such scenarios, users are restricted to interacting with the model through the API endpoints without access to the internal structure or weights of the LLM. This limitation prevents the application of white-box prompt learning techniques. Second, FL is typically applied in scenarios involving thousands of edge devices, each with limited computational resources. However, white-box prompt learning demands substantial computational power, as it requires devices to perform computation on the entire LLM. These operations are computationally intensive, rendering them impractical for edge devices with constrained capabilities. This presents a significant challenge when trying to implement white-box prompt learning in FL environments, as it can lead to excessive resource demands on the participating devices.

In contrast, applying black-box prompt tuning to FL offers several distinct benefits (Lin et al., 2023; Zhang et al., 2023;

---

[*]Equal contribution [1]Western University, London, Ontario, Canada [2]Jilin University, Changchun, Jilin, China [3]McGill University, Montreal, Quebec, Canada [4]Vector Institute, Toronto, Ontario, Canada. Correspondence to: Bin Gu <jsgubin@gmail.com>, Charles Ling <charles.ling@uwo.ca>.

*Proceedings of the $42^{nd}$ International Conference on Machine Learning*, Vancouver, Canada. PMLR 267, 2025. Copyright 2025 by the author(s).

Figure 1: Query-Efficiency Comparison of FedOne and Federated BDPL

Che et al., 2023), enhancing both the practicality and effectiveness. First, this approach preserves the privacy of closed-source LLMs by not requiring access to their internal model weights or architecture. For example, Diao et al. (2022) proposed Black-box Discrete Prompt Learning (BDPL) [1], which exclusively utilizes discrete prompts as inputs and optimizes them based solely on the model's output loss. Besides, black-box prompt tuning reduces the computational burden on the client, as it eliminates the need for computation on the entire model, thereby enabling participation from edge devices with limited computational resources. Moreover, the communication costs associated with discrete prompts are lower compared to white-box prompt tuning methods, as the white-box prompt tuning require transmitting large matrices of numerical values.

Despite these advantages, the application of black-box prompt tuning in FL still faces two significant unresolved challenges, tempering its overall promise. First, previous research (Lin et al., 2023; Sun et al., 2023) has neglected the substantial cost associated with queries to the LLM cloud service (Figure 1, left). Second, a convergence analysis of Federated BDPL for optimizing discrete prompts has not yet been conducted.

Targeting the above problems, we introduce a novel federated learning framework, FedOne, designed to optimize query efficiency in federated BDPL. We offer the first convergence analysis of federated BDPL in this context and further extend the analysis of the query efficiency towards the cloud-based LLM server. The results demonstrate that by limiting activation to a single client per round, FedOne achieves optimal query efficiency (Figure 1, right). Our

approach is particularly well-suited for scenarios involving limited computational resources, such as mobile devices or IoT systems, where local training of LLMs is impractical.

Formally, this paper makes the following key contributions:

- We identify that existing federated black-box prompt tuning methods overlook the significant costs associated with querying cloud-based LLM services.

- To address this gap, we present the first theoretical analysis of federated BDPL, with a focus on understanding and evaluating the query efficiency when interacting with cloud-based LLMs.

- Based on our theoretical result, we introduce the FedOne framework, a novel approach designed to optimize query efficiency in federated black-box prompt tuning by activating only one client per round.

## 2. Method

### 2.1. Federated Black-box Prompt Learning Framework

In the federated black-box prompt tuning framework, there is one central aggregation server and $K$ clients. Each client, indexed by $k$, possesses a dataset $D^k$, consisting of input sentences $\Psi^k$ and their corresponding labels $Y^k$, i.e., $D^k = \{\Psi^k, Y^k\}$. The dataset $\Psi^k$ contains a total of $M^k$ input sentences, each represented as $\psi_m^k$, i.e., $\Psi^k = \{\psi_m^k\}_{m=1}^{M^k}$. Similarly, the corresponding labels $y_m^k$ comprise $Y^k$, i.e. $Y^k = \{y_m^k\}_{m=1}^{M^k}$.

Each client generates a discrete sequence of prompt tokens $\Phi^k = \phi_1^k \cdots \phi_i^k \cdots \phi_n^k$ from a trainable parameter $\boldsymbol{\alpha}^k \in \mathbb{R}^{n \times N}$. This learnable parameter, $\boldsymbol{\alpha}^k$, is transmitted to the FL aggregation server for averaging. Details of how to generate the discrete prompt $\Phi^k$ from $\boldsymbol{\alpha}^k$ and the local train-

---

[1] The term "BDPL" is explicitly used to refer to the work by Diao et al. (2022), whereas "black-box prompt tuning" is used to denote the broader research area.

ing of $\boldsymbol{\alpha}^k$ through interaction with the cloud-based LLM service will be discussed in the next subsection regarding the local black-box prompt learning on the client.

The server and clients collaboratively solve a minimization problem, aiming to reduce a global loss function that aggregates the local loss functions from the clients. This can be expressed as:

$$
\min_{\Phi} \left\{ \mathcal{L}(\Phi; \Psi) \triangleq \sum_{k=1}^{K} \frac{M^k}{M} \mathcal{L}^k(\Phi; \Psi^k) \right\}
$$

$$
\text{where: } \mathcal{L}^k(\Phi; \Psi^k) = \frac{1}{M^k} \sum_{m=1}^{M^k} \ell\left(\Phi; \psi_m^k, y_m^k\right) \quad (1)
$$

where $\mathcal{L}(\Phi; \Psi)$ represents the global objective function of the FL, $\Psi = \left\{\Psi^k\right\}_{k=1}^{K}$. $\mathcal{L}^k(\cdot, \Psi^k)$ is the local objective function of client $k$. $\ell(\cdot; \cdot)$ denotes the loss function.

### 2.2. Black-box Discrete Prompt Learning on the Client

We now discuss the local training process of the client through interactions with the cloud-based LLM service. In this section, all variables have the superscript $k$, indicating that they belong to the $k$-th client. However, the reader can ignore this superscript and treat it as a standalone training process on a single machine.

**Generating the Discrete Prompt Sequence $\Phi^k$** The sequence of the discrete prompt $\Phi^k$ is generated from a vocabulary $\mathcal{V} = \{\mathcal{V}[j]\}_{j=1}^{N}$, which contains a total of $N$ token options. Each token $\phi_i^k$ in the prompt sequence $\Phi^k$ is selected from the vocabulary, i.e., $\Phi^k = \phi_1^k \cdots \phi_i^k \cdots \phi_n^k = \mathcal{V}[j_1^k] \cdots \mathcal{V}[j_i^k] \cdots \mathcal{V}[j_n^k]$. For the $i$-th token $\phi_i^k$, the prompt index $j_i^k$ is sampled from the categorical distribution $\boldsymbol{p}_i^k$, i.e., $j_i^k \sim \text{Cat}(\boldsymbol{p}_i^k)$. Note that $\boldsymbol{p}_i^k = [p_{i,1}^k, ... p_{i,N}^k]$, where the element $p_{i,j}^k$ represents the probability that the token $\phi_i^k$ is selected as $V[j]$ from the vocabulary $\mathcal{V}$.

Directly optimizing the $\boldsymbol{p}_i^k$ may cause trouble in the convergence analysis as the gradient of the categorical distribution is biased. To address this, we re-parameterize the categorical distribution $\boldsymbol{p}_i^k$ using the Gumbel-Softmax technique (Jang et al., 2016) and introduce the parameter $\boldsymbol{\alpha}_i^k = [\alpha_{i,1}^k, ... \alpha_{i,N}^k]$ as the learnable parameter. The re-parameterization is shown below:

$$
p_{i,j}^k = \frac{\exp\left(\frac{\log(\alpha_{i,j}^k) + g_{i,j}^k}{\tau}\right)}{\sum_{l=1}^{N} \exp\left(\frac{\log(\alpha_{i,l}^k) + g_{i,l}^k}{\tau}\right)} \quad (2)
$$

where $\tau > 0$ is the temperature parameter, $g_{i,j}^k \sim \text{Gumbel}(0, 1)$ is the Gumbel random variable. We denote the Gumbel-Softmax function as GS, i.e. $\boldsymbol{p}^k = \text{GS}(\boldsymbol{\alpha}^k)$.

**Optimizing the Learnable Parameter $\boldsymbol{\alpha}^k$** To compute the gradient with respect to the learnable parameter $\boldsymbol{\alpha}^k$, we first define the expected loss over the sequence of prompts in Eq. 3. The $i$-th token $\phi_i^k$ is generated from the vocabulary by sampling the prompt index from the categorical distribution, i.e. $\phi_i^k = \mathcal{V}[j_i^k]$, where $j_i^k \sim \text{Cat}(\boldsymbol{p}_i^k)$. For brevity, this sampling process is denoted as $\phi_i^k \sim \boldsymbol{p}_i^k$. We can define the expectation of the loss for the distribution of the prompt as follows:

$$
\mathbb{E}_{\Phi^k \sim \boldsymbol{p}^k} \left[\mathcal{L}(\Phi^k, \Psi^k)\right]
$$

$$
= \sum_{\phi_1^k \sim \boldsymbol{p}_1^k} \cdots \sum_{\phi_n^k \sim \boldsymbol{p}_n^k} \left( \mathcal{L}(\Phi^k, \Psi^k) \prod_{i=1}^{n} P(\phi_i^k) \right) \quad (3)
$$

Following the same steps in (Diao et al., 2022, Eq. (2)), we can estimate the gradient w.r.t. $\boldsymbol{\alpha}_i^k$ by:

$$
\nabla_{\boldsymbol{\alpha}_i^k} \mathbb{E}_{\Phi^k \sim \text{GS}(\boldsymbol{\alpha}^k)} \left[\mathcal{L}(\Phi^k, \Psi^k)\right]
$$

$$
= \mathbb{E}_{\Phi^k \sim \text{GS}(\boldsymbol{\alpha}^k)} \left[\mathcal{L}(\Phi^k, \Psi^k) \nabla_{\boldsymbol{\alpha}_i^k} \log P(\phi_i^k)\right] \quad (4)
$$

The $j$-th component of $\nabla_{\boldsymbol{\alpha}_i^k} \log P(\phi_i^k)$ could be explicitly computed as follows (with detailed steps provided in Appendix A.2):

$$
\nabla_{\alpha_{i,j}^k} \log P(\phi_i^k) = \nabla_{\alpha_{i,j}^k} \log p_{i,j^k}^k = \begin{cases} \frac{1 - p_{i,j^k}^k}{\tau \alpha_{i,j^k}^k} & j = j_i^k \\ -\frac{p_{i,j}^k}{\tau \alpha_{i,j}^k} & j \neq j_i^k \end{cases} \quad (5)
$$

Then, we employ the mini-batch stochastic variance-reduced policy (MB-SVRP) estimator (Diao et al., 2022; Williams, 1992) to reduce the variance of the sampling when computing the gradient. This involves sampling the prompt sequence $\Phi^k$ from the distribution $\boldsymbol{p}^k$ multiple times, with the number of samplings denoted by $I$. The MB-SVRP estimator is then computed as follows:

$$
\hat{\nabla}_{\boldsymbol{\alpha}_i^k} f^k(\boldsymbol{\alpha}^k, \mathcal{B}^k)
$$

$$
= \frac{1}{I-1} \sum_{r=1}^{I} \left[ \left(\ell(\Phi^{k,r}; \mathcal{B}^k) - \ell_{avg}\right) \nabla_{\boldsymbol{\alpha}_i^k} \log P(\phi_i^{k,r}) \right] \quad (6)
$$

where $\ell_{avg} = \frac{1}{I} \sum_{w=1}^{I} \ell(\Phi^{k,w}; \mathcal{B}^k)$, and $\left\{\Phi^{k,r}\right\}_{r=1}^{I}$ are sampled independently from $\boldsymbol{p}^k = \text{GS}(\boldsymbol{\alpha}^k)$. The mini-batch $\mathcal{B}^k$, with size $B^k$, is sampled from the dataset $\Psi^k$. Note that the clients are unable to directly compute $\ell(\Phi^{k,*}, \mathcal{B}^k)$ on their own. They must transmit both the sampled prompt $\Phi^k$ and the mini-batch $\mathcal{B}^k$ to the cloud-based LLM service, which then computes the loss $\ell(\Phi^{k,*}, \mathcal{B}^k)$ and returns the result. Finally, with the learning rate set to $\eta$, the update of $\boldsymbol{\alpha}_i^k$ at the $t$-th iteration is expressed as follows:

$$
\boldsymbol{\alpha}_{i,(t+1)}^k = \boldsymbol{\alpha}_{i,(t)}^k - \eta \cdot \hat{\nabla}_{\boldsymbol{\alpha}_i^k} f^k(\boldsymbol{\alpha}^k, \mathcal{B}_t^k) \quad (7)
$$

## 2.3. Algorithm

Algorithm 1 outlines the Fed-BDPL framework, which integrates federated averaging with local client training with Gumbel-Softmax-BDPL (GS-BDPL). The FL aggregation server randomly selects a subset of clients and broadcasts the trainable parameters to them. Each selected client $k$ then performs local training on the parameters and return the updated parameters $\boldsymbol{\alpha}^k$ to the server. The local training of the selected client is conducted through black-box prompt learning by querying the cloud-based LLM service, as detailed in Section 2.2. Finally, the server aggregates the updates by averaging the parameters from all participating clients. This process is iteratively repeated throughout the training. Specifically, in the FedOne framework, the number of activated clients is set to 1, as highlighted in the light green box. The reasons behind FedOne's high query efficiency on cloud-based LLM services will be formally analyzed in the next section.

---

**Algorithm 1** Fed-BDPL. $C$ denotes the sampling ratio of the clients. $K_*$ represents the number of selected clients, and $U$ is the set of selected clients.

1: **Server executes:**
2: Initialize $\boldsymbol{\alpha}$
3: **for** $s = 0, 1, \ldots S - 1$ **do**
4:     FedAvg: $K_* \leftarrow \max(C \cdot K, 1)$
5:     FedOne: $K_* \leftarrow 1$
6:     $U_s \leftarrow$ (sampling $K_*$ clients)
7:     **for** $k \in U_s$ **in parallel do**
8:         $\boldsymbol{\alpha}^k \leftarrow \boldsymbol{\alpha}$
9:         $\boldsymbol{\alpha}^k \leftarrow$ Client_Update($k, \boldsymbol{\alpha}^k$)
10:     **end for**
11:     $\boldsymbol{\alpha} \leftarrow \frac{1}{K_*} \sum_{k \in U_s} \boldsymbol{\alpha}^k$
12: **end for**

13: **Client executes:**
14: **function** Client_Update($k, \boldsymbol{\alpha}^k$):
15: **for** $e = 1, \ldots, E$ **do**
16:     Re-parameterize the categorical distribution $\boldsymbol{p}^k =$ GS($\boldsymbol{\alpha}^k$) using Eq. 2.
17:     Query the cloud-based LLM server to obtain $\left\{ \mathcal{L}(\Phi^{k,r}, \mathcal{B}_e^k) \right\}_{r=1}^I$.
18:     Compute $\hat{\nabla}_{\boldsymbol{\alpha}_i^k} f^k(\boldsymbol{\alpha}^k, \mathcal{B}_e^k)$ using Eq. 6.
19:     $\boldsymbol{\alpha}_i^k \leftarrow \boldsymbol{\alpha}_i^k - \eta \cdot \hat{\nabla}_{\boldsymbol{\alpha}_i^k} f^k(\boldsymbol{\alpha}^k, \mathcal{B}_e^k)$
20: **end for**
21: Return $\boldsymbol{\alpha}^k$

---

## 3. Convergence Analysis

### 3.1. Assumptions

**Assumption 3.1. Unbiasedness and Bounded Variance of Stochastic Gradient:** We assume that the stochastic gradient is unbiased and has bounded variance.

$$\mathbb{E}_{\psi_m^k} \left[ \nabla_{\boldsymbol{\alpha}_i^k} f^k(\boldsymbol{\alpha}^k, \psi_m^k) \right] = \nabla_{\boldsymbol{\alpha}_i^k} F(\boldsymbol{\alpha}^k, \Psi^k) \tag{8}$$

$$\mathbb{E}_{\psi_m^k} \left\| \nabla_{\boldsymbol{\alpha}_i^k} f^k(\boldsymbol{\alpha}^k, \psi_m^k) - \mathbb{E}_{\psi_m^k} \left[ \nabla_{\boldsymbol{\alpha}_i^k} f^k(\boldsymbol{\alpha}^k, \psi_m^k) \right] \right\|^2 \le \sigma_\psi^2 \tag{9}$$

Assumption 3.1 is the basic assumption for solving non-convex optimization problems using stochastic gradient descent (Ghadimi & Lan, 2013; Hazan & Kale, 2014; Xu et al., 2019; Liu et al., 2020).

**Assumption 3.2. Bounded Loss**: At the $k$-th client, $\forall \psi_m^k \in \Psi^k$, and $\Phi^k$ sampled by $\boldsymbol{p}^k$, the loss function $l(\cdot, \cdot)$ is upper bounded by a constant $G$, i.e.,

$$\left| \ell(\Phi^k, \psi_m^k) \right| \le G \tag{10}$$

Assumption 3.2 ensures that the loss value is bounded, primarily to regulate the loss during $I$-sample estimation in stochastic policy gradient, thereby facilitating theoretical analysis (Ghadimi & Lan, 2013; Zhang et al., 2019; Liu et al., 2022).

**Assumption 3.3. Bounded Clients' Heterogeneity:** For client $k = 1, ..., K$ with sampling probability vector $\boldsymbol{q} = \left\{ q^{[k]} \right\}_{k=1}^K$, and $\nabla_{\boldsymbol{\alpha}_i^k} F^k(\boldsymbol{\alpha}^k, \Psi^k)$ is local gradient of $\boldsymbol{\alpha}_i$ w.r.t. all input sentence in client $k$. We assume that $\lambda$ is the upper bound on the weighted gradient diversity across local objectives, i.e.

$$\Lambda(\boldsymbol{\alpha}_i, \boldsymbol{q}) \triangleq \frac{\sum_{k=1}^K q^{[k]} \cdot \left\| \nabla_{\boldsymbol{\alpha}_i^k} F^k(\boldsymbol{\alpha}^k, \Psi^k) \right\|^2}{\left\| \sum_{k=1}^K q^{[k]} \cdot \nabla_{\boldsymbol{\alpha}_i^k} F^k(\boldsymbol{\alpha}^k, \Psi^k) \right\|^2} \le \lambda \tag{11}$$

Assumption 3.3 states that the gradient diversity among clients is bounded. Following Yin el al. (2018) and Haddad-pour et al. (2019), we use gradient diversity as a measure of client heterogeneity and assume an upper bound on it.

### 3.2. The Convergence of Federated BDPL

**Theorem 3.4.** *Suppose that assumption 3.1, 3.2 and 3.3 hold, algorithm 1 is used to solve the FedBDPL problem defined in Eq. 1. Let $B = \min \left\{ B^1, ..., B^K \right\}$ where $B^k$ represents the local mini-batch size for each client. Set $\alpha_{i,j} \ge \nu > 0$. The variance of the variance-reduced policy gradient is given by $\sigma_\alpha^2 = \frac{8G^2 N}{\tau^2 \nu^2}$. $\mathbb{E}_{\Phi^k \sim GS(\boldsymbol{\alpha}^k)} \left[ \mathcal{L}(\Phi^k, \Psi^k) \right]$ is $L$-smooth w.r.t. $\boldsymbol{\alpha}^k$ where $L = \frac{nGN(\tau+1)}{\tau^2 \nu^2}$. The learning rate $\eta < \min \left\{ \frac{1}{L\lambda}, \frac{1}{L} \right\}$. Then, the expected gradient $\nabla_{\boldsymbol{\alpha}} F(\boldsymbol{\alpha}_t, \Psi^k)$ of Fed-BDPL, can be bounded as follows:*

$$\frac{1}{T} \sum_{t=0}^{T-1} \left\| \nabla_{\boldsymbol{\alpha}} F(\boldsymbol{\alpha}_t, \Psi^k) \right\|^2$$

$$\le \frac{4G}{\eta T} + \frac{2(E+1)n\sigma_\psi^2(1 + \frac{1}{K_*}) + 2n\sigma_\psi^2}{B}$$

$$+ \frac{2(E+1)n\sigma_\alpha^2(1+\frac{1}{K_*}) + 2n\sigma_\alpha^2}{I^2} \quad (12)$$

*The proof of Theorem 3.4 is provided in Appendix A.*

**Corollary 3.5.** *Convergence Rate of Fed-BDPL: Let $\eta = \min\left\{\frac{1}{L\lambda}, \frac{1}{\sqrt{T}}, \frac{1}{L}\right\}$, $B = \sqrt{T}$ and $I = T^{\frac{1}{4}}$, Fed-BDPL achieves a sub-linear convergence rate, i.e.*

$$\frac{1}{T}\sum_{t=0}^{T-1}\|\nabla_{\boldsymbol{\alpha}}F(\boldsymbol{\alpha}_t, \Psi)\|^2 = \mathcal{O}(\frac{1}{\sqrt{T}}) \quad (13)$$

**Corollary 3.6.** *The Impact of $K_*$ (FedOne): Let $Q_\epsilon$ denote the minimum number of queries submitted to the cloud-based LLM service to achieve an $\epsilon$-solution, formulated as a function of $K_*$, i.e., $Q_\epsilon(K_*)$. We derive its explicit form in Eq. 45 (in Appendix A.4):*

$$Q_\epsilon(K_*) = c\left[c_1\sqrt{K_*} + c_2\frac{1}{\sqrt{K_*}}\right]^2 \quad (14)$$

*and establish that the minimum of $Q_\epsilon(K_*)$ is achieved when $0 < K_* < 1$. Consequently, $Q_\epsilon(K_*)$ is a monotonically increasing function within the feasible region of $K_*$, for $K_* = 1, 2, \ldots, K$. Thus, to minimize query overhead and achieve optimal efficiency, the optimal choice is $K_* = 1$.*

*Remark* 3.7. The intuition behind Corollary 3.6 is that while increasing the number of activated clients $K_*$ leads to a sublinear improvement in the convergence rate, it simultaneously incurs a linear increase in the number of queries to the cloud-based LLM in Fed-BDPL. As a result, the marginal gains in convergence do not justify the increasing query overhead in the LLM model. A toy experiment in the next section will further illustrate this trade-off.

## 4. Experiment

The experiment aimed to evaluate the performance of Fed-BDPL, analyze various aspects of the framework, and examine the query efficiency advantages of FedOne. The implementation is available at: `https://github.com/GanyuWang/FedOne-BDPL`.

**A Toy Experiment on the Intuition of FedOne** To illustrate the intuition behind FedOne, we began with a toy experiment examining the trade-off between query efficiency and the number of activated clients $K_*$ in a federated learning setting using the MNIST dataset (LeCun et al., 2010).[2] The dataset is evenly distributed across 100 clients. We train for 2 epochs with a learning rate of 0.01 and a batch size of 32, varying the number of active clients, i.e., $K_* \in \{1, 5, 10, 20, 40\}$. The model for each client is a

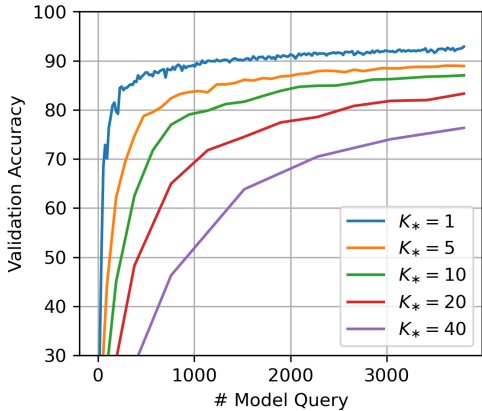

Figure 2: Toy Example on the Intuition of FedOne

Multilayer Perceptron (MLP). It includes a flattening input layer, a fully connected layer with 512 neurons and ReLU activation, a dropout layer with 0.2 dropout rate, and a final fully connected layer that outputs to 10 classes via a Softmax function. As illustrated in Figure 2, utilizing the minimum number of activated clients (FedOne) achieved optimal query efficiency for convergence.[3]

It is a widely held belief that, in traditional federated learning scenarios, increasing the number of participating clients accelerates convergence by leveraging more local updates per round (Li et al., 2019). However, in the context of federated black-box prompt tuning, the number of queries to the LLM server increases linearly with the number of participating clients. This rise in query numbers outpaces the benefits gained from faster convergence due to increased client participation. As a result, FedOne provides a more efficient trade-off by reducing query overhead while still enabling effective model training.

### 4.1. Experiment Setup

For our experiment, we utilized the GLUE benchmark (Wang et al., 2018), which includes a wide range of tasks including MNLI (Williams et al., 2018), QQP (Iyer et al., 2017), SST-2 (Socher et al., 2013), MRPC (Dolan & Brockett, 2005), CoLA (Warstadt et al., 2019), QNLI (Wang et al., 2018), and RTE (Dagan et al., 2005; Haim et al., 2006; Giampiccolo et al., 2007; Bentivogli et al., 2009).

In the baseline experiment within the federated learning framework, we employed 100 clients. In the FedOne framework, there was only one client activated per round for training and aggregation. We adopted the $k$-shot framework from (Perez et al., 2021), adapting it to a federated learning context. Each client received a $k$-shot dataset comprising $k$ samples per class.

---

[2]A similar experiment on the impact of $K_*$ using the SST-2 dataset for Federated BDPL is presented in Appendix C.1.

[3]Additional experiments on varying levels of client heterogeneity are provided in Appendix C.3.

Table 1: The overall performance on the RoBERTa-large. Each trial runs across three random seeds.

| Dataset | MNLI | QQP | SST-2 | MRPC | CoLA | QNLI | RTE | Avg. |
|---|---|---|---|---|---|---|---|---|
| Manual Prompt | $35.9_{1.3}$ | $49.8_{0.9}$ | $77.2_{1.1}$ | $70.4_{1.6}$ | $0.6_{0.0}$ | $49.2_{1.1}$ | $48.2_{0.6}$ | 47.33 |
| In Context Learning | $37.2_{1.6}$ | $50.1_{0.9}$ | $82.8_{2.1}$ | $72.1_{2.3}$ | $1.1_{0.4}$ | $50.8_{0.5}$ | $49.3_{2.3}$ | 49.06 |
| FineTuning | $\mathbf{50.8}_{1.2}$ | $60.8_{1.9}$ | $\mathbf{86.5}_{2.0}$ | $78.4_{1.3}$ | $\mathbf{20.4}_{1.9}$ | $\mathbf{53.2}_{1.8}$ | $55.6_{2.3}$ | 57.96 |
| FedOne-PromptTuning | $41.5_{0.9}$ | $66.4_{0.2}$ | $77.9_{2.1}$ | $79.5_{0.5}$ | $0.8_{1.1}$ | $49.6_{1.0}$ | $53.1_{0.6}$ | 52.69 |
| FedOne-P-Tuning v2 | $42.7_{0.7}$ | $66.7_{0.1}$ | $82.9_{0.3}$ | $80.6_{0.1}$ | $1.0_{1.0}$ | $52.4_{0.2}$ | $56.4_{0.4}$ | 54.67 |
| FedOne-BBT | $41.9_{0.4}$ | $66.3_{0.2}$ | $76.8_{1.6}$ | $80.6_{0.3}$ | $2.5_{1.3}$ | $51.1_{0.4}$ | $55.3_{1.0}$ | 53.50 |
| FedOne-BDPL | $41.0_{1.2}$ | $66.7_{0.1}$ | $80.8_{6.0}$ | $\mathbf{81.1}_{0.1}$ | $5.2_{2.4}$ | $51.7_{1.4}$ | $57.1_{1.9}$ | 54.80 |
| FedOne-GS-BDPL | $41.1_{0.4}$ | $\mathbf{66.9}_{0.2}$ | $80.8_{0.4}$ | $81.0_{0.1}$ | $5.3_{1.1}$ | $52.1_{0.8}$ | $\mathbf{57.1}_{1.1}$ | 54.90 |

The model architecture employed is RoBERTa-large (Liu et al., 2019). The trainable prompts were placed at different positions in the model depending on the algorithm of the baselines. For the training procedure, we conducted a hyperparameter tuning phase using a grid search approach to explore learning rates of $[3e-4, 1e-4, 3e-5, 1e-5]$. The batch size was set at 32, and the optimization algorithm employed was AdamW (Loshchilov & Hutter, 2017). Further details about the dataset and evaluation metrics of them are available in Appendix B.1.

## 4.2. Baselines

We evaluated several approaches categorized as standalone and federated learning models. The standalone model baseline includes Manual Prompt Tuning, In-Context Learning (Brown et al., 2020), and Fine-tuning (Diao et al., 2022, Table 7). In the domain of Federated Prompt Tuning for white-box scenarios, we adapted two established white-box prompt tuning methods to federated learning. Specifically, we implemented prompt-tuning (Lester et al., 2021) and prefix-tuning v2 (Liu et al., 2021) across distributed clients. In the prompt tuning approach, a prompt was integrated into the embedding layer of the model for each client. Each client then undertook local training solely on their respective prompt. In prefix-tuning v2, the prompt was utilized across all embedding layers of the model, providing more trainable parameters and enhancing the capability to adapt to downstream tasks. In the federated prompt tuning for the black-box scenario, we adapted the Black-Box Tuning (BBT) (Sun et al., 2022) to FedOne, incorporating projection from a low-dimensional vector and the Covariance Matrix Adaptation Evolution Strategy (CMA-ES) into federated black-box prompt learning. Each client held a distinct low-dimensional vector, while the projection matrix $A$ was shared among all clients. For every client, the population size of CMA-ES is set to 20, and the dimension of the low-dimensional vector is set to 500, as recommended by (Sun et al., 2022). Finally, we adapted the BDPL (Diao et al., 2022) and Gumbel-Softmax BDPL (GS-BDPL) to the Federated Learning, employing policy gradient methods and Gumbel-Softmax as outlined in Algorithm 1.

## 4.3. Result

**Test Accuracy** The performance results were summarized in Table 1, we observed significant variations in the effectiveness of different learning approaches when applied to the RoBERTa-large model across various NLP tasks. Traditional fine-tuning methods outperformed both Manual Prompt and In-Context Learning techniques, achieving the highest average score of 57.96 across all datasets. This result underscores the effectiveness of complete model retraining over other methods that involve fewer parameter updates or rely solely on contextual adjustments. Although Manual Prompt performed well on some datasets, its performance lacks stability. White-box federated prompt tuning methods demonstrate improvements over Manual Prompt and In Context Learning techniques. Generally, the black-box method (BDPL and GS-BDPL) exhibits performance that is comparable to, or slightly better than, the white-box tuning method (PromptTuning and P-Tuning v2).

**Computational Efficiency and Resource Utilization** Table 2 presents the performance metrics for various federated prompt tuning methods, focusing on the computational and communication efficiencies of these methods. We measured the computation time required for model training. Notably, in the white-box method, training occurs on the client side. We only measure the time for forward and backward propagation of the local model, along with the time for parameter updates. In the black-box prompt learning method, the computation time for model training includes both the execution of the black-box algorithm and the wait time for responses from the LLM model. Regarding GPU memory usage, white-box methods require substantial computational resources, such as GPUs, which may not be feasible in FL environments. In contrast, black-box methods significantly reduce GPU memory consumption. A comparison between the FedOne framework and the federated framework with 10 activated clients demonstrates that FedOne reduces both computation time and FL server communication costs compared with traditional FL. Within black-box prompt learning, although BBT has a slightly smaller trainable parameter size, it requires a significantly longer training

Table 2: Evaluation of computational efficiency and resource utilization

| Baseline | Comp. Time for Training (s) | FL Server Comm. Cost (MB) | FL Server # Queries | LLM Server # Query | Per Client Trainable Parameter Size (MB) | Per Client Loaded GPU Memory (MB) |
|---|---|---|---|---|---|---|
| Fed-Prompt-Tuning | 224.26 | 156.25 | 1000 | - | $7.8 \times 10^{-2}$ | 3564 (Model, Grad., Prompt) |
| Fed-P-Tuning v2 | 276.23 | 11417.20 | 1000 | - | 5.7 | 3656 (Model, Grad., Prompt) |
| Fed-BBT | 24501.83 | 3.81 | 1000 | 200000 | $1.9 \times 10^{-3}$ | $1.90 \times 10^{-3}$ (Prompt Only) |
| Fed-BDPL | 2407.05 | 30.52 | 1000 | 20000 | $1.5 \times 10^{-2}$ | $1.52 \times 10^{-2}$ (Prompt Only) |
| Fed-GS-BDPL | 2400.60 | 30.52 | 1000 | 20000 | $1.5 \times 10^{-2}$ | $1.52 \times 10^{-2}$ (Prompt Only) |
| FedOne-Prompt-Tuning | 16.99 | 15.63 | 100 | - | $7.8 \times 10^{-2}$ | 3564 (Model, Grad., Prompt) |
| FedOne-P-Tuning v2 | 19.95 | 1141.72 | 100 | - | 5.7 | 3656 (Model, Grad., Prompt) |
| FedOne-BBT | 5036.15 | 0.38 | 100 | 20000 | $1.9 \times 10^{-3}$ | $1.90 \times 10^{-3}$ (Prompt Only) |
| FedOne-BDPL | 234.92 | 3.05 | 100 | 2000 | $1.5 \times 10^{-2}$ | $1.52 \times 10^{-2}$ (Prompt Only) |
| FedOne-GS-BDPL | 234.73 | 3.05 | 100 | 2000 | $1.5 \times 10^{-2}$ | $1.52 \times 10^{-2}$ (Prompt Only) |

Table 3: Federated discrete black-box prompt learning on GPT-3.5 Turbo

| | MNLI | QQP | SST-2 | MRPC | CoLA | QNLI | RTE | Avg. |
|---|---|---|---|---|---|---|---|---|
| No Prompt | $14.05_{0.00}$ | $68.04_{0.00}$ | $91.35_{0.00}$ | $79.62_{0.00}$ | $36.01_{0.00}$ | $56.22_{0.00}$ | $72.20_{0.00}$ | 59.64 |
| Prompt w/o. Training | $9.17_{1.31}$ | $62.94_{4.94}$ | $79.47_{2.71}$ | $79.88_{1.94}$ | $25.53_{1.10}$ | $56.69_{4.76}$ | $72.16_{2.51}$ | 54.41 |
| FedOne-BDPL | $13.67_{0.17}$ | $68.25_{3.05}$ | $87.77_{3.82}$ | $81.69_{0.70}$ | $32.16_{6.51}$ | $70.58_{1.35}$ | $77.62_{1.62}$ | 61.67 |
| FedOne-GS-BDPL | $\mathbf{16.00}_{1.78}$ | $\mathbf{69.92}_{2.85}$ | $\mathbf{92.58}_{2.71}$ | $\mathbf{82.93}_{0.67}$ | $\mathbf{36.20}_{2.95}$ | $\mathbf{72.05}_{1.64}$ | $\mathbf{80.33}_{2.32}$ | $\mathbf{64.28}$ |

time, making it less practical. In contrast, BDPL achieves a more balanced trade-off between computational efficiency and training performance.

The advantage of black-box prompt learning lies primarily in its reduced communication costs and the efficiency of the trainable parameter size, as well as in avoiding the need to store and train the entire LLM on the client. As illustrated in the table, the federated black-box prompt tuning method features a significantly smaller parameter size and eliminates the GPU requirement on the client. This allows devices with limited computational resources, such as edge devices, to participate in the federated prompt learning process.

### 4.4. Experiment on Real-world Cloud-based LLM

We implemented the Fed-BDPL framework using GPT-3.5 Turbo, a widely recognized and powerful closed-source language model. We leveraged the OpenAI API (OpenAI, 2024) to enable individual clients to conduct local training. In this implementation of the federated black-box prompt learning, clients sent prompts and input sentences to GPT-3.5, which returns the logarithm of the token probabilities at each position. A key challenge was that OpenAI API only provides probabilities for the top 20 tokens for each position. Consequently, we needed to transform the predictions on these tokens into the categorical prediction of the input sentence, instead of using the straightforward model output as we did in the RoBERTa experiment. To solve this problem, we appended a template question at the end of the input sentence to query the target label token. For example, in the QQP task, we added the phrase "equivalent? yes or

no" to the end of the input sentence. This allowed us to retrieve the top probabilities for all class label tokens ("yes", "no") in the top 20 probability. and use the probability of the target token as the logit output, for the following procedure.

The results were presented in table 3, indicating that GPT-3.5 Turbo achieves a certain level of performance without any prompts. Introducing a random, untrained prompt led to a performance decline, while prompt tuning significantly improved results, surpassing the no-prompt baseline. Additionally, the table highlights that the GS-BDPL method consistently outperforms other black-box approaches. In summary, this method enables prompt tuning in Federated Learning with minimal computational resources while leveraging advanced cloud-based LLMs.

## 5. Related Works

### 5.1. White-box and Black-box Prompt Tuning

Prompt tuning is a technique for adapting LLM to downstream tasks. It tailors the model's responses to specific tasks or styles without requiring the retraining of the entire model. In white-box prompt tuning, the learner is granted full access to the LLM, allowing them to modify and access intermediate results of the model and acquire the gradient. Li et al. (2021) and Lester et al. (2021) proposed a lightweight and modular alternative to full model fine-tuning for natural language generation tasks, which optimizes a sequence of continuous soft prompts, prepend in the embedding layers of the LLM. Liu et al. (2021) proposed the P-tuning v2. Instead of only applying the prompt in the input

layer as used in Li et al. (2021), they adapt trainable parameters on all layers' inputs, which can effectively match the performance of fine-tuning across a wide range of models.

In situations where the learner cannot access the intermediate result of the LLM model, the learner has to use a black-box prompt tuning method. In the black-box prompt tuning, the learner can only query the output of the LLM with the input of the model. Most of the research assumes the input is at the soft prompt layer of the LLM (Sun et al., 2022; Chen et al., 2023). Others research use the discrete prompt which is concrete with the input text, which is more portable, and is usable for any cloud-based LLM API (Diao et al., 2022). Xiao et al. (2023) presented a privacy-preserving, efficient transfer learning method that adapts large foundation models to specific tasks. This method does not require access to the full model or compromise data privacy. It utilizes a lightweight adapter and a compressed emulator for model tuning. Chen et al. (2023) introduces an efficient method for optimizing instructions for black-box LLMs using Bayesian optimization of soft prompts. This approach significantly improves LLM performance across a variety of tasks without requiring direct access to the model's internals. Deng et al. (2022) proposed an efficient method to optimize discrete text prompts using reinforcement learning, demonstrating superior performance compared to other prompt optimization techniques across a variety of tasks. Sun et al. (2022) proposed black-box tuning (BBT), a method that optimizes continuous prompts by optimizing a lower-dimensional vector and projecting them to the prompt searching space. They use the Covariance Matrix Adaptation Evolution Strategy (CMA-ES) for optimizing the vector. Diao et al. (2022) introduced BDPL which optimizes the discrete prompt with the policy gradient method.

The black-box prompt tuning method is versatile and applicable to various tasks and models without model-specific modifications. However, its major drawback is computational inefficiency, requiring multiple forward passes through the model, which leads to high costs and extended training times. Additionally, the convergence of the derivation-free optimization method is often slow.

### 5.2. Federated Learning

Federated learning (McMahan et al., 2017; Karimireddy et al., 2020; Li et al., 2020; 2021; Marfoq et al., 2022; Mishchenko et al., 2022; Chen et al., 2020; Wang et al., 2023; Zhang et al., 2024; Wang et al., 2024; 2025), first introduced by Mcmahan et al. (2017), is a paradigm that enables devices to collaboratively train a shared predictive model by locally aggregating updates. In this framework, each client maintains a copy of the model for local training, and the server selects a subset of clients in each round for aggregation. Since the introduction of FedAvg, it has

lacked formal theoretical convergence guarantees. As a result, researchers have made significant efforts to establish and demonstrate its convergence (Zhou & Cong, 2017; Stich, 2018).

Partial client activation is a key area of research in federated learning and has gained significant attention due to its impact on improving convergence rates and system efficiency. Stich et al. (2018) shows that in the convex case, increasing the number of activated clients significantly improves convergence rates in federated learning with independent and identically distributed (IID) data, achieving a linear speed-up. Li et al. (2019) extended the understanding of federated learning by analyzing the convergence of the FedAvg algorithm, under the convex case. They demonstrate that under the non-IID setting, the convergence rate has a weak dependence on the number of activated clients. This implies that FedAvg cannot achieve linear speedup in this scenario, allowing for a lower participation ratio to mitigate the straggler effect without compromising convergence.

### 5.3. Federated Prompt Tuning

Applying federated learning to prompt tuning can enhance the model by incorporating additional data. This approach leverages distributed datasets to improve model performance while adhering to data privacy. To apply white-box prompt tuning in FL, each client maintains the entire model but trains only the prompt parameters. These parameters are then shared and aggregated across clients (Zhao et al., 2023; Zhang et al., 2023; Che et al., 2023). However, this method assumes white-box access to LLM, which is impractical for closed-source LLMs. Consequently, black-box prompt learning has been adapted for FL to address these limitations (Lin et al., 2023; Sun et al., 2023). Lin et al. (2023), applied the BDPL (Diao et al., 2022) to the FL, where the client can train the probability matrix for the discrete prompt via querying the cloud-based LLM API. Sun et al. (2023) applied the BBT to FL. In this approach, clients train only low-dimensional vectors using CMA-ES via querying the cloud-based LLM API.

For federated prompt tuning, most existing research has focused on conventional issues in FL, such as data heterogeneity (Zhao et al., 2023), privacy (Zhang et al., 2023), security (Zhao et al., 2023), and client computation-communication efficiency (Lin et al., 2023; Sun et al., 2023). However, the query efficiency of federated black-box prompt tuning, a novel challenge arising from deploying black-box prompt tuning via cloud-based APIs, remains unexplored.

## 6. Discussions and Future Works

**Heterogeneous Clients**  Theorem 3.4 and Corollary 3.6 explicitly consider bounded client heterogeneity, and the

result that $K_* = 1$ achieves optimal query cost (FedOne) remains valid under the heterogeneous case. This is because the core intuition behind FedOne continues to hold: the query cost to the LLM increases linearly with the number of activated clients, while increasing $K_*$ is not able to provide a linear speedup in convergence (Haddadpour & Mahdavi, 2019; Li et al., 2019). However, the trade-off is that fewer activated clients per round lead to a slower convergence rate. Besides, we further include an additional experiment in Appendix C.3 to evaluate FedOne's performance under varying levels of client heterogeneity, demonstrating consistency with the theoretical results.

**Cost-aware Client Participation in FL** In conventional FL, increasing the number of participating clients typically improves per-round convergence and is generally considered beneficial. However, when the per-client cost (queries to a black-box oracle, communication, computation, encryption, etc.) becomes significant, a new trade-off arises between convergence rate and the overall system cost. This trade-off becomes particularly important as FL systems transition from small, local models to interactions with large-scale models, such as LLMs. Consequently, rising per-client costs have motivated the development of strategies like FedOne, which prioritize cost-efficiency by limiting client participation. Importantly, FedOne is not limited to scenarios involving cloud-based LLM query costs; it is broadly applicable to any FL setting where per-client costs are non-negligible. Exploring the general balance between convergence and system-level efficiency remains an important direction for future work.

## 7. Conclusion

We identified that previous research on federated black-box prompt tuning had overlooked the significant query costs associated with cloud-based LLM services. To address this issue, we conducted a theoretical analysis of query efficiency in the context of federated black-box prompt tuning, revealing the relationship between query efficiency and the number of activated clients. Based on our theoretical result, we propose the FedOne framework, which achieves optimal query efficiency with respect to the number of activated clients. We conducted extensive numerical experiments on various aspects of its performance and further validated its advantage through real-world experiments using GPT-3.5.

## Acknowledgments

This work has been supported by the Natural Sciences and Engineering Research Council of Canada (NSERC), Discovery Grants program.

## Impact Statement

This paper presents work whose goal is to advance the field of Machine Learning. There are many potential societal consequences of our work, none of which we feel must be specifically highlighted here.

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

# A. Convergence Analysis

## A.1. Notations and Omitted Mathematical Steps

| | |
|---|---|
| $t = 0, ..., T-1$ | Number of global iterations |
| $e = 1, ..., E$ | Number of local iterations |
| $s = 0, ..., S$ | Number of client-server interactions |
| $\|\cdot\|$ | The default 2-norm in this paper |
| $\text{Cat}(\cdot)$ | Categorical distribution function |
| $\phi_i \sim \boldsymbol{p}_i$ | Abbreviation for $j_i \sim \text{Cat}(\boldsymbol{p}_i)$ |
| $D^k$ | Dataset for client $k$ |
| $\Psi = \left\{\psi_m^k\right\}_{m=1}^{M^k}$ | Input sentence |
| $Y^k = \left\{y_m^k\right\}_{m=1}^{M^k}$ | The category corresponding to the input sentence |
| $K$ | The number of all clients |
| $\boldsymbol{q} = \left\{q^{[k]}\right\}_{k=1}^{K}$ | Probability of each client being selected |
| $K_*$ | The number of selected clients. |
| $U$ | The set of selected clients. |
| $\boldsymbol{\alpha}^k = \left\{\boldsymbol{\alpha}_i^k\right\}_{i=1}^{n}$ | Gumbel-Softmax parameters for $k$-th client |
| $\boldsymbol{\alpha} = \frac{1}{K_*}\sum_{k \in U_t} \boldsymbol{\alpha}^k$ | Average of $\boldsymbol{\alpha}^k$ |
| $\Phi^k = \phi_1^k \cdots \phi_n^k$ | Prompt |
| $\mathcal{V}$ | Vocabulary list of total $N$ tokens choice |
| $\boldsymbol{p}_i^k = [p_{i,1}^k, \cdots, p_{i,N}^k]$ | The probability distribution over the $N$ token indexes. |
| $\nabla_{\boldsymbol{\alpha}_i^k} F^k(\cdot, \cdot)$ | The local full gradient of the client $k$ (17) |
| $\nabla_{\boldsymbol{\alpha}_i^k} f^k(\cdot, \cdot)$ | The local stochastic mini-batch gradient of the client $k$ (16) |
| $\hat{\nabla}_{\boldsymbol{\alpha}_i^k} f^k(\cdot, \cdot)$ | The local stochastic mini-batch variance-reduced policy gradient of the client $k$ (6) |
| $\nabla_{\boldsymbol{\alpha}_i^k} F^*(\cdot, \cdot)$ | The average full gradient of the client group $U_t$ (31) |
| $\nabla_{\boldsymbol{\alpha}_i^k} f^*(\cdot, \cdot)$ | The average stochastic mini-batch gradient of the client group $U_t$ (32) |
| $\hat{\nabla}_{\boldsymbol{\alpha}_i^k} f^*(\cdot, \cdot)$ | The average stochastic mini-batch variance-reduced policy gradient of the client group $U_t$ (33) |
| $\nabla_{\boldsymbol{\alpha}_i^k} F(\boldsymbol{\alpha}^k, \cdot)$ | The average full gradient of all $K$ clients for $\left\{\boldsymbol{\alpha}^k\right\}_{k=1}^{K}$ (30) |
| $\nabla_{\boldsymbol{\alpha}_i} F(\boldsymbol{\alpha}, \cdot)$ | The average full gradient of all $K$ clients for $\boldsymbol{\alpha}$ (29) |

Table 4: Notation Table

For $i = 1, ..., n$, we define the stochastic gradient, stochastic mini-batch gradient and full gradient with respect to $\boldsymbol{\alpha}_i^k$ as following:

$$\nabla_{\boldsymbol{\alpha}_i^k} f^k(\boldsymbol{\alpha}^k, \psi_m^k) \stackrel{def}{=} \nabla_{\boldsymbol{\alpha}_i^k} \mathbb{E}_{\Phi^k \sim \text{GS}(\boldsymbol{\alpha}^k)} \left[\mathcal{L}(\Phi^k, \psi_m^k)\right] \tag{15}$$

$$\nabla_{\boldsymbol{\alpha}_i^k} f^k(\boldsymbol{\alpha}^k, \mathcal{B}^k) \stackrel{def}{=} \nabla_{\boldsymbol{\alpha}_i^k} \frac{1}{B^k} \sum_{\psi_m^k \in \mathcal{B}^k} \mathbb{E}_{\Phi^k \sim \text{GS}(\boldsymbol{\alpha}^k)}[\mathcal{L}(\Phi^k, \psi_m^k)] \tag{16}$$

$$\nabla_{\boldsymbol{\alpha}_i^k} F^k(\boldsymbol{\alpha}^k, \Psi^k) \stackrel{def}{=} \nabla_{\boldsymbol{\alpha}_i^k} \frac{1}{M^k} \sum_{\psi_m^k \in \Psi^k} \mathbb{E}_{\Phi^k \sim \text{GS}(\boldsymbol{\alpha}^k)}[\mathcal{L}(\Phi^k, \psi_m^k)] \tag{17}$$

## A.2. The Omitted Derivative Process for Eq. 5

$$\frac{\partial \log p_{i,j^k_i}^k}{\partial \alpha_{i,j}^k} = \frac{\partial}{\partial \alpha_{i,j}^k} \left( \log \left( \frac{\exp \left( \frac{\log(\alpha_{i,j^k_i}^k) + g_{i,j^k_i}^k}{\tau} \right)}{\sum_{l=1}^{N} \exp \left( \frac{\log(\alpha_{i,l}^k) + g_{i,l}^k}{\tau} \right)} \right) \right)$$

$$
\begin{aligned}
&= \frac{1}{p_{i,j_i^k}^k} \cdot \frac{\partial \left( \frac{\exp\left(\frac{\log(\alpha_{i,j_i^k}^k) + g_{i,j_i^k}^k}{\tau}\right)}{\sum_{l=1}^N \exp\left(\frac{\log(\alpha_{i,l}^k) + g_{i,l}^k}{\tau}\right)} \right)}{\partial \left( \frac{\log(\alpha_{i,j}^k) + g_{i,j}^k}{\tau} \right)} \cdot \frac{\partial \left( \frac{\log(\alpha_{i,j}^k) + g_{i,j}}{\tau} \right)}{\partial \alpha_{i,j}^k} \\
&= \frac{1}{\tau \alpha_{i,j}^k p_{i,j_i^k}^k} \cdot \frac{\partial \left( \frac{\exp\left(\frac{\log(\alpha_{i,j_i^k}^k) + g_{i,j_i^k}^k}{\tau}\right)}{\sum_{l=1}^N \exp\left(\frac{\log(\alpha_{i,l}^k) + g_{i,l}^k}{\tau}\right)} \right)}{\partial \left( \frac{\log(\alpha_{i,j}^k) + g_{i,j}}{\tau} \right)}
\end{aligned}
\tag{18}
$$

According to the derivation rule of the Softmax function,
when $j = j_i^k$:

$$
\frac{\partial p_{i,j_i^k}^k}{\partial \alpha_{i,j_i^k}^k} = \frac{1}{\tau \alpha_{i,j_i^k}^k p_{i,j_i^k}^k} \cdot \left(1 - p_{i,j_i^k}^k\right) \cdot p_{i,j_i^k}^k = \frac{\left(1 - p_{i,j_i^k}^k\right)}{\tau \alpha_{i,j_i^k}^k}
\tag{19}
$$

when $j \neq j_i^k$:

$$
\frac{\partial p_{i,j_i^k}^k}{\partial \alpha_{i,j}^k} = \frac{1}{\tau \alpha_{i,j}^k p_{i,j_i^k}^k} \cdot \left(-p_{i,j_i^k}^k \cdot p_{i,j}^k\right) = \frac{-p_{i,j}^k}{\tau \alpha_{i,j}^k}
\tag{20}
$$

## A.3. Lemmas

The following lemma shows that the unbiasedness and bounded variance of variance-reduced policy gradient. This is important for bounding the randomness introduced by prompt sampling.

**Lemma A.1.** *Unbiasedness and bounded variance of variance-reduced policy gradient:* At the $k$-th client, $\forall \psi_m^k \in \Psi^k$, $r = 1, \cdots, I$ denotes the $r$-th sampling of $\Phi^k$ from $\boldsymbol{p}^k$ w.r.t. $\left\{\Phi^{k,r}\right\}_{r=1}^I \sim \boldsymbol{p}^k$, $\boldsymbol{p}^k = GS(\boldsymbol{\alpha}^k)$, $\alpha_{i,j} \geq \nu > 0$ for $i = 1, ..., n$ and $j = 1, ..., N$, $\tau > 0$ is the temperature parameter, and $\sigma_\alpha^2 = \frac{8G^2 N}{\tau^2 \nu^2}$, then the variance-reduced policy gradient is unbiased and bounded by :

$$
\mathbb{E}_{\{\Phi^{k,r} \sim GS(\boldsymbol{\alpha}^k)\}_{r=1}^I} \left[ \hat{\nabla}_{\boldsymbol{\alpha}_i^k} f^k(\boldsymbol{\alpha}^k, \psi_m^k) \right] = \nabla_{\boldsymbol{\alpha}_i^k} f^k(\boldsymbol{\alpha}^k, \psi_m^k)
\tag{21}
$$

$$
\mathbb{E}_{\{\Phi^{k,r} \sim GS(\boldsymbol{\alpha}^k)\}_{r=1}^I} \left[ \left\| \hat{\nabla}_{\boldsymbol{\alpha}_i^k} f^k(\boldsymbol{\alpha}^k, \psi_m^k) - \mathbb{E}_{\{\Phi^{k,r} \sim GS(\boldsymbol{\alpha}^k)\}_{r=1}^I} \left[ \hat{\nabla}_{\boldsymbol{\alpha}_i^k} f^k(\boldsymbol{\alpha}^k, \psi_m^k) \right] \right\|^2 \right] \leq \frac{\sigma_\alpha^2}{I^2}
\tag{22}
$$

*Proof.* We abbreviate $\mathbb{E}_{\{\Phi^{k,r} \sim GS(\boldsymbol{\alpha}^k)\}_{r=1}^I}$ as $\mathbb{E}_{\{\Phi^{k,r}\}_{r=1}^I}$.
1) The unbiasedness of variance-reduced policy gradient:

$$
\begin{aligned}
&\mathbb{E}_{\{\Phi^{k,r} \sim GS(\boldsymbol{\alpha}^k)\}_{r=1}^I} \left[ \hat{\nabla}_{\boldsymbol{\alpha}_i^k} f^k(\boldsymbol{\alpha}^k, \psi_m^k) \right] \\
&= \mathbb{E}_{\{\Phi^{k,r}\}_{r=1}^I} \left\{ \frac{1}{I-1} \sum_{r=1}^I \left[ \left( \mathcal{L}(\Phi^{k,r}, \psi_m^k) - \frac{1}{I} \sum_{w=1}^I \mathcal{L}(\Phi^{k,w}, \psi_m^k) \right) \nabla_{\boldsymbol{\alpha}_i^k} \log P(\phi_i^{k,r}) \right] \right\} \\
&= \mathbb{E}_{\{\Phi^{k,r}\}_{r=1}^I} \left\{ \frac{1}{I-1} \sum_{r=1}^I \left[ \left( \frac{I-1}{I} \cdot \mathcal{L}(\Phi^{k,r}, \psi_m^k) - \frac{1}{I} \sum_{\substack{w=1 \\ w \neq r}}^I \mathcal{L}(\Phi^{k,w}, \psi_m^k) \right) \nabla_{\boldsymbol{\alpha}_i^k} \log P(\phi_i^{k,r}) \right] \right\} \\
&= \mathbb{E}_{\{\Phi^{k,r}\}_{r=1}^I} \left[ \frac{1}{I} \sum_{r=1}^I \left( \mathcal{L}(\Phi^{k,r}, \psi_m^k) \cdot \nabla_{\boldsymbol{\alpha}_i^k} \log P(\phi_i^{k,r}) \right) \right]
\end{aligned}
$$

$$- \mathbb{E}_{\{\Phi^{k,r}\}_{r=1}^{I}} \left\{ \frac{1}{I} \sum_{r=1}^{I} \left[ \left( \frac{1}{I-1} \sum_{\substack{w=1 \\ w \neq r}}^{I} \mathcal{L}(\Phi^{k,w}, \psi_m^k) \right) \nabla_{\boldsymbol{\alpha}_i^k} \log P(\phi_i^{k,r}) \right] \right\}$$

$$\stackrel{(1)}{=} \frac{1}{I} \sum_{r=1}^{I} \mathbb{E}_{\Phi^{k,r}} \left[ \mathcal{L}(\Phi^{k,r}, \psi_m^k) \cdot \nabla_{\boldsymbol{\alpha}_i^k} \log P(\phi_i^{k,r}) \right]$$

$$- \frac{1}{I} \sum_{r=1}^{I} \left[ \left( \frac{1}{I-1} \sum_{\substack{w=1 \\ w \neq r}}^{I} \mathbb{E}_{\Phi^w} \mathcal{L}(\Phi^{k,w}, \psi_m^k) \right) \mathbb{E}_{\Phi^r} \nabla_{\boldsymbol{\alpha}_i^k} \log P(\phi_i^{k,r}) \right]$$

$$= \mathbb{E}_{\Phi} \left[ \mathcal{L}(\Phi^k, \psi_m^k) \cdot \nabla_{\boldsymbol{\alpha}_i^k} \log P(\phi_i^k) \right]$$

$$- \frac{1}{I} \sum_{r=1}^{I} \left[ \left( \frac{1}{I-1} \sum_{\substack{w=1 \\ w \neq r}}^{I} \mathbb{E}_{\Phi^w} \left[ \mathcal{L}(\Phi^{k,w}, \psi_m^k) \right] \right) \mathbb{E}_{\Phi^r} \left[ \nabla_{\boldsymbol{\alpha}_i^k} \log P(\phi_i^{k,r}) \right] \right]$$

$$= \nabla_{\boldsymbol{\alpha}_i^k} f^k(\boldsymbol{\alpha}^k, \psi_m^k) - \frac{1}{I} \sum_{r=1}^{I} \left( \frac{1}{I-1} \sum_{\substack{w=1 \\ w \neq r}}^{I} \mathbb{E}_{\Phi^w} \left[ \mathcal{L}(\Phi^{k,w}, \psi_m^k) \right] \right) \cdot \mathbb{E}_{\Phi^r} \left( \frac{\nabla_{\boldsymbol{\alpha}_i^k} P(\phi_i^{k,r})}{P(\phi_i^{k,r})} \right)$$

$$= \nabla_{\boldsymbol{\alpha}_i^k} f^k(\boldsymbol{\alpha}^k, \psi_m^k) - \frac{1}{I} \sum_{r=1}^{I} \left( \frac{1}{I-1} \sum_{\substack{w=1 \\ w \neq r}}^{I} \mathbb{E}_{\Phi^w} \left[ \mathcal{L}(\Phi^{k,w}, \psi_m^k) \right] \right) \cdot \sum_{\phi_i^{k,r} \sim \boldsymbol{p}_i^k} \left( \frac{\nabla_{\boldsymbol{\alpha}_i^k} P(\phi_i^{k,r})}{P(\phi_i^{k,r})} \cdot P(\phi_i^{k,r}) \right)$$

$$= \nabla_{\boldsymbol{\alpha}_i^k} f^k(\boldsymbol{\alpha}^k, \psi_m^k) - \frac{1}{I} \sum_{r=1}^{I} \left( \frac{1}{I-1} \sum_{\substack{w=1 \\ w \neq r}}^{I} \mathbb{E}_{\Phi^w} \left[ \mathcal{L}(\Phi^{k,w}, \psi_m^k) \right] \right) \cdot \sum_{\phi_i^{k,r} \sim \boldsymbol{p}_i^k} \left( \nabla_{\boldsymbol{\alpha}_i^k} P(\phi_i^{k,r}) \right)$$

$$\stackrel{(2)}{=} \nabla_{\boldsymbol{\alpha}_i^k} f^k(\boldsymbol{\alpha}^k, \psi_m^k) - \frac{1}{I} \sum_{r=1}^{I} \left( \frac{1}{I-1} \sum_{\substack{w=1 \\ w \neq r}}^{I} \mathbb{E}_{\Phi^w} \left[ \mathcal{L}(\Phi^{k,w}, \psi_m^k) \right] \right) \cdot \nabla_{\boldsymbol{\alpha}_i^k} \left( \sum_{\phi_i^{k,r} \sim \boldsymbol{p}_i^k} P(\phi_i^{k,r}) \right)$$

$$\stackrel{(3)}{=} \nabla_{\boldsymbol{\alpha}_i^k} f^k(\boldsymbol{\alpha}^k, \psi_m^k) - \frac{1}{I} \sum_{r=1}^{I} \left( \frac{1}{I-1} \sum_{\substack{w=1 \\ w \neq r}}^{I} \mathbb{E}_{\Phi^w} \left[ \mathcal{L}(\Phi^{k,w}, \psi_m^k) \right] \right) \cdot \nabla_{\boldsymbol{\alpha}_i^k} (1)$$

$$= \nabla_{\boldsymbol{\alpha}_i^k} f^k(\boldsymbol{\alpha}^k, \psi_m^k)$$

where (1) uses independence of each sampling for $\Phi^k$; (2) is because $n$ is not infinite and the GS function is continuous and derivable with respect to $\boldsymbol{\alpha}_i$; (3) uses the property that the elements of probability vector sum to 1.

2) The bounded variance of variance-reduced policy gradient:

$$\text{Var}_{\{\Phi^{k,r}\sim\text{GS}(\boldsymbol{\alpha}^k)\}_{r=1}^I}\left[\hat{\nabla}_{\boldsymbol{\alpha}_i^k}f^k(\boldsymbol{\alpha}^k,\psi_m^k)\right]$$

$$= \mathbb{E}_{\{\Phi^{k,r}\sim\text{GS}(\boldsymbol{\alpha}^k)\}_{r=1}^I}\left[\left\|\hat{\nabla}_{\boldsymbol{\alpha}_i^k}f^k(\boldsymbol{\alpha}^k,\psi_m^k)-\nabla_{\boldsymbol{\alpha}_i^k}f^k(\boldsymbol{\alpha}^k,\psi_m^k)\right\|^2\right]$$

$$= \mathbb{E}_{\{\Phi^{k,r}\}_{r=1}^I}\left\{\left\|\frac{1}{I-1}\sum_{r=1}^I\left[\left(\mathcal{L}(\Phi^{k,r},\psi_m^k)-\frac{1}{I}\sum_{w=1}^I\mathcal{L}(\Phi^{k,w},\psi_m^k)\right)\nabla_{\boldsymbol{\alpha}_i^k}\log P(\phi_i^{k,r})\right]-\nabla_{\boldsymbol{\alpha}_i^k}f^k(\boldsymbol{\alpha}^k,\psi_m^k)\right\|^2\right\}$$

$$= \mathbb{E}_{\{\Phi^{k,r}\}_{r=1}^I}\left\{\left\|\frac{1}{I}\sum_{r=1}^I\left[\frac{1}{I-1}\sum_{\substack{w=1\\w\neq r}}^I\left(\mathcal{L}(\Phi^{k,r},\psi_m^k)-\mathcal{L}(\Phi^{k,w},\psi_m^k)\right)\nabla_{\boldsymbol{\alpha}_i^k}\log P(\phi_i^{k,r})-\nabla_{\boldsymbol{\alpha}_i^k}f^k(\boldsymbol{\alpha}^k,\psi_m^k)\right]\right\|^2\right\}$$

$$\overset{(1)}{=} \frac{1}{I^2}\sum_{r=1}^I\mathbb{E}_{\Phi^{k,r}}\left[\left\|\frac{1}{I-1}\sum_{\substack{w=1\\w\neq r}}^I\left(\mathcal{L}(\Phi^{k,r},\psi_m^k)-\mathcal{L}(\Phi^{k,w},\psi_m^k)\right)\nabla_{\boldsymbol{\alpha}_i^k}\log P(\phi_i^{k,r})-\nabla_{\boldsymbol{\alpha}_i^k}f^k(\boldsymbol{\alpha}^k,\psi_m^k)\right\|^2\right]$$

$$\overset{(2)}{\leq} \frac{1}{I^2(I-1)^2}\sum_{r=1}^I\mathbb{E}_{\Phi^{k,r}}\left[\left\|\sum_{\substack{w=1\\w\neq r}}^I\left(\mathcal{L}(\Phi^{k,r},\psi_m^k)-\mathcal{L}(\Phi^{k,w},\psi_m^k)\right)\nabla_{\boldsymbol{\alpha}_i^k}\log P(\phi_i^{k,r})\right\|^2\right]$$

$$\overset{(3)}{\leq} \frac{4G^2}{I^2(I-1)^2}\sum_{r=1}^I\mathbb{E}_{\Phi^{k,r}}\left[\left\|\sum_{\substack{w=1\\w\neq r}}^I\nabla_{\boldsymbol{\alpha}_i^k}\log P(\phi_i^{k,r})\right\|^2\right]\overset{(4)}{\leq}\frac{4G^2N}{I(I-1)\tau^2\nu^2}\overset{(5)}{\leq}\frac{8G^2N}{I^2\tau^2\nu^2}$$

where (1) uses

$$\mathbb{E}_{\{\Phi^{k,r}\}_{r=1}^I}\left[\frac{1}{I-1}\sum_{\substack{w=1\\w\neq r}}^I\left(\mathcal{L}(\Phi^{k,r},\psi_m^k)-\mathcal{L}(\Phi^{k,w},\psi_m^k)\right)\nabla_{\boldsymbol{\alpha}_i^k}\log P(\phi_i^{k,r})-\nabla_{\boldsymbol{\alpha}_i^k}f^k(\boldsymbol{\alpha}^k,\psi_m^k)\right]$$

$$= \frac{1}{I-1}\sum_{\substack{w=1\\w\neq r}}^I\mathbb{E}_{\Phi^{k,r},\Phi^{k,w},w\neq r}\left[\left(\mathcal{L}(\Phi^{k,r},\psi_m^k)-\mathcal{L}(\Phi^{k,w},\psi_m^k)\right)\nabla_{\boldsymbol{\alpha}_i^k}\log P(\phi_i^{k,r})\right]-\nabla_{\boldsymbol{\alpha}_i^k}f^k(\boldsymbol{\alpha}^k,\psi_m^k)$$

$$= \frac{1}{I-1}\sum_{\substack{w=1\\w\neq r}}^I\mathbb{E}_{\Phi^{k,r}}\left[\mathcal{L}(\Phi^{k,r},\psi_m^k)\nabla_{\boldsymbol{\alpha}_i^k}\log P(\phi_i^{k,r})\right]-\nabla_{\boldsymbol{\alpha}_i^k}f^k(\boldsymbol{\alpha}^k,\psi_m^k)$$

$$= 0$$

and the independence of each sampling for $\Phi^k$. (2) uses inequality $\mathbb{E} \|a - \mathbb{E}a\|^2 \leq \mathbb{E} \|a\|^2$; (3) uses Assumption 3.2; (4) uses $\alpha_{i,j}^{k,r} \geq \nu > 0$ and (5):

$$
\nabla_{\boldsymbol{\alpha}_i^k} \log P(\phi_i^{k,r}) \leq \sqrt{N \cdot \max \left\{ \left| \frac{1 - p_{i,j_i}^{k,r}}{\tau \alpha_{i,j_i}^{k,r}} \right|, \left| -\frac{p_{i,j}^{k,r}}{\tau \alpha_{i,j}^{k,r}} \right| \right\}^2} \leq \sqrt{\frac{N}{\tau^2 \nu^2}}
$$

5) is because when $I \geq 2$:

$$
\frac{1}{I(I-1)} \leq \frac{2}{I^2}
$$

$\square$

The following lemma shows that the $\mathbb{E}_{\Phi^k \sim GS(\boldsymbol{\alpha}^k)} \left[ \mathcal{L}(\Phi^k, \Psi^k) \right]$ is L-smooth for $\boldsymbol{\alpha}$. This is crucial for the later convergence analysis of BDPL and Fed-BDPL and is a necessity for convergence proofs.

**Lemma A.2. L-smooth for $\boldsymbol{\alpha}^k$:** *At the $k$-th client, the $\Phi^k$ is sampled from probability matrix $\boldsymbol{p}^k$, and $\boldsymbol{p}^k = GS(\boldsymbol{\alpha}^k)$, $\alpha_{i,j} \geq \nu > 0$ for $i = 1, ..., n$ and $j = 1, ..., N$, $\tau > 0$ is the temperature parameter, $\mathbb{E}_{\Phi^k \sim GS(\boldsymbol{\alpha}^k)} \left[ \mathcal{L}(\Phi^k, \Psi^k) \right]$ is L-smooth for $\boldsymbol{\alpha}^k$ and $L = \frac{nGN(\tau+1)}{\tau^2 \nu^2}$, and then for $t$-th iteration, the following inequality is satisfied:*

$$
\mathbb{E}_{\Phi_{t+1}^k \sim GS(\boldsymbol{\alpha}_{t+1}^k)} \left[ \mathcal{L}(\Phi_{t+1}^k, \Psi^k) \right] - \mathbb{E}_{\Phi_t^k \sim GS(\boldsymbol{\alpha}_t^k)} \left[ \mathcal{L}(\Phi_t^k, \Psi^k) \right]
$$

$$
\leq \left\langle \nabla_{\boldsymbol{\alpha}^k} \mathbb{E}_{\Phi_t^k \sim GS(\boldsymbol{\alpha}_t^k)} \left[ \mathcal{L}(\Phi_t^k, \Psi^k) \right], \boldsymbol{\alpha}_{t+1}^k - \boldsymbol{\alpha}_t^k \right\rangle + \frac{L}{2} \left\| \boldsymbol{\alpha}_{t+1}^k - \boldsymbol{\alpha}_t^k \right\|^2
$$

*Proof.* The objective function:

$$
\mathbb{E}_{\Phi^k \sim GS(\boldsymbol{\alpha}^k)} \left[ \mathcal{L}(\Phi^k, \Psi^k) \right] = \sum_{\phi_1^k \sim GS(\boldsymbol{\alpha}_1^k)} \cdots \sum_{\phi_n^k \sim GS(\boldsymbol{\alpha}_n^k)} \left( \mathcal{L}(\Phi^k, \Psi^k) \prod_{i=1}^n P(\phi_i^k) \right)
$$

We can compute the Hessian of the objective function, $\forall i', i'' \in 1, \cdots, n$ and $j', j'' \in 1, \cdots, N$:
If $i' \neq i''$:

$$
\frac{\partial^2}{\partial \alpha_{i',j'} \partial \alpha_{i'',j''}} \mathbb{E}_{\Phi^k \sim GS(\boldsymbol{\alpha}^k)} \left[ \mathcal{L}(\Phi^k, \Psi^k) \right]
$$

$$
= \sum_{\phi_1^k \sim \boldsymbol{p}_1^k} \cdots \sum_{\phi_n^k \sim \boldsymbol{p}_n^k} \left( \mathcal{L}(\Phi^k, \Psi^k) \frac{\partial^2}{\partial \alpha_{i',j'} \partial \alpha_{i'',j''}} \prod_{i=1}^n P(\phi_i^k) \right)
$$

$$
= \sum_{\phi_1^k \sim \boldsymbol{p}_1^k} \cdots \sum_{\phi_{i'-1}^k \sim \boldsymbol{p}_{i'-1}^k} \sum_{\phi_{i'+1}^k \sim \boldsymbol{p}_{i'+1}^k} \cdots \sum_{\phi_{i''-1}^k \sim \boldsymbol{p}_{i''-1}^k} \sum_{\phi_{i''+1}^k \sim \boldsymbol{p}_{i''+1}^k} \cdots \sum_{\phi_n^k \sim \boldsymbol{p}_n^k}
$$

$$
\left( \sum_{\phi_{i'}^k \sim \boldsymbol{p}_{i'}^k, \phi_{i''}^k \sim \boldsymbol{p}_{i''}^k} \left( \mathcal{L}(\Phi^k, \Psi^k) \frac{\partial^2}{\partial \alpha_{i',j'} \partial \alpha_{i'',j''}} \prod_{i=1}^n P(\phi_i^k) \right) \right)
$$

$$
= \sum_{\phi_1^k \sim \boldsymbol{p}_1^k} \cdots \sum_{\phi_{i'-1}^k \sim \boldsymbol{p}_{i'-1}^k} \sum_{\phi_{i'+1}^k \sim \boldsymbol{p}_{i'+1}^k} \cdots \sum_{\phi_{i''-1}^k \sim \boldsymbol{p}_{i''-1}^k} \sum_{\phi_{i''+1}^k \sim \boldsymbol{p}_{i''+1}^k} \cdots \sum_{\phi_n^k \sim \boldsymbol{p}_n^k}
$$

$$
\left( \mathcal{L}(\Phi^k, \Psi^k) \frac{\partial^2 \left[ P(\phi_{i'}^k) P(\phi_{i''}^k) \right]}{\partial \alpha_{i',j'} \partial \alpha_{i'',j''}} \prod_{\substack{i=1 \\ i \neq i' \\ i \neq i''}}^n P(\phi_i^k) \right)
$$

Then:

$$\frac{\partial^2 \left[P(\phi_{i'}^k)P(\phi_{i''}^k)\right]}{\partial \alpha_{i',j'} \partial \alpha_{i'',j''}} = \begin{cases} \frac{1-p_{i',j_i'}^k}{\tau \alpha_{i',j_i'}^k} \cdot \frac{1-p_{i'',j_i''}^k}{\tau \alpha_{i'',j_i''}^k}, & \text{if } j' = j_i' \text{ and } j'' = j_i'' \\ \frac{1-p_{i',j_i'}^k}{\tau \alpha_{i',j_i'}^k} \cdot \left(-\frac{p_{i'',j''}^k}{\tau \alpha_{i'',j''}^k}\right), & \text{if } j' = j_i' \text{ and } j'' \neq j_i'' \\ -\frac{p_{i',j'}^k}{\tau \alpha_{i',j'}^k} \cdot \frac{1-p_{i'',j_i''}^k}{\tau \alpha_{i'',j_i''}^k}, & \text{if } j' \neq j_i' \text{ and } j'' = j_i'' \\ -\frac{p_{i',j'}^k}{\tau \alpha_{i',j'}^k} \cdot \left(-\frac{p_{i'',j''}^k}{\tau \alpha_{i'',j''}^k}\right), & \text{if } j' \neq j_i' \text{ and } j'' \neq j_i'' \end{cases} \tag{23}$$

Based on $\alpha_{i,j}^k \geq \nu > 0$:

$$\left| \frac{\partial^2 \left[P(\phi_{i'}^k)P(\phi_{i''}^k)\right]}{\partial \alpha_{i',j'} \partial \alpha_{i'',j''}} \right| \leq \frac{1}{\tau^2 \nu^2} \tag{24}$$

Further, based on Assumption 3.2:

$$\left| \frac{\partial^2}{\partial \alpha_{i',j'} \partial \alpha_{i'',j''}} \mathbb{E}_{\Phi^k \sim \text{GS}(\boldsymbol{\alpha}^k)} \left[\mathcal{L}(\Phi^k, \Psi^k)\right] \right|$$

$$\leq \sum_{\phi_1^k \sim \boldsymbol{p}_1^k} \cdots \sum_{\phi_{i'-1}^k \sim \boldsymbol{p}_{i'-1}^k} \sum_{\phi_{i'+1}^k \sim \boldsymbol{p}_{i'+1}^k} \cdots \sum_{\phi_{i''-1}^k \sim \boldsymbol{p}_{i''-1}^k} \sum_{\phi_{i''+1}^k \sim \boldsymbol{p}_{i''+1}^k} \cdots \sum_{\phi_n^k \sim \boldsymbol{p}_n^k} \left( \mathcal{L}(\Phi^k, \Psi^k) \prod_{\substack{i=1 \\ i \neq i' \\ i \neq i''}}^{n} P(\phi_i^k) \right) \cdot \frac{1}{\tau^2 \nu^2}$$

$$\leq \frac{G}{\tau^2 \nu^2}$$

If $i' = i''$:

$$\frac{\partial^2}{\partial \alpha_{i',j'} \partial \alpha_{i',j''}} \mathbb{E}_{\Phi^k \sim \text{GS}(\boldsymbol{\alpha}^k)} \left[\mathcal{L}(\Phi^k, \Psi^k)\right]$$

$$= \sum_{\phi_1^k \sim \boldsymbol{p}_1^k} \cdots \sum_{\phi_{i'-1}^k \sim \boldsymbol{p}_{i'-1}^k} \sum_{\phi_{i'+1}^k \sim \boldsymbol{p}_{i'+1}^k} \cdots \sum_{\phi_n^k \sim \boldsymbol{p}_n^k} \left( \mathcal{L}(\Phi^k, \Psi^k) \frac{\partial^2 \left[P(\phi_{i'}^k)\right]}{\partial \alpha_{i',j'} \partial \alpha_{i',j''}} \prod_{\substack{i=1 \\ i \neq i'}}^{n} P(\phi_i^k) \right)$$

Similar to the analysis in case $i' \neq i''$, we can get:

$$\left| \frac{\partial^2 \left[P(\phi_{i'}^k)\right]}{\partial \alpha_{i',j'} \partial \alpha_{i',j''}} \right| \leq \max\{p, 1-p\} \cdot \frac{(\tau+1)}{\tau^2 \nu^2} \leq \frac{\tau+1}{\tau^2 \nu^2}$$

$$\left| \frac{\partial^2}{\partial \alpha_{i',j'} \partial \alpha_{i',j''}} \mathbb{E}_{\Phi^k \sim \text{GS}(\boldsymbol{\alpha}^k)} \left[\mathcal{L}(\Phi^k, \Psi^k)\right] \right| \leq \frac{(\tau+1) G}{\tau^2 \nu^2} \tag{25}$$

Finally, with $H(\boldsymbol{\alpha})$ denoting the Hessian matrix of $\mathbb{E}_{\Phi^k \sim \text{GS}(\boldsymbol{\alpha}^k)} \left[\mathcal{L}(\Phi^k, \Psi^k)\right]$, we can get:

$$\|H(\boldsymbol{\alpha})\|_2 \leq \|H(\boldsymbol{\alpha})\|_F \leq \sqrt{n(n-1)N^2 \left(\frac{G}{\tau^2 \nu^2}\right)^2 + nN^2 \left(\frac{(\tau+1) G}{\tau^2 \nu^2}\right)^2} \leq \frac{nGN(\tau+1)}{\tau^2 \nu^2} \tag{26}$$

According to Lemma 1.2.2 in (Nesterov et al., 2018), $\mathbb{E}_{\Phi^k \sim \text{GS}(\boldsymbol{\alpha}^k)} \left[\mathcal{L}(\Phi^k, \Psi^k)\right]$ is L-smooth for $\boldsymbol{\alpha}^k$ and $L = \frac{nGN(\tau+1)}{\tau^2 \nu^2}$. $\quad \square$

**Lemma A.3.** *Convergence of BDPL*: *Suppose Assumption [3.1] and [3.2] hold, for $t = 0, ..., T-1$, $\alpha_{i,j} \geq \nu > 0$ for $i = 1, ..., n$ and $j = 1, ..., N$, $\tau > 0$ is the temperature parameter, $\mathbb{E}_{\Phi^k \sim GS(\boldsymbol{\alpha}^k)}\left[\mathcal{L}(\Phi^k, \Psi^k)\right]$ is L-smooth for $\boldsymbol{\alpha}^k$ and $L = \frac{nGN(\tau+1)}{\tau^2 \nu^2}$, $\sigma_\psi^2$ is the variance of the stochastic gradient, $\sigma_\alpha^2 = \frac{8G^2 N}{\tau^2 \nu^2}$ is the variance of the variance-reduced policy gradient. Let $\eta \leq \frac{1}{L}$, then the BDPL's full gradient $\nabla F^k(\boldsymbol{\alpha}_t^k, \Psi^k)$ satisfies the following inequality:*

$$\frac{1}{T} \sum_{t=0}^{T-1} \left\| \nabla_{\boldsymbol{\alpha}^k} F^k(\boldsymbol{\alpha}_t^k, \Psi^k) \right\|^2 \leq \frac{4G}{T\eta} + \frac{2\eta n L \sigma_\psi^2}{B^k} + \frac{2\eta n L \sigma_\alpha^2}{I^2} \tag{27}$$

*Proof.* According to Lemma [A.2]:

$$\mathbb{E}_{\Phi_{t+1}^k \sim GS(\boldsymbol{\alpha}_{t+1}^k)}\left[\mathcal{L}(\Phi_{t+1}^k, \Psi^k)\right] - \mathbb{E}_{\Phi_t^k \sim GS(\boldsymbol{\alpha}_t^k)}\left[\mathcal{L}(\Phi_t^k, \Psi^k)\right]$$

$$\leq \left\langle \nabla_{\boldsymbol{\alpha}^k} \mathbb{E}_{\Phi_t^k \sim GS(\boldsymbol{\alpha}_t^k)}\left[\mathcal{L}(\Phi_t^k, \Psi^k)\right], \boldsymbol{\alpha}_{t+1}^k - \boldsymbol{\alpha}_t^k \right\rangle + \frac{L}{2}\left\| \boldsymbol{\alpha}_{t+1}^k - \boldsymbol{\alpha}_t^k \right\|^2$$

$$\leq \sum_{i=1}^n \left[ \left\langle \nabla_{\boldsymbol{\alpha}_i^k} F^k(\boldsymbol{\alpha}_t^k, \Psi^k), -\eta \cdot \hat{\nabla}_{\boldsymbol{\alpha}_i^k} f^k(\boldsymbol{\alpha}_t^k, \mathcal{B}_t^k) \right\rangle + \frac{L\eta^2}{2}\left\| \hat{\nabla}_{\boldsymbol{\alpha}_i^k} f^k(\boldsymbol{\alpha}_t^k, \mathcal{B}_t^k) \right\|^2 \right]$$

Both sides take expectations for $\mathbb{E}_{\{\Phi_t^r \sim GS(\boldsymbol{\alpha}_t)\}_{r=1}^I}$ and $\mathbb{E}_{\mathcal{B}_t}$ at the same time, we abbreviate $\mathbb{E}_{\{\Phi^r \sim GS(\boldsymbol{\alpha})_{r=1}^I}$ as $\mathbb{E}_{\{\Phi^r\}_{r=1}^I}$:

$$\mathbb{E}_{\{\Phi^r\}_{r=1}^I} \mathbb{E}_{\mathcal{B}_t} \left\{ \mathbb{E}_{\Phi_{t+1}^k \sim GS(\boldsymbol{\alpha}_{t+1}^k)}\left[\mathcal{L}(\Phi_{t+1}^k, \Psi^k)\right] - \mathbb{E}_{\Phi_t^k \sim GS(\boldsymbol{\alpha}_t^k)}\left[\mathcal{L}(\Phi_t^k, \Psi^k)\right] \right\}$$

$$\leq \mathbb{E}_{\{\Phi^r\}_{r=1}^I} \mathbb{E}_{\mathcal{B}_t} \left\{ \sum_{i=1}^n \left[ \left\langle \nabla_{\boldsymbol{\alpha}_i^k} F^k(\boldsymbol{\alpha}_t^k, \Psi^k), -\eta \cdot \hat{\nabla}_{\boldsymbol{\alpha}_i^k} f^k(\boldsymbol{\alpha}_t^k, \mathcal{B}_t^k) \right\rangle + \frac{L\eta^2}{2}\left\| \hat{\nabla}_{\boldsymbol{\alpha}_i^k} f^k(\boldsymbol{\alpha}_t^k, \mathcal{B}_t^k) \right\|^2 \right] \right\}$$

$$= \sum_{i=1}^n \underbrace{\mathbb{E}_{\{\Phi^r\}_{r=1}^I} \mathbb{E}_{\mathcal{B}_t} \left[ \left\langle \nabla_{\boldsymbol{\alpha}_i^k} F^k(\boldsymbol{\alpha}_t^k, \Psi^k), -\eta \cdot \hat{\nabla}_{\boldsymbol{\alpha}_i^k} f^k(\boldsymbol{\alpha}_t^k, \mathcal{B}_t^k) \right\rangle + \frac{L\eta^2}{2}\left\| \hat{\nabla}_{\boldsymbol{\alpha}_i^k} f^k(\boldsymbol{\alpha}_t^k, \mathcal{B}_t^k) \right\|^2 \right]}_{a)}$$

For a):

$$\mathbb{E}_{\{\Phi^r\}_{r=1}^I} \mathbb{E}_{\mathcal{B}_t} \left[ \left\langle \nabla_{\boldsymbol{\alpha}_i^k} F^k(\boldsymbol{\alpha}_t^k, \Psi^k), -\eta \cdot \hat{\nabla}_{\boldsymbol{\alpha}_i^k} f^k(\boldsymbol{\alpha}_t^k, \mathcal{B}_t^k) \right\rangle + \frac{L\eta^2}{2}\left\| \hat{\nabla}_{\boldsymbol{\alpha}_i^k} f^k(\boldsymbol{\alpha}_t^k, \mathcal{B}_t^k) \right\|^2 \right]$$

$$= \left\langle \nabla_{\boldsymbol{\alpha}_i^k} F^k(\boldsymbol{\alpha}_t^k, \Psi^k), -\eta \cdot \mathbb{E}_{\{\Phi^r\}_{r=1}^I} \mathbb{E}_{\mathcal{B}_t} \left[ \hat{\nabla}_{\boldsymbol{\alpha}_i^k} f^k(\boldsymbol{\alpha}_t^k, \mathcal{B}_t^k) \right] \right\rangle + \frac{L\eta^2}{2} \mathbb{E}_{\{\Phi^r\}_{r=1}^I} \mathbb{E}_{\mathcal{B}_t} \left[ \left\| \hat{\nabla}_{\boldsymbol{\alpha}_i^k} f^k(\boldsymbol{\alpha}_t^k, \mathcal{B}_t^k) \right\|^2 \right]$$

$$\overset{(1)}{=} \left\langle \nabla_{\boldsymbol{\alpha}_i^k} F^k(\boldsymbol{\alpha}_t^k, \Psi^k), -\eta \cdot \mathbb{E}_{\mathcal{B}_t} \mathbb{E}_{\{\Phi^r\}_{r=1}^I} \left[ \hat{\nabla}_{\boldsymbol{\alpha}_i^k} f^k(\boldsymbol{\alpha}_t^k, \mathcal{B}_t^k) \right] \right\rangle$$

$$+ \frac{L\eta^2}{2} \left\{ \left\| \mathbb{E}_{\{\Phi^r\}_{r=1}^I} \mathbb{E}_{\mathcal{B}_t} \left[ \hat{\nabla}_{\boldsymbol{\alpha}_i^k} f^k(\boldsymbol{\alpha}_t^k, \mathcal{B}_t^k) \right] \right\|^2 + \text{Var}_{\mathcal{B}_t, \{\Phi^r\}_{r=1}^I} \left[ \hat{\nabla}_{\boldsymbol{\alpha}_i^k} f^k(\boldsymbol{\alpha}_t^k, \mathcal{B}_t^k) \right] \right\}$$

$$\overset{(2)}{=} -\eta \cdot \left\langle \nabla_{\boldsymbol{\alpha}_i^k} F^k(\boldsymbol{\alpha}_t^k, \Psi^k), \nabla_{\boldsymbol{\alpha}_i^k} F^k(\boldsymbol{\alpha}_t^k, \Psi^k) \right\rangle + \frac{L\eta^2}{2}\left\| \nabla_{\boldsymbol{\alpha}_i^k} F^k(\boldsymbol{\alpha}_t^k, \Psi^k) \right\|^2$$

$$+ \frac{L\eta^2}{2} \mathbb{E}_{\{\Phi^r\}_{r=1}^I} \mathbb{E}_{\mathcal{B}_t} \left[ \left\| \hat{\nabla}_{\boldsymbol{\alpha}_i^k} f^k(\boldsymbol{\alpha}_t^k, \mathcal{B}_t^k) - \mathbb{E}_{\{\Phi^r\}_{r=1}^I} \mathbb{E}_{\mathcal{B}_t} \left[ \hat{\nabla}_{\boldsymbol{\alpha}_i^k} f^k(\boldsymbol{\alpha}_t^k, \mathcal{B}_t^k) \right] \right\|^2 \right]$$

$$\overset{(3)}{\leq} \left( \frac{L\eta^2}{2} - \eta \right) \left\| \nabla_{\boldsymbol{\alpha}_i^k} F^k(\boldsymbol{\alpha}_t^k, \Psi^k) \right\|^2$$

$$+ L\eta^2 \cdot \mathbb{E}_{\{\Phi^r\}_{r=1}^I} \mathbb{E}_{\mathcal{B}_t} \left[ \left\| \hat{\nabla}_{\boldsymbol{\alpha}_i^k} f^k(\boldsymbol{\alpha}_t^k, \mathcal{B}_t^k) - \mathbb{E}_{\mathcal{B}_t} \left[ \hat{\nabla}_{\boldsymbol{\alpha}_i^k} f^k(\boldsymbol{\alpha}_t^k, \mathcal{B}_t^k) \right] \right\|^2 \right]$$

$$+ L\eta^2 \cdot \mathbb{E}_{\mathcal{B}_t} \mathbb{E}_{\{\Phi^r\}_{r=1}^I} \left[ \left\| \mathbb{E}_{\mathcal{B}_t} \left[ \hat{\nabla}_{\boldsymbol{\alpha}_i^k} f^k(\boldsymbol{\alpha}_t^k, \mathcal{B}_t^k) \right] - \mathbb{E}_{\{\Phi^r\}_{r=1}^I} \mathbb{E}_{\mathcal{B}_t} \left[ \hat{\nabla}_{\boldsymbol{\alpha}_i^k} f^k(\boldsymbol{\alpha}_t^k, \mathcal{B}_t^k) \right] \right\|^2 \right]$$

$$= \left( \frac{L\eta^2}{2} - \eta \right) \left\| \nabla_{\boldsymbol{\alpha}_i^k} F^k(\boldsymbol{\alpha}_t^k, \Psi^k) \right\|^2 + L\eta^2 \cdot \mathbb{E}_{\{\Phi^r\}_{r=1}^I} \text{Var}_{\mathcal{B}_t} \left[ \hat{\nabla}_{\boldsymbol{\alpha}_i^k} f^k(\boldsymbol{\alpha}_t^k, \mathcal{B}_t^k) \right]$$

$$+ L\eta^2 \cdot \mathbb{E}_{\mathcal{B}_t} \text{Var}_{\{\Phi^r\}_{r=1}^I} \left[ \mathbb{E}_{\mathcal{B}_t} \left[ \hat{\nabla}_{\boldsymbol{\alpha}_i^k} f^k(\boldsymbol{\alpha}_t^k, \mathcal{B}_t^k) \right] \right]$$

$$\overset{(4)}{\leq} \left(\frac{L\eta^2}{2} - \eta\right) \left\|\nabla_{\boldsymbol{\alpha}_i^k} F^k(\boldsymbol{\alpha}_t^k, \Psi^k)\right\|^2 + L\eta^2 \cdot \frac{\sigma_\psi^2}{B} + L\eta^2 \cdot \mathbb{E}_{\mathcal{B}_t} \text{Var}_{\{\Phi^r\}_{r=1}^I} \left[\hat{\nabla}_{\boldsymbol{\alpha}_i} f^k(\boldsymbol{\alpha}_t^k, \Psi^k)\right]$$

$$\overset{(5)}{\leq} \left(\frac{L\eta^2}{2} - \eta\right) \left\|\nabla_{\boldsymbol{\alpha}_i^k} F^k(\boldsymbol{\alpha}_t^k, \Psi^k)\right\|^2 + L\eta^2 \cdot \frac{\sigma_\psi^2}{B} + L\eta^2 \cdot \frac{\sigma_\alpha^2}{I^2}$$

where (1) uses the independence between $\Psi$ sampling and $\Phi$ sampling; (2) use the unbiasedness of stochastic gradient and variance-reduced policy gradient in Assumption 3.1 and Lemma A.1; (3) use the inequality $\|a + b\|^2 \leq 2\|a\|^2 + 2\|b\|^2$; (4) and (5) use the bounded variance of stochastic gradient and variance-reduced policy gradient in Assumption 3.1 and Lemma A.1.

Then:

$$\mathbb{E}_{\Phi_{t+1} \sim \text{GS}(\boldsymbol{\alpha}_{t+1}^k)} \left[\mathcal{L}(\Phi_{t+1}^k, \Psi^k)\right] - \mathbb{E}_{\Phi_t^k \sim \text{GS}(\boldsymbol{\alpha}_t^k)} \left[\mathcal{L}(\Phi_t^k, \Psi^k)\right]$$

$$\leq \sum_{i=1}^n \left[\left(\frac{L\eta^2}{2} - \eta\right) \left\|\nabla_{\boldsymbol{\alpha}_i^k} F^k(\boldsymbol{\alpha}_t^k, \Psi^k)\right\|^2 + L\eta^2 \cdot \frac{\sigma_\psi^2}{B} + L\eta^2 \cdot \frac{\sigma_\alpha^2}{I^2}\right]$$

We combine the gradient of $\boldsymbol{\alpha}_i$ for each prompt token:

$$\mathbb{E}_{\Phi_{t+1}^k \sim \text{GS}(\boldsymbol{\alpha}_{t+1})} \left[\mathcal{L}(\Phi_{t+1}^k, \Psi^k)\right] - \mathbb{E}_{\Phi_t^k \sim \text{GS}(\boldsymbol{\alpha}_t^k)} \left[\mathcal{L}(\Phi_t^k, \Psi^k)\right]$$

$$\leq \left(\frac{L\eta^2}{2} - \eta\right) \left\|\nabla_{\boldsymbol{\alpha}^k} F^k(\boldsymbol{\alpha}_t^k, \Psi^k)\right\|^2 + nL\eta^2 \cdot \frac{\sigma_\psi^2}{B} + nL\eta^2 \cdot \frac{\sigma_\alpha^2}{I^2}$$

where $\boldsymbol{\alpha}^k = (\boldsymbol{\alpha}_1^k, \cdots, \boldsymbol{\alpha}_i^k, \cdots \boldsymbol{\alpha}_n^k)$.
We let $\eta \leq \frac{1}{L}$, then both sides accumulate with respect to $t = 0, 1, \cdots, T-1$ and divide by $T$:

$$\frac{1}{T} \sum_{t=0}^{T-1} \left(\eta - \frac{L\eta^2}{2}\right) \left\|\nabla_{\boldsymbol{\alpha}^k} F^k(\boldsymbol{\alpha}_t^k, \Psi^k)\right\|^2$$

$$\leq \frac{1}{T} \sum_{t=0}^{T-1} \left[\mathbb{E}_{\Phi_t^k \sim \text{GS}(\boldsymbol{\alpha}_t^k)} \left[\mathcal{L}(\Phi_t^k, \Psi^k)\right] - \mathbb{E}_{\Phi_{t+1}^k \sim \text{GS}(\boldsymbol{\alpha}_{t+1}^k)} \left[\mathcal{L}(\Phi_{t+1}^k, \Psi^k)\right]\right] + \frac{nL\eta^2\sigma_\psi^2}{B} + \frac{nL\eta^2\sigma_\alpha^2}{I^2}$$

Then,

$$\frac{1}{T} \sum_{t=0}^{T-1} \left\|\nabla_{\boldsymbol{\alpha}^k} F^k(\boldsymbol{\alpha}_t^k, \Psi^k)\right\|^2$$

$$\leq \frac{\mathbb{E}_{\Phi_0^k \sim \text{GS}(\boldsymbol{\alpha}_0^k)} \left[\mathcal{L}(\Phi_0^k, \Psi^k)\right] - \mathbb{E}_{\Phi_T^k \sim \text{GS}(\boldsymbol{\alpha}_T^k)} \left[\mathcal{L}(\Phi_T^k, \Psi^k)\right]}{T} \frac{2}{2\eta - L\eta^2} + \frac{2\eta}{2 - L\eta}\left(\frac{nL\sigma_\psi^2}{B} + \frac{nL\sigma_\alpha^2}{I^2}\right)$$

$$\leq \frac{2\left(\mathbb{E}_{\Phi_0^k \sim \text{GS}(\boldsymbol{\alpha}_0^k)} \left[\mathcal{L}(\Phi_0^k, \Psi^k)\right] - \mathbb{E}_{\Phi_T^k \sim \text{GS}(\boldsymbol{\alpha}_T^k)} \left[\mathcal{L}(\Phi_T^k, \Psi^k)\right]\right)}{T\eta} + \frac{2\eta nL\sigma_\psi^2}{B} + \frac{2\eta nL\sigma_\alpha^2}{I^2}$$

According to Assumption 3.2 , $\mathbb{E}_{\Phi_0^k \sim \text{GS}(\boldsymbol{\alpha}_0^k)} \left[\mathcal{L}(\Phi_0^k, \Psi^k)\right] - \inf_t \mathbb{E}_{\Phi_t^k \sim \text{GS}(\boldsymbol{\alpha}_t^k)} \left[\mathcal{L}(\Phi_t^k, \Psi^k)\right] \leq 2G$, then:

$$\frac{1}{T} \sum_{t=0}^{T-1} \left\|\nabla_{\boldsymbol{\alpha}^k} F^k(\boldsymbol{\alpha}_t^k, \Psi^k)\right\|^2 \leq \frac{4G}{T\eta} + \frac{2\eta nL\sigma_\psi^2}{B} + \frac{2\eta nL\sigma_\alpha^2}{I^2}$$

$\square$

*Remark* A.4. The convergence term of BDPL consists of three parts. The first term is typical of first-order optimization algorithms converging to non-convex functions. The second term is the stochasticity due to random mini-batch gradients. The first term and second term can be combined when $\eta = \frac{c}{\sqrt{T}}$ and $c = \sqrt{\frac{2B^kG}{nL\sigma_\psi^2}}$:

$$\frac{4G}{T\eta} + \frac{2\eta nL\sigma_\psi^2}{B^k} = \frac{4}{\sqrt{T}}\sqrt{\frac{2nL\sigma_\psi^2 G}{B}},$$

which can decrease as the number of iterations and mini-batch size increase (Reddi et al., 2016); and the third term is the stochasticity due to prompt sampling, which decreases with the number of prompt samples.

*Remark* A.5. According to the definition of Gumbel-Softmax (2), there is randomness about $\boldsymbol{u} = \{u_i\}_{i=1}^n \sim \text{Uniform}(\boldsymbol{0}, \boldsymbol{1}_n)$ in $\nabla_{\boldsymbol{\alpha}} F^k(\boldsymbol{\alpha}_t^k, \Psi^k)$ i.e. $\boldsymbol{p}^k = \text{GS}(\boldsymbol{\alpha}^k, \boldsymbol{u})$ in fact and $\Phi^k \sim \boldsymbol{p}^k$, we can further discuss the result in Lemma A.3:

$$\mathbb{E}_{\boldsymbol{u}} \left\| \nabla_{\boldsymbol{\alpha}^k} F^k(\boldsymbol{\alpha}_t^k, \Psi^k) \right\|^2 = \int_{\boldsymbol{0}}^{\boldsymbol{1}} \left\| \nabla_{\boldsymbol{\alpha}^k} F^k(\boldsymbol{\alpha}_t^k, \Psi^k) \right\|^2 \boldsymbol{1}_n d\boldsymbol{u} \le \left\| \nabla_{\boldsymbol{\alpha}^k} F^k(\boldsymbol{\alpha}_t^k, \Psi^k) \right\|^2$$

Considering the randomness of $\boldsymbol{u}$, we can still obtain the convergence of BDPL. In subsequent analyzes of Fed-BDPL, we can obtain similar result for $\boldsymbol{u}$.

**Corollary A.6.** *Convergence rate of BDPL*: *Let* $\eta = \min\left\{\frac{1}{L}, \frac{1}{\sqrt{T}}\right\}$, *we can get the following convergence rate for BDPL:*

$$\frac{1}{T} \sum_{t=0}^{T-1} \left\| \nabla_{\boldsymbol{\alpha}^k} F^k(\boldsymbol{\alpha}_t^k, \Psi^k) \right\|^2 \le \mathcal{O}(\frac{1}{\sqrt{T}}) \tag{28}$$

*Proof.* Convergence rate:

$$\frac{1}{T} \sum_{t=0}^{T-1} \left\| \nabla_{\boldsymbol{\alpha}^k} F^k(\boldsymbol{\alpha}_t^k, \Psi^k) \right\|^2 = \frac{4G}{\sqrt{T}} + \frac{1}{\sqrt{T}} \cdot \frac{2nL\sigma_\psi^2}{B} + \frac{1}{\sqrt{T}} \cdot \frac{2nL\sigma_\alpha^2}{I^2} = \mathcal{O}\left(\frac{1}{\sqrt{T}}\right)$$

$\square$

### A.4. Convergence Analysis of Federated Prompt Tuning

Definition in the Federated Black-box Discrete Prompt Learning:

**Data slicing**: The data slices in each client are defined as $D^k = \left\{D^k\right\}_{k=1}^K = \left\{\Psi^k, Y^k\right\}_{k=1}^K$, where $\Psi^k = \left\{\psi_m^k\right\}_{m=1}^{M^k}$ denote the input sentence and $Y^k = \left\{y_m^k\right\}_{m=1}^{M^k}$ denote the label, $k$ is the index of the client, and $m$ is the index of the sample, there are totally $M^k$ samples in the dataset $D^k$. The clients hold the sampling probability vector $\boldsymbol{q} = \left\{q^{[k]}\right\}_{k=1}^K = \left\{\frac{M^k}{M}\right\}_{k=1}^K$. $K_*$ is the number of clients selected and $U_t$ is the set of corresponding clients.

**Average parameter $\boldsymbol{\alpha}$**:

$$\boldsymbol{\alpha}_t = \frac{1}{K_*} \sum_{k \in U_t} \boldsymbol{\alpha}_t^k$$

$$\boldsymbol{\alpha}_{i,(t)} = \frac{1}{K_*} \sum_{k \in U_t} \boldsymbol{\alpha}_{i,(t)}^k$$

**Average loss**:

$$\mathbb{E}_{\Phi_t \sim \text{GS}(\boldsymbol{\alpha}_t)} \left[\mathcal{L}(\Phi_t, \Psi)\right] = \sum_{k=1}^K q^{[k]} \cdot \mathbb{E}_{\Phi_t^k \sim \text{GS}(\boldsymbol{\alpha}_t^k)} \left[\mathcal{L}(\Phi_t^k, \Psi^k)\right]$$

**Average gradient**:

$$\nabla_{\boldsymbol{\alpha}_i} F(\boldsymbol{\alpha}, \Psi^k) \stackrel{def}{=} \sum_{k=1}^K q^{[k]} \cdot \nabla_{\boldsymbol{\alpha}_i} F^k(\boldsymbol{\alpha}, \Psi^k) \tag{29}$$

$$\nabla_{\boldsymbol{\alpha}_i^k} F(\boldsymbol{\alpha}^k, \Psi^k) \stackrel{def}{=} \sum_{k=1}^K q^{[k]} \cdot \nabla_{\boldsymbol{\alpha}_i^k} F^k(\boldsymbol{\alpha}^k, \Psi^k) \tag{30}$$

$$\nabla_{\boldsymbol{\alpha}_i^k} F^*(\boldsymbol{\alpha}^k, \Psi^k) \stackrel{def}{=} \frac{1}{K_*} \sum_{k \in U_t} \nabla_{\boldsymbol{\alpha}_i^k} F^k(\boldsymbol{\alpha}^k, \Psi^k) \tag{31}$$

$$\nabla_{\boldsymbol{\alpha}_i^k} f^*(\boldsymbol{\alpha}^k, \mathcal{B}^k) \stackrel{def}{=} \frac{1}{K_*} \sum_{k \in U_t} \nabla_{\boldsymbol{\alpha}_i^k} f^k(\boldsymbol{\alpha}^k, \mathcal{B}^k) \tag{32}$$

$$\hat{\nabla}_{\boldsymbol{\alpha}_i^k} f^*(\boldsymbol{\alpha}^k, \mathcal{B}^k) \stackrel{def}{=} \frac{1}{K_*} \sum_{k \in U_t} \hat{\nabla}_{\boldsymbol{\alpha}_i} f^k(\boldsymbol{\alpha}^k, \mathcal{B}^k) \tag{33}$$

**Average mini-batch stochastic variance-reduced policy gradient descent**:

$$\boldsymbol{\alpha}_{i,(t+1)} = \boldsymbol{\alpha}_{i,(t)} - \eta \cdot \hat{\nabla}_{\boldsymbol{\alpha}_i^k} f^*(\boldsymbol{\alpha}_t^k, \mathcal{B}^k)$$

Note that when $t \neq sE$, although the above some definitions don't exist in algorithm and experiment, we can still calculate and analyze them. In order to facilitate the analysis of the iteration process, we assume that they exist.

The following lemma shows that the local gradient is biased compared to the global gradient due to the fact that the distribution of data on different clients may be different (heterogeneity), and the bias of the gradient can be analyzed using the bias about $\boldsymbol{\alpha}$ ((Haddadpour & Mahdavi, 2019)).

**Lemma A.7. *Bound bias between local and average $\boldsymbol{\alpha}$*.** *Let $\alpha_{i,j} \geq \nu > 0$ for $i = 1, ..., n$ and $j = 1, ..., N$, $\eta$ is the learning rate, let $\mathbb{E}$ represents $\mathbb{E}_{\{\Phi_t^{k,r} \sim GS(\boldsymbol{\alpha}_t^k)\}_{r=1}^I}$ and $\mathbb{E}_{\mathcal{B}_t^k}$, the bias between the local and the average $\boldsymbol{\alpha}$ can be bounded:*

$$\frac{1}{T} \sum_{t=0}^{T-1} \sum_{k=1}^{K} q^{[k]} \cdot \mathbb{E} \left\| \boldsymbol{\alpha}_{i,(t)} - \boldsymbol{\alpha}_{i,(t)}^k \right\|^2$$

$$\leq E\eta^2 \left( \frac{2\sigma_\psi^2}{B} + \frac{2\sigma_\alpha^2}{I^2} \right) \left( 1 + \frac{1}{K_*} \right) + \frac{2\eta^2 E^2 \lambda}{T} \left( 1 + \frac{1}{K_*} \right) \sum_{t=0}^{T-1} \left\| \nabla_{\boldsymbol{\alpha}_i^k} F(\boldsymbol{\alpha}_t^k, \Psi^k) \right\|^2$$

*Proof.* Define:

$$t_c \triangleq \lfloor \frac{t}{E} \rfloor E$$

$$\boldsymbol{\alpha}_{i,(t_c)} = \frac{1}{K_*} \sum_{k \in U_{t_c}} \boldsymbol{\alpha}_{i,(t_c)}^k$$

Then, for $t_c + 1 \leq t < t_c + E$:

$$\boldsymbol{\alpha}_{i,(t)}^k = \boldsymbol{\alpha}_{i,(t_c)} - \sum_{\rho=t_c}^{t-1} \eta \cdot \hat{\nabla}_{\boldsymbol{\alpha}_i^k} f^k(\boldsymbol{\alpha}_\rho^k, \mathcal{B}_\rho^k) \tag{34}$$

$$\boldsymbol{\alpha}_{i,(t)} = \boldsymbol{\alpha}_{i,(t_c)} - \frac{1}{K_*} \sum_{k \in U_\rho} \sum_{\rho=t_c}^{t-1} \eta \cdot \hat{\nabla}_{\boldsymbol{\alpha}_i^k} f^k(\boldsymbol{\alpha}_\rho^k, \mathcal{B}_\rho^k) \tag{35}$$

For the $k$-th client, let $\mathbb{E}$ represents $\mathbb{E}_{\{\Phi^{k,r}\}_{r=1}^I}$ and $\mathbb{E}_{\mathcal{B}_t^k}$, and we take $\mathbb{E}$ for $\left\| \boldsymbol{\alpha}_{i,(t)} - \boldsymbol{\alpha}_{i,(t)}^k \right\|^2$:

$$\mathbb{E} \left\| \boldsymbol{\alpha}_{i,(t)} - \boldsymbol{\alpha}_{i,(t)}^k \right\|^2$$

$$= \mathbb{E} \left\| \boldsymbol{\alpha}_{i,(t_c)} - \frac{1}{K_*} \sum_{k \in U_\rho} \sum_{\rho=t_c}^{t-1} \eta \cdot \hat{\nabla}_{\boldsymbol{\alpha}_i^k} f^k(\boldsymbol{\alpha}_\rho^k, \mathcal{B}_\rho^k) - \boldsymbol{\alpha}_{i,(t_c)} + \sum_{\rho=t_c}^{t-1} \eta \cdot \hat{\nabla}_{\boldsymbol{\alpha}_i} f^k(\boldsymbol{\alpha}_\rho^k, \mathcal{B}_\rho^k) \right\|^2$$

$$= \mathbb{E} \left\| \sum_{\rho=t_c}^{t-1} \eta \cdot \hat{\nabla}_{\boldsymbol{\alpha}_i^k} f^k(\boldsymbol{\alpha}_\rho^k, \mathcal{B}_\rho^k) - \frac{1}{K_*} \sum_{k \in U_\rho} \sum_{\rho=t_c}^{t-1} \eta \cdot \hat{\nabla}_{\boldsymbol{\alpha}_i^k} f^k(\boldsymbol{\alpha}_\rho^k, \mathcal{B}_\rho^k) \right\|^2$$

$$\stackrel{(1)}{\leq} 2 \left[ \mathbb{E} \left\| \sum_{\rho=t_c}^{t-1} \eta \cdot \hat{\nabla}_{\boldsymbol{\alpha}_i^k} f^k(\boldsymbol{\alpha}_\rho^k, \mathcal{B}_\rho^k) \right\|^2 + \mathbb{E} \left\| \frac{1}{K_*} \sum_{k \in U_\rho} \sum_{\rho=t_c}^{t-1} \eta \cdot \hat{\nabla}_{\boldsymbol{\alpha}_i^k} f^k(\boldsymbol{\alpha}_\rho^k, \mathcal{B}_\rho^k) \right\|^2 \right]$$

$$\overset{(2)}{=} 2\mathbb{E} \left\| \sum_{\rho=t_c}^{t-1} \eta \cdot \hat{\nabla}_{\boldsymbol{\alpha}_i^k} f^k(\boldsymbol{\alpha}_\rho^k, \mathcal{B}_\rho^k) - \mathbb{E}\left[ \sum_{\rho=t_c}^{t-1} \eta \cdot \hat{\nabla}_{\boldsymbol{\alpha}_i^k} f^k(\boldsymbol{\alpha}_\rho^k, \mathcal{B}_\rho^k) \right] \right\|^2 + 2\left\| \mathbb{E}\left[ \sum_{\rho=t_c}^{t-1} \eta \cdot \hat{\nabla}_{\boldsymbol{\alpha}_i^k} f^k(\boldsymbol{\alpha}_\rho^k, \mathcal{B}_\rho^k) \right] \right\|^2$$

$$+ 2\mathbb{E} \left\| \frac{1}{K_*} \sum_{k \in U_\rho} \sum_{\rho=t_c}^{t-1} \eta \cdot \hat{\nabla}_{\boldsymbol{\alpha}_i^k} f^k(\boldsymbol{\alpha}_\rho^k, \mathcal{B}_\rho^k) - \mathbb{E}\left[ \frac{1}{K_*} \sum_{k \in U_\rho} \sum_{\rho=t_c}^{t-1} \eta \cdot \hat{\nabla}_{\boldsymbol{\alpha}_i^k} f^k(\boldsymbol{\alpha}_\rho^k, \mathcal{B}_\rho^k) \right] \right\|^2$$

$$+ 2\left\| \mathbb{E}\left[ \frac{1}{K_*} \sum_{k \in U_\rho} \sum_{\rho=t_c}^{t-1} \eta \cdot \hat{\nabla}_{\boldsymbol{\alpha}_i^k} f^k(\boldsymbol{\alpha}_\rho^k, \mathcal{B}_\rho^k) \right] \right\|^2$$

$$\overset{(3)}{=} 2\mathbb{E} \left\| \sum_{\rho=t_c}^{t-1} \eta \cdot \hat{\nabla}_{\boldsymbol{\alpha}_i^k} f^k(\boldsymbol{\alpha}_\rho^k, \mathcal{B}_\rho^k) - \sum_{\rho=t_c}^{t-1} \eta \cdot \nabla_{\boldsymbol{\alpha}_i^k} F^k(\boldsymbol{\alpha}_\rho^k, \Psi^k) \right\|^2 + 2\left\| \sum_{\rho=t_c}^{t-1} \eta \cdot \nabla_{\boldsymbol{\alpha}_i^k} F^k(\boldsymbol{\alpha}_\rho^k, \Psi^k) \right\|^2$$

$$+ 2\mathbb{E} \left\| \frac{1}{K_*} \sum_{k \in U_\rho} \sum_{\rho=t_c}^{t-1} \eta \cdot \hat{\nabla}_{\boldsymbol{\alpha}_i^k} f^k(\boldsymbol{\alpha}_\rho^k, \mathcal{B}_\rho^k) - \frac{1}{K_*} \sum_{k \in U_\rho} \sum_{\rho=t_c}^{t-1} \eta \cdot \nabla_{\boldsymbol{\alpha}_i^k} F^k(\boldsymbol{\alpha}_\rho^k, \Psi^k) \right\|^2$$

$$+ 2\left\| \frac{1}{K_*} \sum_{k \in U_\rho} \sum_{\rho=t_c}^{t-1} \eta \cdot \nabla_{\boldsymbol{\alpha}_i^k} F^k(\boldsymbol{\alpha}_\rho^k, \Psi^k) \right\|^2$$

$$= 2\mathbb{E} \left\| \sum_{\rho=t_c}^{t-1} \eta \cdot \left[ \hat{\nabla}_{\boldsymbol{\alpha}_i^k} f^k(\boldsymbol{\alpha}_\rho^k, \mathcal{B}_\rho^k) - \nabla_{\boldsymbol{\alpha}_i^k} F^k(\boldsymbol{\alpha}_\rho^k, \Psi^k) \right] \right\|^2 + 2\left\| \sum_{\rho=t_c}^{t-1} \eta \cdot \nabla_{\boldsymbol{\alpha}_i^k} F^k(\boldsymbol{\alpha}_\rho^k, \Psi^k) \right\|^2$$

$$+ 2\mathbb{E} \left\| \frac{1}{K_*} \sum_{k \in U_\rho} \sum_{\rho=t_c}^{t-1} \eta \cdot \left[ \hat{\nabla}_{\boldsymbol{\alpha}_i^k} f^k(\boldsymbol{\alpha}_\rho^k, \mathcal{B}_\rho^k) - \nabla_{\boldsymbol{\alpha}_i^k} F^k(\boldsymbol{\alpha}_\rho^k, \Psi^k) \right] \right\|^2 + 2\left\| \frac{1}{K_*} \sum_{k \in U_\rho} \sum_{\rho=t_c}^{t-1} \eta \cdot \nabla_{\boldsymbol{\alpha}_i^k} F^k(\boldsymbol{\alpha}_\rho^k, \Psi^k) \right\|^2$$

$$\overset{(4)}{=} 2\eta^2 \sum_{\rho=t_c}^{t-1} \mathbb{E} \left\| \hat{\nabla}_{\boldsymbol{\alpha}_i^k} f^k(\boldsymbol{\alpha}_\rho^k, \mathcal{B}_\rho^k) - \nabla_{\boldsymbol{\alpha}_i^k} F^k(\boldsymbol{\alpha}_\rho^k, \Psi^k) \right\|^2 + 2\left\| \sum_{\rho=t_c}^{t-1} \eta \cdot \nabla_{\boldsymbol{\alpha}_i^k} F^k(\boldsymbol{\alpha}_\rho^k, \Psi^k) \right\|^2$$

$$+ \frac{2\eta^2}{K_*^2} \sum_{k \in U_\rho} \sum_{\rho=t_c}^{t-1} \mathbb{E} \left\| \hat{\nabla}_{\boldsymbol{\alpha}_i^k} f^k(\boldsymbol{\alpha}_\rho^k, \mathcal{B}_\rho^k) - \nabla_{\boldsymbol{\alpha}_i^k} F^k(\boldsymbol{\alpha}_\rho^k, \Psi^k) \right\|^2 + 2\left\| \frac{1}{K_*} \sum_{k \in U_\rho} \sum_{\rho=t_c}^{t-1} \eta \cdot \nabla_{\boldsymbol{\alpha}_i^k} F^k(\boldsymbol{\alpha}_\rho^k, \Psi^k) \right\|^2$$

$$\overset{(5)}{=} 2\eta^2 \sum_{\rho=t_c}^{t-1} \mathbb{E} \left\| \hat{\nabla}_{\boldsymbol{\alpha}_i^k} f^k(\boldsymbol{\alpha}_\rho^k, \mathcal{B}_\rho^k) - \nabla_{\boldsymbol{\alpha}_i^k} F^k(\boldsymbol{\alpha}_\rho^k, \Psi^k) \right\|^2 + 2\eta^2(t-t_c) \sum_{\rho=t_c}^{t-1} \left\| \nabla_{\boldsymbol{\alpha}_i^k} F^k(\boldsymbol{\alpha}_\rho^k, \Psi^k) \right\|^2$$

$$+ \frac{2\eta^2}{K_*^2} \sum_{k \in U_\rho} \sum_{\rho=t_c}^{t-1} \mathbb{E} \left\| \hat{\nabla}_{\boldsymbol{\alpha}_i^k} f^k(\boldsymbol{\alpha}_\rho^k, \mathcal{B}_\rho^k) - \nabla_{\boldsymbol{\alpha}_i^k} F^k(\boldsymbol{\alpha}_\rho^k, \Psi^k) \right\|^2 + \frac{2\eta^2(t-t_c)}{K_*} \sum_{k \in U_\rho} \sum_{\rho=t_c}^{t-1} \left\| \nabla_{\boldsymbol{\alpha}_i^k} F^k(\boldsymbol{\alpha}_\rho^k, \Psi^k) \right\|^2$$

$$\overset{(6)}{\le} 2\eta^2 \sum_{\rho=t_c}^{t-1} \left( \frac{2\sigma_\psi^2}{B^k} + \frac{2\sigma_\alpha^2}{I^2} \right) + \frac{2\eta^2}{K_*^2} \sum_{k \in U_\rho} \sum_{\rho=t_c}^{t-1} \left( \frac{2\sigma_\psi^2}{B^k} + \frac{2\sigma_\alpha^2}{I^2} \right)$$

$$+ 2\eta^2(t-t_c) \sum_{\rho=t_c}^{t-1} \left\| \nabla_{\boldsymbol{\alpha}_i^k} F^k(\boldsymbol{\alpha}_\rho^k, \Psi^k) \right\|^2 + \frac{2\eta^2(t-t_c)}{K_*} \sum_{k \in U_\rho} \sum_{\rho=t_c}^{t-1} \left\| \nabla_{\boldsymbol{\alpha}_i^k} F^k(\boldsymbol{\alpha}_\rho^k, \Psi^k) \right\|^2$$

$$= 2\eta^2 \sum_{\rho=t_c}^{t-1} \left( \frac{2\sigma_\psi^2}{B^k} + \frac{2\sigma_\alpha^2}{I^2} \right) \left( 1 + \frac{1}{K_*} \right)$$

$$+ 2\eta^2(t-t_c) \left( \sum_{\rho=t_c}^{t-1} \left\| \nabla_{\boldsymbol{\alpha}_i^k} F^k(\boldsymbol{\alpha}_\rho^k, \Psi^k) \right\|^2 + \frac{1}{K_*} \sum_{k \in U_\rho} \sum_{\rho=t_c}^{t-1} \left\| \nabla_{\boldsymbol{\alpha}_i^k} F^k(\boldsymbol{\alpha}_\rho^k, \Psi^k) \right\|^2 \right)$$

where (1) use inequality $\|a-b\|^2 \le 2\|a\|^2 + 2\|b\|^2$; (2) use $\mathbb{E}\|x - \mathbb{E}[x]\|^2 = \mathbb{E}\|x\|^2 - \|\mathbb{E}[x]\|^2$; (3) use the unbiasedness of stochastic gradient and variance-reduced policy gradient in Assumption 3.1 and Lemma A.1; (4) use the independence

of mini-batch and prompt sampling in each client; (5) use inequality $\|\sum_{z=1}^Z a_z\|^2 \le Z \sum_{z=1}^Z \|a_z\|^2$; (6) use the bounded variance of stochastic gradient and variance-reduced policy gradient in Assumption 3.1 and Lemma A.1.

Then, we sum both sides with respect to $k \in U_\rho$:

$$
\sum_{k \in U_\rho} \mathbb{E} \left\| \boldsymbol{\alpha}_{i,(t)} - \boldsymbol{\alpha}_{i,(t)}^k \right\|^2
$$

$$
\le 2\eta^2 \sum_{\rho=t_c}^{t-1} \left( \frac{2\sigma_\psi^2}{B^k} + \frac{2\sigma_\alpha^2}{I^2} \right) (K_* + 1) + 2\eta^2 (t - t_c) \left( 1 + \frac{1}{K_*} \right) \sum_{k \in U_\rho} \sum_{\rho=t_c}^{t-1} \left\| \nabla_{\boldsymbol{\alpha}_i^k} F^k(\boldsymbol{\alpha}_\rho^k, \Psi^k) \right\|^2
$$

We take the expectation of both sides about $\mathbb{E}_{U_\rho}$:

$$
\mathbb{E}_{U_\rho} \left[ \sum_{k \in U_\rho} \mathbb{E} \left\| \boldsymbol{\alpha}_{i,(t)} - \boldsymbol{\alpha}_{i,(t)}^k \right\|^2 \right]
$$

$$
= K_* \sum_{k=1}^K q^{[k]} \cdot \mathbb{E} \left\| \boldsymbol{\alpha}_{i,(t)} - \boldsymbol{\alpha}_{i,(t)}^k \right\|^2
$$

$$
\overset{(1)}{\le} 2\eta^2 \sum_{\rho=t_c}^{t-1} \left( \frac{2\sigma_\psi^2}{B^k} + \frac{2\sigma_\alpha^2}{I^2} \right) (K_* + 1) + 2\eta^2 (t - t_c)(K_* + 1) \sum_{k=1}^K \sum_{\rho=t_c}^{t-1} q^{[k]} \cdot \left\| \nabla_{\boldsymbol{\alpha}_i^k} F^k(\boldsymbol{\alpha}_\rho^k, \Psi^k) \right\|^2
$$

where (1) is because we sample client set $U_\rho$ uniformly at random where client $k$ is sampled with probability $q^{[k]}$ for $1 \le k \le K$ with replacement, and we define $U_\rho = \{k_1, ..., k_b, ..., k_{K_*}\}$, then

$$
\mathbb{E}_{U_\rho} \left[ \sum_{k \in U_\rho} \left\| \nabla_{\boldsymbol{\alpha}_i^k} F^k(\boldsymbol{\alpha}_\rho^k, \Psi^k) \right\|^2 \right]
$$

$$
= \sum_{b=1}^{K_*} \mathbb{E}_{k_b} \left[ \left\| \nabla_{\boldsymbol{\alpha}_i^k} F^k(\boldsymbol{\alpha}_\rho^{k_b}, \Psi^{k_b}) \right\|^2 \right]
$$

$$
= \sum_{b=1}^{K_*} \sum_{k=1}^K q^{[k]} \cdot \left\| \nabla_{\boldsymbol{\alpha}_i^k} F^k(\boldsymbol{\alpha}_\rho^k, \Psi^k) \right\|^2
$$

$$
= K_* \sum_{k=1}^K q^{[k]} \cdot \left\| \nabla_{\boldsymbol{\alpha}_i^k} F^k(\boldsymbol{\alpha}_\rho^k, \Psi^k) \right\|^2
$$

Thus:

$$
\sum_{k=1}^K q^{[k]} \cdot \mathbb{E} \left\| \boldsymbol{\alpha}_{i,(t)} - \boldsymbol{\alpha}_{i,(t)}^k \right\|^2
$$

$$
\le 2\eta^2 \sum_{\rho=t_c}^{t-1} \left( \frac{2\sigma_\psi^2}{B^k} + \frac{2\sigma_\alpha^2}{I^2} \right) \left( 1 + \frac{1}{K_*} \right) + 2\eta^2 (t - t_c) \left( 1 + \frac{1}{K_*} \right) \sum_{k=1}^K \sum_{\rho=t_c}^{t-1} q^{[k]} \cdot \left\| \nabla_{\boldsymbol{\alpha}_i^k} F^k(\boldsymbol{\alpha}_\rho^k, \Psi^k) \right\|^2
$$

We take $v = \rho - t_c$, $\gamma = t - 1 - t_c$, and add iteration on both sides:

$$
\sum_{t=0}^{T-1} \sum_{k=1}^K q^{[k]} \cdot \mathbb{E} \left\| \boldsymbol{\alpha}_{i,(t)} - \boldsymbol{\alpha}_{i,(t)}^k \right\|^2
$$

$$
= \sum_{s=0}^{\lfloor \frac{T-1}{E} \rfloor} \sum_{\gamma=0}^{E-1} \sum_{k=1}^K q^{[k]} \cdot \mathbb{E} \left\| \boldsymbol{\alpha}_{i,(sE+\gamma)} - \boldsymbol{\alpha}_{i,(sE+\gamma)}^k \right\|^2
$$

$$
\le \sum_{s=0}^{\lfloor \frac{T-1}{E} \rfloor} \sum_{\gamma=0}^{E-1} 2\eta^2 \sum_{v=0}^{\gamma} \left( \frac{2\sigma_\psi^2}{B^k} + \frac{2\sigma_\alpha^2}{I^2} \right) \left( 1 + \frac{1}{K_*} \right)
$$

$$+ \sum_{s=0}^{\lfloor \frac{T-1}{E} \rfloor} \sum_{\gamma=0}^{E-1} 2\eta^2(\gamma+1)\left(1+\frac{1}{K_*}\right) \sum_{k=1}^{K} \sum_{v=0}^{\gamma} q^{[k]} \cdot \left\| \nabla_{\boldsymbol{\alpha}_i^k} F^k(\boldsymbol{\alpha}_{sE+v}^k, \Psi^k) \right\|^2$$

$$\overset{(1)}{\le} (E+1)T\eta^2\left(\frac{2\sigma_\psi^2}{B^k} + \frac{2\sigma_\alpha^2}{I^2}\right)\left(1+\frac{1}{K_*}\right)$$

$$+ \sum_{s=0}^{\lfloor \frac{T-1}{E} \rfloor} 2\eta^2 E^2 \left(1+\frac{1}{K_*}\right) \sum_{k=1}^{K} \sum_{v=0}^{E-1} q^{[k]} \cdot \left\| \nabla_{\boldsymbol{\alpha}_i^k} F^k(\boldsymbol{\alpha}_{sE+v}^k, \Psi^k) \right\|^2$$

$$\le (E+1)T\eta^2\left(\frac{2\sigma_\psi^2}{B^k} + \frac{2\sigma_\alpha^2}{I^2}\right)\left(1+\frac{1}{K_*}\right) + 2\eta^2 E^2 \left(1+\frac{1}{K_*}\right) \sum_{k=1}^{K} \sum_{t=0}^{T-1} q^{[k]} \cdot \left\| \nabla_{\boldsymbol{\alpha}_i^k} F^k(\boldsymbol{\alpha}_t^k, \Psi^k) \right\|^2$$

where (1) use $0 \le \gamma \le E-1$ and $1 + ... + E = \frac{E(E+1)}{2}$.
Finally multiply both sides simultaneously by $\frac{1}{T}$:

$$\frac{1}{T} \sum_{t=0}^{T-1} \sum_{k=1}^{K} q^{[k]} \cdot \mathbb{E} \left\| \boldsymbol{\alpha}_{i,(t)} - \boldsymbol{\alpha}_{i,(t)}^k \right\|^2$$

$$\le (E+1)\eta^2\left(\frac{2\sigma_\psi^2}{B^k} + \frac{2\sigma_\alpha^2}{I^2}\right)\left(1+\frac{1}{K_*}\right) + \frac{2\eta^2 E^2}{T}\left(1+\frac{1}{K_*}\right) \sum_{k=1}^{K} \sum_{t=0}^{T-1} q^{[k]} \cdot \left\| \nabla_{\boldsymbol{\alpha}_i^k} F^k(\boldsymbol{\alpha}_t^k, \Psi^k) \right\|^2$$

$$\overset{(1)}{\le} (E+1)\eta^2\left(\frac{2\sigma_\psi^2}{B^k} + \frac{2\sigma_\alpha^2}{I^2}\right)\left(1+\frac{1}{K_*}\right) + \frac{2\eta^2 E^2 \lambda}{T}\left(1+\frac{1}{K_*}\right) \sum_{t=0}^{T-1} \left\| \sum_{k=1}^{K} q^{[k]} \cdot \nabla_{\boldsymbol{\alpha}_i^k} F^k(\boldsymbol{\alpha}_t^k, \Psi^k) \right\|^2$$

$$= (E+1)\eta^2\left(\frac{2\sigma_\psi^2}{B^k} + \frac{2\sigma_\alpha^2}{I^2}\right)\left(1+\frac{1}{K_*}\right) + \frac{2\eta^2 E^2 \lambda}{T}\left(1+\frac{1}{K_*}\right) \sum_{t=0}^{T-1} \left\| \nabla_{\boldsymbol{\alpha}_i} F(\boldsymbol{\alpha}_t^k, \Psi^k) \right\|^2$$

where (1) use Assumption 3.3. $\qquad\square$

**Theorem 3.4.** Suppose Assumption 3.1, 3.2 and 3.3 hold, for $t = 0, 1, ..., T-1$, $B = \min\{B^1, ..., B^K\}$ where $B^k$ is the local mini-batch size. $\alpha_{i,j} \ge \nu > 0$ for $i = 1, ..., n$ and $j = 1, ..., N$, $I$ is the sampling times for prompt. $\sigma_\alpha^2 = \frac{8G^2 N}{\tau^2 \nu^2}$ is the variance of the variance-reduced policy gradient, $\sigma_\psi^2$ is the variance of the stochastic gradient, $\mathbb{E}_{\Phi^k \sim \mathrm{GS}(\boldsymbol{\alpha}^k)}\left[\mathcal{L}(\Phi^k, \Psi^k)\right]$ is L-smooth for $\boldsymbol{\alpha}^k$ and $L = \frac{nGN(\tau+1)}{\tau^2 \nu^2}$, and $\eta$ satisfies the following inequality:

$$0 < \eta < \frac{1}{L\lambda} \tag{36}$$

Then, the Fed-BDPL's full gradient $\nabla_{\boldsymbol{\alpha}} F(\boldsymbol{\alpha}_t, \Psi^k)$ satisfies the following inequality:

$$\frac{1}{T} \sum_{t=0}^{T-1} \left\| \nabla_{\boldsymbol{\alpha}} F(\boldsymbol{\alpha}_t, \Psi^k) \right\|^2$$

$$\le \frac{4G}{\eta T} + \frac{2(E+1)n\sigma_\psi^2(1+\frac{1}{K_*}) + 2n\sigma_\psi^2}{B} + \frac{2(E+1)n\sigma_\alpha^2(1+\frac{1}{K_*}) + 2n\sigma_\alpha^2}{I^2}$$

*Proof.* According to Lemma A.2:

$$\mathbb{E}_{\Phi_{t+1} \sim \mathrm{GS}(\boldsymbol{\alpha}_{t+1})}\left[\mathcal{L}(\Phi_{t+1}, \Psi)\right] - \mathbb{E}_{\Phi_t \sim \mathrm{GS}(\boldsymbol{\alpha}_t)}\left[\mathcal{L}(\Phi_t, \Psi)\right]$$

$$\le \left\langle \nabla_{\boldsymbol{\alpha}} \mathbb{E}_{\Phi_t \sim \mathrm{GS}(\boldsymbol{\alpha}_t)}\left[\mathcal{L}(\Phi_t, \Psi)\right], \boldsymbol{\alpha}_{t+1} - \boldsymbol{\alpha}_t \right\rangle + \frac{L}{2}\left\| \boldsymbol{\alpha}_{t+1} - \boldsymbol{\alpha}_t \right\|^2$$

$$\le \sum_{i=1}^{n} \left[ \left\langle \nabla_{\boldsymbol{\alpha}_i} F(\boldsymbol{\alpha}_t, \Psi^k), -\eta \cdot \hat{\nabla}_{\boldsymbol{\alpha}_i^k} f^*(\boldsymbol{\alpha}_t^k, \mathcal{B}_t^k) \right\rangle + \frac{L\eta^2}{2}\left\| \hat{\nabla}_{\boldsymbol{\alpha}_i^k} f^*(\boldsymbol{\alpha}_t^k, \mathcal{B}_t^k) \right\|^2 \right]$$

We take the expectations about $\left\{\Phi^{k,r}\right\}_{r=1}^{I}$, $\mathcal{B}_t^k$ and $U_t$ on both sides respectively:

$$\mathbb{E}_{\{\Phi^{k,r}\}_{r=1}^{I}}\mathbb{E}_{\mathcal{B}_t^k}\mathbb{E}_{U_t}\left\{\mathbb{E}_{\Phi_{t+1}\sim\text{GS}(\boldsymbol{\alpha}_{t+1})}\left[\mathcal{L}(\Phi_{t+1},\Psi)\right]-\mathbb{E}_{\Phi_t\sim\text{GS}(\boldsymbol{\alpha}_t)}\left[\mathcal{L}(\Phi_t,\Psi)\right]\right\}$$

$$\leq\sum_{i=1}^{n}\left\{\underbrace{\mathbb{E}_{\{\Phi^{k,r}\}_{r=1}^{I}}\mathbb{E}_{\mathcal{B}_t^k}\mathbb{E}_{U_t}\left[\left\langle\nabla_{\boldsymbol{\alpha}_i}F(\boldsymbol{\alpha}_t,\Psi^k),-\eta\cdot\hat{\nabla}_{\boldsymbol{\alpha}_i^k}f^*(\boldsymbol{\alpha}_t^k,\mathcal{B}_t^k)\right\rangle\right]}_{a)}\right\}$$

$$+\sum_{i=1}^{n}\left\{\underbrace{\frac{L\eta^2}{2}\mathbb{E}_{\{\Phi^{k,r}\}_{r=1}^{I}}\mathbb{E}_{\mathcal{B}_t^k}\mathbb{E}_{U_t}\left[\left\|\hat{\nabla}_{\boldsymbol{\alpha}_i^k}f^*(\boldsymbol{\alpha}_t^k,\mathcal{B}_t^k)\right\|^2\right]}_{b)}\right\}$$

For a):

$$\mathbb{E}_{\{\Phi^{k,r}\}_{r=1}^{I}}\mathbb{E}_{\mathcal{B}_t^k}\mathbb{E}_{U_t}\left[\left\langle\nabla_{\boldsymbol{\alpha}_i}F(\boldsymbol{\alpha}_t,\Psi^k),-\eta\cdot\hat{\nabla}_{\boldsymbol{\alpha}_i^k}f^*(\boldsymbol{\alpha}_t^k,\mathcal{B}_t^k)\right\rangle\right]$$

$$=\left\langle\nabla_{\boldsymbol{\alpha}_i}F(\boldsymbol{\alpha}_t,\Psi^k),-\eta\cdot\mathbb{E}_{\{\Phi^{k,r}\}_{r=1}^{I}}\mathbb{E}_{\mathcal{B}_t^k}\mathbb{E}_{U_t}\left[\hat{\nabla}_{\boldsymbol{\alpha}_i^k}f^*(\boldsymbol{\alpha}_t^k,\mathcal{B}_t^k)\right]\right\rangle$$

$$=\left\langle\nabla_{\boldsymbol{\alpha}_i}F(\boldsymbol{\alpha}_t,\Psi^k),-\eta\cdot\mathbb{E}_{U_t}\mathbb{E}_{\{\Phi^{k,r}\}_{r=1}^{I}}\mathbb{E}_{\mathcal{B}_t^k}\left[\frac{1}{K_*}\sum_{k\in U_t}\hat{\nabla}_{\boldsymbol{\alpha}_i}f^k(\boldsymbol{\alpha}^k,\Psi^k)\right]\right\rangle$$

$$=\left\langle\nabla_{\boldsymbol{\alpha}_i}F(\boldsymbol{\alpha}_t,\Psi^k),-\eta\cdot\mathbb{E}_{U_t}\left[\frac{1}{K_*}\sum_{k\in U_t}\nabla_{\boldsymbol{\alpha}_i^k}F^k(\boldsymbol{\alpha}^k,\Psi^k)\right]\right\rangle$$

$$=\left\langle\nabla_{\boldsymbol{\alpha}_i}F(\boldsymbol{\alpha}_t,\Psi^k),-\eta\cdot\frac{1}{K_*}\left[K_*\sum_{k=1}^{K}q^{[k]}\cdot\nabla_{\boldsymbol{\alpha}_i^k}F^k(\boldsymbol{\alpha}^k,\Psi^k)\right]\right\rangle$$

$$\overset{(1)}{=}\frac{\eta}{2}\left[-\left\|\nabla_{\boldsymbol{\alpha}_i}F(\boldsymbol{\alpha}_t,\Psi^k)\right\|^2-\left\|\nabla_{\boldsymbol{\alpha}_i^k}F(\boldsymbol{\alpha}_t^k,\Psi^k)\right\|^2+\left\|\nabla_{\boldsymbol{\alpha}_i}F(\boldsymbol{\alpha}_t,\Psi^k)-\sum_{k=1}^{K}q^{[k]}\cdot\nabla_{\boldsymbol{\alpha}_i^k}F^k(\boldsymbol{\alpha}^k,\Psi^k)\right\|^2\right]$$

$$=\frac{\eta}{2}\left[-\left\|\nabla_{\boldsymbol{\alpha}_i}F(\boldsymbol{\alpha}_t,\Psi^k)\right\|^2-\left\|\nabla_{\boldsymbol{\alpha}_i^k}F(\boldsymbol{\alpha}_t^k,\Psi^k)\right\|^2+\left\|\sum_{k=1}^{K}q^{[k]}\cdot\left(\nabla_{\boldsymbol{\alpha}_i}F^k(\boldsymbol{\alpha}_t,\Psi^k)-\nabla_{\boldsymbol{\alpha}_i^k}F^k(\boldsymbol{\alpha}^k,\Psi^k)\right)\right\|^2\right]$$

$$\overset{(2)}{\leq}\frac{\eta}{2}\left[-\left\|\nabla_{\boldsymbol{\alpha}_i}F(\boldsymbol{\alpha}_t,\Psi^k)\right\|^2-\left\|\nabla_{\boldsymbol{\alpha}_i^k}F(\boldsymbol{\alpha}_t^k,\Psi^k)\right\|^2+\sum_{k=1}^{K}q^{[k]}\cdot\left\|\nabla_{\boldsymbol{\alpha}_i}F^k(\boldsymbol{\alpha}_t,\Psi^k)-\nabla_{\boldsymbol{\alpha}_i^k}F^k(\boldsymbol{\alpha}^k,\Psi^k)\right\|^2\right]$$

$$=\frac{\eta}{2}\left[-\left\|\nabla_{\boldsymbol{\alpha}_i}F(\boldsymbol{\alpha}_t,\Psi^k)\right\|^2-\left\|\nabla_{\boldsymbol{\alpha}_i^k}F(\boldsymbol{\alpha}_t^k,\Psi^k)\right\|^2\right]$$

$$+\frac{\eta}{2}\cdot\sum_{k=1}^{K}q^{[k]}\cdot\left\|\frac{1}{M^k}\sum_{\psi_m^k\in\Psi^k}\nabla_{\boldsymbol{\alpha}_i}\mathbb{E}_{\Phi_t\sim\text{GS}(\boldsymbol{\alpha}_t)}[\mathcal{L}(\Phi_t,\psi_m^k)]-\frac{1}{M^k}\sum_{\psi_m^k\in\Psi^k}\nabla_{\boldsymbol{\alpha}_i^k}\mathbb{E}_{\Phi_t^k\sim\text{GS}(\boldsymbol{\alpha}_t^k)}[\mathcal{L}(\Phi_t^k,\psi_m^k)]\right\|^2$$

$$\overset{(3)}{\leq}-\frac{\eta}{2}\left[\left\|\nabla_{\boldsymbol{\alpha}_i}F(\boldsymbol{\alpha}_t,\Psi^k)\right\|^2+\left\|\nabla_{\boldsymbol{\alpha}_i^k}F(\boldsymbol{\alpha}_t^k,\Psi^k)\right\|^2\right]+\frac{\eta}{2}\left[\sum_{k=1}^{K}q^{[k]}L^2\cdot\left\|\boldsymbol{\alpha}_{i,(t)}-\boldsymbol{\alpha}_{i,(t)}^k\right\|^2\right]$$

where (1) use inequality $2\langle a,b\rangle=\|a\|^2+\|b\|^2-\|a-b\|^2$; (2) use the convexity of $\ell_2$ norm; (3) use L-smooth in Lemma A.2.

For b):

$$\frac{L\eta^2}{2}\mathbb{E}_{\{\Phi^{k,r}\}_{r=1}^{I}}\mathbb{E}_{\mathcal{B}_t^k}\mathbb{E}_{U_t}\left[\left\|\hat{\nabla}_{\boldsymbol{\alpha}_i^k}f^*(\boldsymbol{\alpha}_t^k,\mathcal{B}_t^k)\right\|^2\right]$$

$$=\mathbb{E}_{U_t}\left\{\frac{L\eta^2}{2}\mathbb{E}_{\{\Phi^{k,r}\}_{r=1}^{I}}\mathbb{E}_{\mathcal{B}_t^k}\left[\left\|\hat{\nabla}_{\boldsymbol{\alpha}_i^k}f^*(\boldsymbol{\alpha}_t^k,\mathcal{B}_t^k)\right\|^2\right]\right\}$$

$$\overset{(1)}{=} \mathbb{E}_{U_t} \left\{ \frac{L\eta^2}{2} \mathbb{E}_{\{\Phi^{k,r}\}_{r=1}^I} \mathbb{E}_{\mathcal{B}_t^k} \left[ \left\| \hat{\nabla}_{\boldsymbol{\alpha}_i^k} f^*(\boldsymbol{\alpha}_t^k, \mathcal{B}_t^k) - \mathbb{E}_{\{\Phi^{k,r}\}_{r=1}^I} \mathbb{E}_{\mathcal{B}_t^k} \left[ \hat{\nabla}_{\boldsymbol{\alpha}_i^k} f^*(\boldsymbol{\alpha}_t^k, \mathcal{B}_t^k) \right] \right\|^2 \right] \right\}$$

$$+ \mathbb{E}_{U_t} \left\{ \frac{L\eta^2}{2} \left\| \mathbb{E}_{\{\Phi^{k,r}\}_{r=1}^I} \mathbb{E}_{\mathcal{B}_t^k} \left[ \hat{\nabla}_{\boldsymbol{\alpha}_i^k} f^*(\boldsymbol{\alpha}_t^k, \mathcal{B}_t^k) \right] \right\|^2 \right\}$$

$$\overset{(2)}{=} \mathbb{E}_{U_t} \left\{ \frac{L\eta^2}{2} \mathbb{E}_{\{\Phi^{k,r}\}_{r=1}^I} \mathbb{E}_{\mathcal{B}_t^k} \left[ \left\| \hat{\nabla}_{\boldsymbol{\alpha}_i^k} f^*(\boldsymbol{\alpha}_t^k, \mathcal{B}_t^k) - \nabla_{\boldsymbol{\alpha}_i^k} F^*(\boldsymbol{\alpha}_t^k, \Psi^k) \right\|^2 \right] \right\}$$

$$+ \mathbb{E}_{U_t} \left[ \frac{L\eta^2}{2} \left\| \nabla_{\boldsymbol{\alpha}_i^k} F^*(\boldsymbol{\alpha}_t^k, \Psi^k) \right\|^2 \right]$$

$$\overset{(3)}{\leq} \frac{L\eta^2}{2} \mathbb{E}_{U_t} \left\| \nabla_{\boldsymbol{\alpha}_i^k} F^*(\boldsymbol{\alpha}_t^k, \Psi^k) \right\|^2$$

$$+ \mathbb{E}_{U_t} \left\{ L\eta^2 \mathbb{E}_{\{\Phi^{k,r}\}_{r=1}^I} \mathbb{E}_{\mathcal{B}_t^k} \left[ \left\| \hat{\nabla}_{\boldsymbol{\alpha}_i^k} f^*(\boldsymbol{\alpha}_t^k, \mathcal{B}_t^k) - \mathbb{E}_{\{\Phi^{k,r}\}_{r=1}^I} \hat{\nabla}_{\boldsymbol{\alpha}_i^k} f^*(\boldsymbol{\alpha}_t^k, \mathcal{B}_t^k) \right\|^2 \right] \right\}$$

$$+ \mathbb{E}_{U_t} \left\{ L\eta^2 \mathbb{E}_{\{\Phi^{k,r}\}_{r=1}^I} \mathbb{E}_{\mathcal{B}_t^k} \left[ \left\| \mathbb{E}_{\{\Phi^{k,r}\}_{r=1}^I} \hat{\nabla}_{\boldsymbol{\alpha}_i^k} f^*(\boldsymbol{\alpha}_t^k, \mathcal{B}_t^k) - \mathbb{E}_{\{\Phi^{k,r}\}_{r=1}^I} \mathbb{E}_{\mathcal{B}_t^k} \hat{\nabla}_{\boldsymbol{\alpha}_i^k} f^*(\boldsymbol{\alpha}_t^k, \mathcal{B}_t^k) \right\|^2 \right] \right\}$$

$$\overset{(4)}{\leq} \frac{L\eta^2}{2} \mathbb{E}_{U_t} \left\| \nabla_{\boldsymbol{\alpha}_i^k} F^*(\boldsymbol{\alpha}_t^k, \Psi^k) \right\|^2 + L\eta^2 \cdot \mathbb{E}_{U_t} \left[ \frac{\sigma_\alpha^2}{I^2} \right] + L\eta^2 \cdot \mathbb{E}_{U_t} \left[ \frac{\sigma_\psi^2}{B} \right]$$

$$= \frac{L\eta^2}{2} \sum_{k=1}^K q^{[k]} \cdot \left\| \nabla_{\boldsymbol{\alpha}_i^k} F^k(\boldsymbol{\alpha}_t^k, \Psi^k) \right\|^2 + \frac{L\eta^2 \sigma_\alpha^2}{I^2} + \frac{L\eta^2 \sigma_\psi^2}{B}$$

$$\overset{(5)}{\leq} \frac{L\eta^2 \lambda}{2} \left\| \sum_{k=1}^K q^{[k]} \cdot \nabla_{\boldsymbol{\alpha}_i^k} F^k(\boldsymbol{\alpha}_t^k, \Psi^k) \right\|^2 + \frac{L\eta^2 \sigma_\alpha^2}{I^2} + \frac{L\eta^2 \sigma_\psi^2}{B} \tag{37}$$

where (1) use $\mathbb{E} \|x - \mathbb{E}[x]\|^2 = \mathbb{E} \|x\|^2 - \|\mathbb{E}[x]\|^2$; (2) use the unbiasedness of stochastic gradient and variance-reduced policy gradient in Assumption 3.1 and Lemma A.1; (3) use $\|a + b\|^2 \leq 2 \|a\|^2 + 2 \|b\|^2$; (4) use the bounded variance of stochastic gradient and variance-reduced policy gradient in Assumption 3.1 and Lemma A.1; (5) use Assumption 3.3. Combining a) and b):

$$\frac{1}{T} \sum_{t=0}^{T-1} \left\{ \mathbb{E}_{\Phi_{t+1} \sim \mathrm{GS}(\boldsymbol{\alpha}_{t+1})} \left[ \mathcal{L}(\Phi_{t+1}, \Psi) \right] - \mathbb{E}_{\Phi_t \sim \mathrm{GS}(\boldsymbol{\alpha}_t)} \left[ \mathcal{L}(\Phi_t, \Psi) \right] \right\}$$

$$\leq \frac{1}{T} \sum_{t=0}^{T-1} \sum_{i=1}^n \left\{ -\frac{\eta}{2} \left[ \left\| \nabla_{\boldsymbol{\alpha}_i} F(\boldsymbol{\alpha}_t, \Psi^k) \right\|^2 + \left\| \nabla_{\boldsymbol{\alpha}_i^k} F(\boldsymbol{\alpha}_t^k, \Psi^k) \right\|^2 \right] + \frac{\eta}{2} \left[ \sum_{k=1}^K q^{[k]} L^2 \cdot \left\| \boldsymbol{\alpha}_{i,(t)} - \boldsymbol{\alpha}_{i,(t)}^k \right\|^2 \right] \right\}$$

$$+ \frac{1}{T} \sum_{t=0}^{T-1} \sum_{i=1}^n \left\{ \frac{L\eta^2 \lambda}{2} \left\| \sum_{k=1}^K q^{[k]} \cdot \nabla_{\boldsymbol{\alpha}_i^k} F^k(\boldsymbol{\alpha}_t^k, \Psi^k) \right\|^2 + \frac{L\eta^2 \sigma_\alpha^2}{I^2} + \frac{L\eta^2 \sigma_\psi^2}{B} \right\}$$

$$\overset{(1)}{\leq} -\frac{\eta}{2} \frac{1}{T} \sum_{t=0}^{T-1} \left\| \nabla_{\boldsymbol{\alpha}} F(\boldsymbol{\alpha}_t, \Psi^k) \right\|^2 + \left[ -\frac{\eta}{2} + \frac{\lambda L\eta^2}{2} + \eta^3 L^2 E (1 + \frac{1}{K_*}) \right] \frac{1}{T} \sum_{t=0}^{T-1} \left\| \nabla_{\boldsymbol{\alpha}^k} F(\boldsymbol{\alpha}_t^k, \Psi^k) \right\|^2$$

$$+ \frac{(E+1)L^2 \eta^3 n \sigma_\psi^2 (1 + \frac{1}{K_*}) + n L \eta^2 \sigma_\psi^2}{B} + \frac{(E+1)L^2 \eta^3 n \sigma_\alpha^2 (1 + \frac{1}{K_*}) + n L \eta^2 \sigma_\alpha^2}{I^2}$$

where (1) use Lemma A.7.
Then, we can get:

$$\frac{1}{T} \sum_{t=0}^{T-1} \left\| \nabla_{\boldsymbol{\alpha}} F(\boldsymbol{\alpha}_t, \Psi^k) \right\|^2$$

$$\leq \frac{\mathbb{E}_{\Phi_0 \sim \mathrm{GS}(\boldsymbol{\alpha}_0)} \left[ \mathcal{L}(\Phi_0, \Psi) \right] - \mathbb{E}_{\Phi_T \sim \mathrm{GS}(\boldsymbol{\alpha}_t)} \left[ \mathcal{L}(\Phi_T, \Psi) \right]}{\eta T}$$

$$+ \left[ -1 + \lambda L \eta + 2\eta^2 L^2 E (1 + \frac{1}{K_*}) \right] \frac{1}{T} \sum_{t=0}^{T-1} \left\| \nabla_{\boldsymbol{\alpha}^k} F(\boldsymbol{\alpha}_t^k, \Psi^k) \right\|^2$$

$$+ \frac{2(E+1)L^2\eta^2 n\sigma_\psi^2(1+\frac{1}{K_*})+2nL\eta\sigma_\psi^2}{B} + \frac{2(E+1)L^2\eta^2 n\sigma_\alpha^2(1+\frac{1}{K_*})+2nL\eta\sigma_\alpha^2}{I^2}$$

Based on Assumption 3.2, $\mathbb{E}_{\Phi_0\sim\text{GS}(\boldsymbol{\alpha}_0)}\left[\mathcal{L}(\Phi_0,\Psi)\right] - \inf_t \mathbb{E}_{\Phi_t\sim\text{GS}(\boldsymbol{\alpha}_t)}\left[\mathcal{L}(\Phi_t,\Psi)\right] \le 2G$ and:

$$-1 + \lambda L\eta + 2\eta^2 L^2 E(1+\frac{1}{K_*}) \le 0$$

$$0 < \eta \le \eta^* = \frac{-\lambda L + \sqrt{\lambda^2 L^2 + 8L^2 E\left(1+\frac{1}{K_*}\right)}}{4L^2 E\left(1+\frac{1}{K_*}\right)}$$

According to Taylor's inequality: $\sqrt{1+x} \ge 1 + \frac{x}{2}$ for $x \ge 0$:

$$\begin{aligned}
&\sqrt{\lambda^2 L^2 + 8L^2 E\left(1+\frac{1}{K_*}\right)} \\
&= \lambda L\sqrt{1 + \frac{8L^2 E\left(1+\frac{1}{K_*}\right)}{\lambda^2 L^2}} \\
&\ge \lambda L\left(1 + \frac{4E\left(1+\frac{1}{K_*}\right)}{\lambda^2}\right)
\end{aligned} \tag{38}$$

Then, we can get:

$$\begin{aligned}
&\frac{-\lambda L + \sqrt{\lambda^2 L^2 + 8L^2 E\left(1+\frac{1}{K_*}\right)}}{4L^2 E\left(1+\frac{1}{K_*}\right)} \\
&\ge \frac{-\lambda L + \lambda L\left(1 + \frac{4E\left(1+\frac{1}{K_*}\right)}{\lambda^2}\right)}{4L^2 E\left(1+\frac{1}{K_*}\right)} \\
&= \frac{1}{L\lambda}
\end{aligned} \tag{39}$$

Finally:

$$\begin{aligned}
&\frac{1}{T}\sum_{t=0}^{T-1}\left\|\nabla_{\boldsymbol{\alpha}} F(\boldsymbol{\alpha}_t,\Psi^k)\right\|^2 \\
&\le \frac{4G}{\eta T} + \frac{2(E+1)L^2\eta^2 n\sigma_\psi^2(1+\frac{1}{K_*})+2nL\eta\sigma_\psi^2}{B} + \frac{2(E+1)L^2\eta^2 n\sigma_\alpha^2(1+\frac{1}{K_*})+2nL\eta\sigma_\alpha^2}{I^2} \\
&\le \frac{4G}{\eta T} + \frac{2(E+1)n\sigma_\psi^2(1+\frac{1}{K_*})+2n\sigma_\psi^2}{B} + \frac{2(E+1)n\sigma_\alpha^2(1+\frac{1}{K_*})+2n\sigma_\alpha^2}{I^2}
\end{aligned} \tag{40}$$

$\square$

**Corollary 3.5.** **Convergence Rate of Fed-BDPL**:
Let $\eta = \min\left\{\frac{1}{L\lambda}, \frac{1}{\sqrt{T}}, \frac{1}{L}\right\}$, $B = \sqrt{T}$ and $I = T^{\frac{1}{4}}$, the following holds:

$$\frac{1}{T}\sum_{t=0}^{T-1}\left\|\nabla_{\boldsymbol{\alpha}} F(\boldsymbol{\alpha}_t,\Psi)\right\|^2 = \mathcal{O}(\frac{1}{\sqrt{T}}) \tag{41}$$

*Proof.* Let $\eta = \min\left\{\frac{1}{L\lambda}, \frac{1}{\sqrt{T}}, \frac{1}{L}\right\}$, $B = \sqrt{T}$ and $I = T^{\frac{1}{4}}$, plunging in Eq. 40, then we have:

$$\frac{1}{T}\sum_{t=0}^{T-1}\left\|\nabla_{\boldsymbol{\alpha}}F(\boldsymbol{\alpha}_t, \Psi^k)\right\|^2$$

$$\leq \frac{1}{\sqrt{T}}(4G) + \frac{1}{\sqrt{T}}\left(2n\sigma_\psi^2 + 2n\sigma_\alpha^2\right) + \frac{1}{\sqrt{T}}\left[2(E+1)n\sigma_\psi^2(1+\frac{1}{K_*}) + 2(E+1)n\sigma_\alpha^2(1+\frac{1}{K_*})\right] \quad (42)$$

Therefore, $\frac{1}{T}\sum_{t=0}^{T-1}\left\|\nabla_{\boldsymbol{\alpha}}F(\boldsymbol{\alpha}_t, \Psi)\right\|^2 = \mathcal{O}(\frac{1}{\sqrt{T}})$. $\qquad\square$

**Corollary 3.6**. **The Impact of** $K_*$ **(FedOne)**: Let $Q_\epsilon$ denote the minimum number of queries submitted to the cloud-based LLM service to achieve an $\epsilon$-solution, formulated as a function of $K_*$, i.e., $Q_\epsilon(K_*)$. We derive its explicit form in Eq. 45 and show that $Q_\epsilon(K_*)$ is a monotonically increasing function of $K_*$ for $K_* = 1, 2, \ldots, K$. Therefore, to minimize query overhead and achieve optimal efficiency, the optimal choice is $K_* = 1$.

*Proof.* Considering the iterative complexity based on Eq. 42, to obtain an $\epsilon$-solution satisfying

$$\frac{1}{T}\sum_{t=0}^{T-1}\left\|\nabla_{\boldsymbol{\alpha}}F(\boldsymbol{\alpha}_t, \Psi^k)\right\|^2 \leq \frac{1}{\sqrt{T}}\left[(4G) + \left(2n\sigma_\psi^2 + 2n\sigma_\alpha^2\right) + 2(E+1)n\sigma_\psi^2(1+\frac{1}{K_*}) + 2(E+1)n\sigma_\alpha^2(1+\frac{1}{K_*})\right] \leq \epsilon^2 \quad (43)$$

the minimum number of iterations, $T_\epsilon$, to achieve the $\epsilon$-solution can be derived as follows[4]:

$$T_\epsilon = \left[\frac{4G}{\epsilon^2} + \frac{2(E+1)n\sigma_\psi^2(1+\frac{1}{K_*}) + 2n\sigma_\psi^2}{\epsilon^2} + \frac{2(E+1)n\sigma_\alpha^2(1+\frac{1}{K_*}) + 2n\sigma_\alpha^2}{\epsilon^2}\right]^2. \quad (44)$$

Note that the minimum number of LLM queries $Q_\epsilon$ required to achieve an $\epsilon$-solution is given by $Q_\epsilon = c \cdot T_\epsilon K_*$, where $c$ is a constant representing the number of query per iteration per client.[5] The term $T_\epsilon K_*$ can be represented as the following:

$$Q_\epsilon = cT_\epsilon K_* = c\left[\frac{4G}{\epsilon^2} + \frac{2(E+1)n\sigma_\psi^2(1+\frac{1}{K_*}) + 2n\sigma_\psi^2}{\epsilon^2} + \frac{2(E+1)n\sigma_\alpha^2(1+\frac{1}{K_*}) + 2n\sigma_\alpha^2}{\epsilon^2}\right]^2 \cdot K_*$$

$$= c\left[\left(\frac{4G}{\epsilon^2} + \frac{2(E+2)n\sigma_\psi^2}{\epsilon^2} + \frac{2(E+2)n\sigma_\alpha^2}{\epsilon^2}\right)\cdot\sqrt{K_*} + \left(\frac{2(E+1)n\sigma_\psi^2}{\epsilon^2} + \frac{2(E+1)n\sigma_\alpha^2}{\epsilon^2}\right)\cdot\frac{1}{\sqrt{K_*}}\right]^2$$

$$\overset{1)}{=} c\left(c_1\sqrt{K_*} + c_2\frac{1}{\sqrt{K_*}}\right)^2 \quad (45)$$

where 1) substituting $c_1 = \frac{4G}{\epsilon^2} + \frac{2(E+2)n\sigma_\psi^2}{\epsilon^2} + \frac{2(E+2)n\sigma_\alpha^2}{\epsilon^2}$ and $c_2 = \frac{2(E+1)n\sigma_\psi^2}{\epsilon^2} + \frac{2(E+1)n\sigma_\alpha^2}{\epsilon^2}$ for brevity. Observe that Eq. 45 is a function of $K_*$. Analyzing its main form, $c_1\sqrt{K_*} + c_2\frac{1}{\sqrt{K_*}}$, and considering its graph and solving it using calculus, we determine that its minimum occurs at $K_*^{opt} = \frac{c_2}{c_1}$, i.e.:

$$K_*^{opt} = \frac{c2}{c1} = \frac{\frac{2(E+1)n\sigma_\psi^2}{\epsilon^2} + \frac{2(E+1)n\sigma_\alpha^2}{\epsilon^2}}{\frac{4G}{\epsilon^2} + \frac{2(E+2)n\sigma_\psi^2}{\epsilon^2} + \frac{2(E+2)n\sigma_\alpha^2}{\epsilon^2}} \overset{1)}{<} 1$$

where 1) note that $c_1 = c_2 + \frac{4G}{\epsilon^2} + \frac{2n\sigma\psi^2}{\epsilon^2} + \frac{2n\sigma\alpha^2}{\epsilon^2}$, therefore the denominator $(c_1)$ is greater than the numerator $(c_2)$.

Finally, considering $K_* \in \mathbb{N}^+$, we observe that $Q_\epsilon$ increases monotonically within the feasible region of $K_*$. This implies that the optimal $K_*$ for query efficiency is $K_* = 1$. $\qquad\square$

---

[4]$T_\epsilon$ can be regarded as a function of $K_*$, treating all other variables as constants.
[5]$Q_\epsilon = c \cdot T_\epsilon K_*$ can be expressed as: (number of query per iteration per client) $\times$ (number of iteration to achieve $\epsilon$-solution) $\times$ (number of activated clients per iteration)

# B. Experiment Details

## B.1. GLUE Dataset Metrics

Below is the table for the detail of the dataset we use, and their associated tasks, metrics, and data domains.

| Dataset | $|\mathbf{L}|$ | $|\mathbf{Train}|$ | $|\mathbf{Dev}|$ | $|\mathbf{Test}|$ | Type | Metrics | Domain |
|---------|-----|--------|-------|--------|-------|---------|--------|
| MNLI | 3 | 393K | 9.8K | 9.8K | NLI | acc. | fiction, reports |
| QQP | 2 | 364K | 40K | 391K | paraphrase | F1 | Quora |
| SST-2 | 2 | 6.7K | 872 | 1.8K | sentiment | acc. | movie reviews |
| MRPC | 2 | 3.7K | 408 | 1.7K | paraphrase | F1 | news |
| CoLA | 2 | 8.6K | 1K | 1K | acceptability | Matthews corr. | books, articles |
| QNLI | 2 | 105K | 5.5K | 5.5K | NLI | acc. | Wikipedia |
| RTE | 2 | 2.5K | 277 | 3K | NLI | acc. | news, Wikipedia |

Table 5: The statistics and metrics of seven datasets in GLUE benchmark, $|\mathbf{L}|$: number of classes for classification tasks.

# C. Supplement Experiment

## C.1. Impact of Activated Clients $K_*$ on Query Efficiency in Federated Black-Box Prompt Learning

We also evaluated the impact of $K_*$ on query efficiency in Federated Black-box Prompt Tuning, as presented in Table 6. The SST2 dataset was used to explore how varying the number of activated clients per round affects model convergence efficiency in FL environments. We tested the number of activated clients per round within the range of $[1, 3, 10, 30]$. For each configuration, we monitored the number of queries required to achieve target accuracy on the validation dataset. Specifically, we evaluated the prompt at the end of each epoch and stop training once the target accuracy was reached, reporting the number of LLM queries at that point. To ensure reliability and account for variability in the learning process, each experimental setup was replicated 20 times, and outliers in the number of queries were removed. The primary aim of our study was to explore how the number of activated clients affects the speed and efficiency of model convergence in FL, specifically to demonstrate the query efficiency of the FedOne approach.

The results presented in Table 6 demonstrate a clear trend where fewer activated clients are associated with greater query efficiency. This relationship is evidenced by a consistent decrease in the number of cloud-based LLM queries as the number of activated clients is reduced, a pattern observed across both the Federated BDPL and Federated BBT.

Table 6: Query Efficiency in Federated Black-Box Prompt Learning

| AC | Fed-BDPL | | Fed-BBT | |
|----|----------|---------------|---------|---------------|
| | # Epoch | # LLM Queries | # Epoch | # LLM Queries |
| 1 | $25.4_{\pm 19.2}$ | $\mathbf{528.6}_{\pm 382.4}$ | $10.8_{\pm 4.4}$ | $\mathbf{2350.0}_{\pm 870.3}$ |
| 3 | $13.8_{\pm 5.2}$ | $889.4_{\pm 310.9}$ | $5.4_{\pm 2.3}$ | $3825.0_{\pm 1356.2}$ |
| 10 | $11.5_{\pm 6.8}$ | $2430.8_{\pm 1378.8}$ | $4.2_{\pm 1.8}$ | $10444.4_{\pm 3624.3}$ |
| 30 | $3.5_{\pm 2.8}$ | $2672.7_{\pm 1625.4}$ | $1.7_{\pm 0.9}$ | $16000.0_{\pm 5656.9}$ |

## C.2. Impact of Prompt Length $n$ on Federated BDPL

To explore the impact of prompt length on the performance of the FedOne framework, we conducted additional experiments using various prompt lengths across multiple tasks from the GLUE benchmark. We varied the length of the discrete prompts by increasing the number of prompt tokens while keeping other hyperparameters constant. The tested prompt lengths included $n \in \{10, 20, 40, 60\}$.

Figure 3 presents the results of these experiments, where test accuracy is reported as the mean across five independent runs with different random seeds, and error bars represent the standard deviation. The results indicate that the overall performance remains relatively stable across different prompt lengths, with minor fluctuations. Notably, the shortest prompt lengths

($n = 10$) and largest prompt lengths ($n = 60$) explore a larger variance, the medium prompt length explores a more stable result.

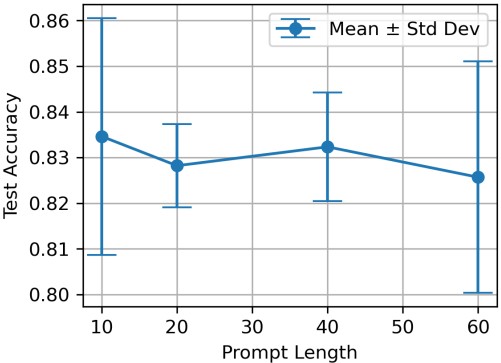

Figure 3: The impact of prompt length

## C.3. Experiments on Heterogeneous Client

We further included an additional experiment to demonstrate FedOne's performance under varying levels of client heterogeneity. Following prior works from Lin et al. (2020) and Yurochin et al. (2019), we model heterogeneity using a Dirichlet distribution with concentration parameters $\alpha = 0.5$ for medium heterogeneity and $\alpha = 0.1$ for high heterogeneity. Each experiment is conducted independently three times. The full results, including three figures corresponding to varying levels of heterogeneity, are presented in Figures 4, 5, and 6. As observed from the results, activating a single client consistently yields the optimal query efficiency, which is aligned with the theoretical result.

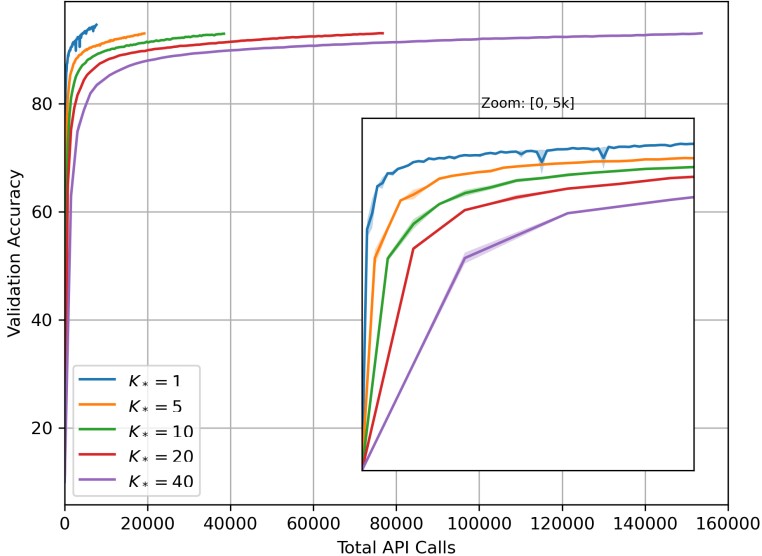

Figure 4: IID client

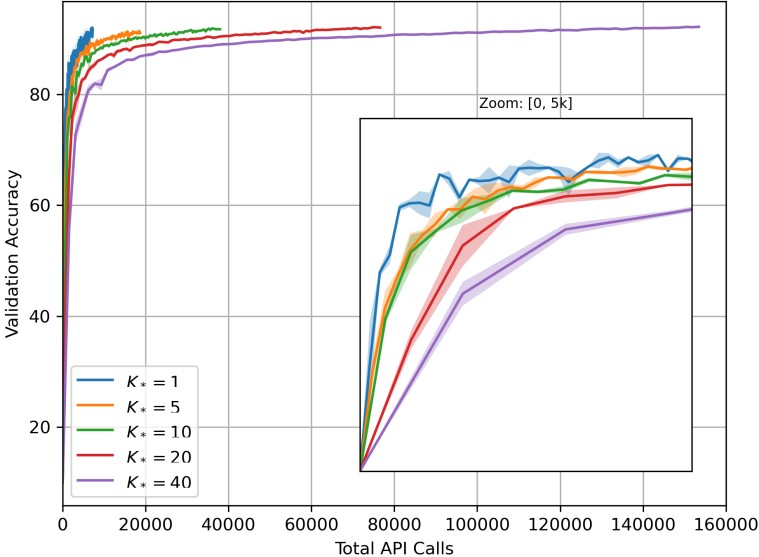

Figure 5: Medium Heterogeneity Clients

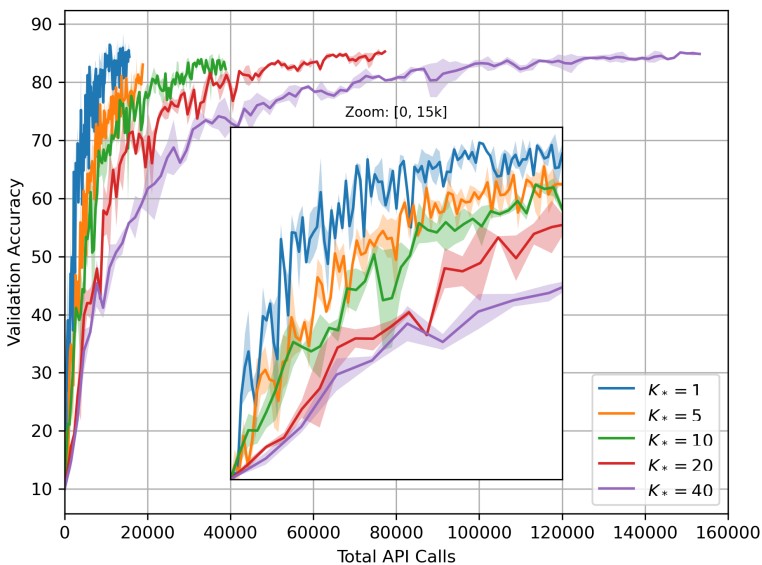

Figure 6: High Heterogeneity Clients

## C.4. Test Metrics for the RoBERTa-Large Experiment for FedAvg

To complement our main results in table 1, we present the detailed test metrics for the RoBERTa-Large experiment using the FedAvg algorithm with 10 activated clients per round. Table 7 serves as a supplementary to Table 1, providing the FedAvg result of each prompt tuning methods.

Table 7: The test metrics on the RoBERTa-large of FedAvg ($K_* = 10$). Each trial runs across three random seeds.

| Dataset | MNLI | QQP | SST-2 | MRPC | CoLA | QNLI | RTE | Avg. |
|---|---|---|---|---|---|---|---|---|
| Fed-PromptTuning | $41.8_{0.0}$ | $66.3_{0.1}$ | $78.3_{0.2}$ | $80.7_{0.2}$ | $0.6_{0.0}$ | $51.1_{0.0}$ | $51.4_{0.3}$ | 52.89 |
| Fed-P-Tuning v2 | $42.4_{0.1}$ | $66.6_{0.1}$ | $82.4_{0.5}$ | $80.6_{0.4}$ | $0.9_{0.1}$ | $52.9_{0.1}$ | $53.7_{0.4}$ | 54.21 |
| Fed-BBT | $41.4_{1.2}$ | $66.7_{0.2}$ | $77.6_{0.4}$ | $80.1_{1.2}$ | $2.7_{0.6}$ | $51.9_{0.6}$ | $52.3_{0.7}$ | 53.24 |
| Fed-BDPL | $40.8_{1.0}$ | $67.0_{0.2}$ | $82.9_{2.8}$ | $80.5_{1.6}$ | $5.9_{0.2}$ | $52.0_{0.6}$ | $53.8_{0.5}$ | 54.70 |
| Fed-GS-BDPL | $41.2_{0.4}$ | $67.1_{0.1}$ | $82.8_{1.4}$ | $80.5_{1.0}$ | $5.8_{0.5}$ | $52.3_{0.2}$ | $54.5_{0.4}$ | 54.89 |

## C.5. Test Accuracy for Various Total Number of Clients

To demonstrate the scalability of FedOne, we further report the test accuracy of FedOne-BDPL and FedOne-GS-BDPL on the SST-2 dataset under varying total numbers of clients (100, 1,000, and 10,000). Both methods exhibit stable performance as the number of clients increases, demonstrating strong scalability.

Table 8: Performance of FedOne on SST-2 with varying total number of clients

| Framework | 100 clients | 1000 clients | 10000 clients |
|---|---|---|---|
| FedOne-BDPL | $80.8_{6.0}$ | $79.9_{1.7}$ | $79.7_{2.4}$ |
| FedOne-GS-BDPL | $80.8_{0.4}$ | $79.7_{1.6}$ | $79.8_{1.0}$ |

