# OpenReview forum: "FedOne: Query-Efficient Federated Learning for Black-box Discrete Prompt Learning"
_ICML.cc/2025/Conference — ICML 2025 poster_

### Official Review · Reviewer_kF7d · 2025-03-02

**Overall Recommendation:** 3

**Summary:**

The authors propose FedOne to increase the query efficiency prompt learning method for cloud-based LLM, which activates only one client per round for optimal efficiency. The proposed method is shown to be effective by extensive experiments.

**Claims And Evidence:**

Good.

**Essential References Not Discussed:**

This paper is cited by not referenced and compared in the experiments:
[1] Sun, Jingwei, et al. "Fedbpt: Efficient federated black-box prompt tuning for large language models." arXiv preprint arXiv:2310.01467 (2023).

**Experimental Designs Or Analyses:**

Good.

**Methods And Evaluation Criteria:**

Fair.

**Other Comments Or Suggestions:**

Please refer to Strengths And Weaknesses.

**Other Strengths And Weaknesses:**

Strengths:
1. The authors provide theoretical analysis.
2. The authors conduct experiments on multiple datasets.

Weaknesses:
1. The motivation behind query efficiency is unclear. What is the main purpose of optimizing it? I would assume that a server-deployed LLM would have sufficient system capacity and tailored algorithms to handle concurrent queries efficiently, such as through parallelization techniques.
2. The authors make assumption on client heterogeneity but ignore the experiments for heterogeneous FL settings.
3. In Table 1, what are the preformance of Fed-X, i.e., sampling multiple active clients?

**Questions For Authors:**

Please refer to Strengths And Weaknesses.

**Relation To Broader Scientific Literature:**

Good.

**Theoretical Claims:**

Good.

---

> ### Author Rebuttal · Authors · 2025-04-01
>
> We sincerely thank you for your insightful comments and constructive criticisms. Your feedback has been invaluable in improving the quality and clarity of our manuscript. Below, we address the **Weaknesses and Essential Reference part**.
>
> >**W1**: Motivation behind optimizing query efficiency
>
> The motivation for improving query efficiency is to **reduce the monetary cost and practical constraints associated with training federated black-box prompt learning models using cloud-based LLMs**. Each query to commercial APIs incurs usage fees and is subject to rate limits, both of which scale with the number of active clients and training iterations in Fed-BDPL. (e.g. GPT-4o is \\$2.5/1M input tokens, \\$10/1M output tokens, with rate limits depending on the usage tier.)
>
> Our goal is to make federated black-box prompt tuning cost-effective and scalable in real-world deployments. To that end, our analysis and design focus on the FL framework’s query behavior to cloud-based LLM, rather than how the LLM server handles concurrent API calls.
>
> We will revise the introduction to emphasize the motivation behind optimizing query efficiency.
>
> >**W2**: Experiment on heterogeneity
>
> We provide an additional experiment to **demonstrate FedOne’s performance under varying levels of client heterogeneity**. Following prior works [1, 2], we simulate heterogeneity using a Dirichlet distribution with concentration parameters $\alpha=0.5$ for medium heterogeneity and $\alpha = 0.1$ for high heterogeneity. Each experiment is run three times independently. The complete results, including three figures corresponding to different heterogeneity levels, are available at this anonymous link ( https://anonymous.4open.science/r/ICML-Rebuttal-FedOne-649F/Converge2-Medium_Hetero.png ).
>
> These experiments further support our theoretical results. **Activating a single client per round ($K_*=1$) consistently achieves the highest query efficiency, even in highly heterogeneous settings**. This is because the core intuition behind FedOne remains valid under the heterogeneous case: the query cost to the LLM increases linearly with the number of activated clients (since each client issues one query), while increasing K is not able to provide a linear speedup in convergence rate. Our original submission did not include these heterogeneity results because our primary goal was to theoretically identify and validate the optimal query-efficient strategy for Federated Black-box Prompt Learning. Heterogeneity is outside the core scope of this work.
>
> We appreciate the reviewer’s suggestion, and we will add the heterogeneity experiments to the Appendix C.
>
> >**W3**: Fed-X result for Table 1
>
> The corresponding result is provided at this anonymous link ( https://anonymous.4open.science/r/ICML-Rebuttal-FedOne-649F/kF7d_Fed-X_in_table1.jpg ), and it demonstrates that FedOne-X and Fed-X achieve comparable test metrics. We did not include the test metrics for Fed-X in the main text because the **primary goal of our research is to improve the query efficiency of Fed-BDPL; optimizing test accuracy is beyond the scope of this work**. Our theoretical results also specifically focus on convergence behavior and query efficiency, and do not deal with generalization errors.
>
> We will include the results of Fed-X of Table 1 in Appendix C for completeness.
>
> >**Essential References**: Experiment comparison with FedBPT
>
> Thank you for raising this important point. We would like to clarify that **FedBPT is already included in our experiments under the name Fed-BBT**. *(For context, FedBPT (Sun et al., 2023) applies BBT (Sun et al., 2022) to federated learning, with specific adaptations to enhance performance in FL.)* Specifically, our Fed-BBT baseline implements the FedBPT, where the CMA-ES parameters are aggregated across clients, instead of full model parameters.
>
> We chose the name "Fed-BBT" (instead of "FedBPT") to help the reader intuitively compare methods within our framework. The naming convention **"Fed-BBT vs. FedOne-BBT"** makes it immediately clear that the two methods share the same prompt tuning backbone (BBT) but differ in client activation mechanisms (Fed- vs. FedOne-). In contrast, naming "FedBPT vs. FedOne-BBT" requires additional prior knowledge to recognize that they shared the same backbone.
>
> We will revise Section 4.2 to explicitly state that Fed-BBT is adapted from FedBPT, with BBT as the underlying prompt tuning method.
>
> ---
> **References**
>
> *[1] Lin, Tao, et al. "Ensemble distillation for robust model fusion in federated learning." NeurIPS 33 (2020): 2351-2363.*
>
> *[2] Yurochkin, Mikhail, et al. "Bayesian nonparametric federated learning of neural networks." ICML. PMLR, 2019.*

---

### Official Review · Reviewer_vcwQ · 2025-03-06

**Overall Recommendation:** 3

**Summary:**

This paper explores Federated Black-Box Discrete Prompt Learning and introduces FedOne, a novel approach that selects a single client per round. The chosen client updates the sampling probability for each token at different positions, optimizing prompt learning in a federated setting. Comprehensive experiments confirm the effectiveness.

**Claims And Evidence:**

Yes.

**Essential References Not Discussed:**

Yes.

**Experimental Designs Or Analyses:**

Yes.

**Methods And Evaluation Criteria:**

Yes.

**Other Comments Or Suggestions:**

Refer to weakness.

**Other Strengths And Weaknesses:**

The author provides detailed theory analysis and experimental comparison.

Weakness.

1.	Background Introduction
The rationale for employing Black-Box Discrete Prompt Learning in a federated setting is unclear. Given the popularity of Federated Prompt Learning, as demonstrated in prior works [1,2,3,4], why not leverage standard federated prompt learning approaches? The authors should refine the paper’s logical structure to better justify the proposed approach.
References:
	•	[1] Tao Guo et al., PromptFL: Let Federated Participants Cooperatively Learn Prompts Instead of Models—Federated Learning in the Age of Foundation Models, IEEE TMC, 2023.
	•	[2] Guoyizhe Wei et al., Dual Prompt Tuning for Domain-Aware Federated Learning, arXiv preprint arXiv:2310.03103, 2023.
	•	[3] Hongxia Li et al., Global and Local Prompts Cooperation via Optimal Transport for Federated Learning, CVPR, 2024.
	•	[4] Hangchao Su et al., Federated Adaptive Prompt Tuning for Multi-Domain Collaborative Learning, AAAI, 2024.

2.	Client Selection Strategy
The motivation behind selecting only one client per round is unclear. Why not select multiple clients (e.g., two or three) to balance convergence speed and robustness? The rationale for this design choice should be explicitly discussed, and a convergence comparison with alternative selection strategies would strengthen the justification.

3.	Prompt Length Impact
The effect of prompt length is not sufficiently discussed. In Section C1, the authors examine different prompt lengths but do not explain why increasing the prompt length leads to worse performance. A detailed analysis of this phenomenon is needed.

4.    Lack comparison. As far as I know, the FedBPT is the closest with your work. Could you explain the rationale difference and compare with them?

[1] FedBPT: Efficient Federated Black-box Prompt Tuning for Large Language Models

**Questions For Authors:**

Refer to weakness.

**Relation To Broader Scientific Literature:**

Adapting federated learning to Black-Box Discrete Prompt Learning could further enhance prompt tuning performance by leveraging data from diverse sources, which provides the chance for multi-clients to collaboratively learn the large model in a parameter-efficient way.

**Theoretical Claims:**

The notation in the paper is relatively complex, even for standard federated learning pipelines, as seen in Eq. (1) and (2). The authors could simplify the notation to enhance clarity and readability.

---

> ### Author Rebuttal · Authors · 2025-04-01
>
> We sincerely thank you for your insightful comments and constructive criticisms. Below, we address the weaknesses.
> >**W1**: Rationale of Fed-BDPL
>
> The rationale for employing **Black-box** Discrete Prompt Learning is grounded in two key real-world constraints:
>
> 1. **Lack of Access to Model Internals**: The white-box federated prompt learning approaches (reference [1–4] you mentioned) assume direct access to model weights or gradients, which is infeasible in realistic scenarios when we want to leverage the most advance commercial, closed-source LLMs (e.g., GPT, Claude), that are only accessible via APIs. Our work specifically targets this black-box setting, where the prompt tuning is conducted without model access—a practical and increasingly common constraint in modern applications.
> 2. **Resource Constraints on FL Clients**: In typical FL scenarios, clients are resource-constrained devices (e.g., mobile phones) with limited compute and memory. However, white-box methods ([1-4]) require storing and training on the LLM model locally, which is computationally infeasible for such device. In contrast, black-box prompt tuning offloads the computation to the LLM API, making it scalable and deployable in realistic FL scenario.
>
> We will revise the introduction (par. 3, 4) to make the above rationale more explicit, and ensure that references [1–4] you mentioned are properly cited in the Related Work section.
>
> >**W2**: Motivation of FedOne
>
> The motivation to activate only one client per round is not heuristic, but rather **grounded in our theoretical analysis of query efficiency in Fed-BDPL**, which includes the convergence of Fed-BDPL (Theorems 3.4, 3.5) and query efficiency (Corollary 3.6). Specifically, **Corollary 3.6 demonstrates that setting K=1 achieves optimal query efficiency for reaching an $\epsilon$-solution in the Fed-BDPL framework**. The FedOne framework is directly derived from this theoretical result to achieve optimal query efficiency.
>
> The **intuition** behind FedOne is further elaborated in **Remark 3.7**: the query cost to the LLM increases linearly with the number of activated clients (since each activated client issues one query), while increasing $K_*$ is not able to provide a linear speedup in convergence rate. Consequently, **the marginal gain in convergence rate does not compensate for the linear increase in query cost, making $K_*=1$ the most query-efficient choice under this framework.**
>
> We further empirically validate this in **Figure 2**, where different client selection strategies ($K_*=1,5,10,20,40$) are compared. The results show that activating a single client per round consistently achieves the optimal query efficiency, aligning with our theory.
> >**W3**: The Impact of Prompt Length
>
> While prompt length is not the primary focus, we agree that it is an important factor. As shown in Fig. 3, performance remains relatively stable across a moderate range of prompt lengths, with the average accuracy staying in $0.83\pm0.005$. This suggests **no statistically significant performance drop** when increasing the prompt length.
>
> However, it is worth noting that very short or very long prompts introduce larger variance due to the properties of BDPL.
> 1. Very short prompts lack sufficient capacity to instruct the LLM effectively.
> 2. Very long prompts lead to a larger search space and increased training difficulty.
>
> Based on these observations, we selected a prompt length of 20 in our main experiments, as it offers a good trade-off, with the lowest variance. We will clarify this rationale in Appendix C.
> >**W4**: Compare with FedBPT
>
> Thank you for raising this important point. We would like to clarify that **FedBPT is already included in our experiments under the name "Fed-BBT"**. *(For context, FedBPT (Sun et al. 2023) applies BBT (Sun et al. 2022) to FL, with adaptation designs to enhance its performance in FL.)* Specifically, our Fed-BBT baseline implements the FedBPT, where the CMA-ES parameters are aggregated across clients, instead of full model parameters.
>
> We chose the name "Fed-BBT" (instead of "FedBPT") to help the reader compare methods intuitively. E.g., using **"Fed-BBT vs. FedOne-BBT"** immediately tells that they share the same backbone but differ in client activation mechanisms (Fed- vs. FedOne-). In contrast, naming "FedBPT vs. FedOne-BBT" requires additional prior knowledge to recognize that they are comparable.
>
> We will revise Section 4.2 to explicitly state that Fed-BBT is adapted from FedBPT, with BBT as the underlying prompt tuning method.
> >**W4**: The rationale difference between FedBPT and ours
>
> The main rationale difference between FedBPT and our work lies in the **research focus**: FedBPT focuses on adapting BBT to the federated learning setting, with specific designs to improve optimization performance across clients. In contrast, our work focuses on query efficiency, a crucial and underexplored challenge in federated black-box prompt learning with cloud-based LLMs.

---

> > ### Comment · Reviewer_vcwQ · 2025-04-03
> >
> > After reading the rebuttal, I decided to decrease my score.
> >
> > **First**, the motivation based on theoretical analysis remains somewhat unclear. Theorems 3.4 and 3.5 on page 4 only discuss the convergence conditions, but what I care more about is whether the setting itself is reasonable in a federated learning scenario. Although Figure 2 shows query efficiency and validation accuracy, I would suggest also plotting validation accuracy versus training epochs to provide a better understanding of training dynamics. Additionally, you mention using a toy model, but it is unclear why the analysis is conducted on **MNIST**, while your main experiments are performed on datasets such as MNLI, QQP, SST-2, MRPC, CoLA, QNLI, and RTE. Why not perform the theoretical analysis and ablation studies on these actual benchmark datasets? Furthermore, it would strengthen your work to demonstrate that your method scales well across varying numbers of clients. Lastly, explaining how your method handles conflicting objectives or consensus among participating clients—a central issue in traditional federated learning—would be important.
> >
> > **Second**, as you mention “Resource Constraints on FL Clients,” it is questionable that your experiments are based on RoBERTa-large, which is computationally heavy. Moreover, while you state “Lack of Access to Model Internals,” your paper (page 5) also mentions that “the trainable prompts were placed at different positions in the model depending on the algorithm of the baselines.” This appears contradictory and should be clarified by referring to your own descriptions.
> >
> > **Third**, the discussion on the difference from FedBPT lacks depth. It is not enough to just list the differences; the paper should clearly articulate why FedBPT fails to address the core challenges targeted by your approach.

---

> > > ### Author Response · Authors · 2025-04-05
> > >
> > > We sincerely appreciate your time and effort in reviewing our work. Thank you for your constructive feedback.
> > >
> > > ## **1st:**
> > >
> > > >*Convergence and "whether the setting is reasonable in FL?"*
> > >
> > > The federated black-box prompt learning setting is essential when working with **closed-source commercial LLMs, where the white-box prompt learning is infeasible**. It also aligns with real-world FL scenarios involving resource-constrained clients, as **computation is offloaded to the cloud-based LLM server**, making it a more practical and scalable approach for real-world deployments.
> > >
> > > Regarding the theoretical analysis on convergence and query efficiency: The motivation to analyze query efficiency under the federated black-box setting arises from the fact that **each query to the cloud-based LLM incurs monetary cost and latency, which has become a dominant factor in the overall system cost**. Our analysis is designed to directly address this practical challenge.
> > >
> > > >*Fig. 2, validation accuracy vs. training epochs*
> > >
> > > Thank you for your suggestion. We add plots of validation accuracy vs. epochs and vs. total API calls, extending the experiment shown in Fig. 2. The plots are presented side by side to better illustrate the training dynamics. The results are available at ( https://anonymous.4open.science/r/ICML-Rebuttal-FedOne-649F/Fig2_Valid-Epoch-API.png ).
> > >
> > > >*MNIST and GLUE benchmark. Why not perform the theoretical analysis and ablation studies on these actual benchmark datasets?*
> > >
> > > We would like to point out that **the ablation study on varying the number of activated clients using the SST-2 dataset has already been provided in Appendix C.1 (as noted in footnote 3)**. The results consistently show that activating one client per round yields the best query efficiency, aligning with our theoretical analysis.
> > >
> > > We chose MNIST as the toy experiment due to its familiarity within the ML community, allowing us to illustrate the core intuition of FedOne in a simple and accessible manner to researchers across various subfields of ML and FL.
> > >
> > > > *How does FedOne scale with varying numbers of clients?*
> > >
> > > Thank you for your suggestion, we add one more experiment on varying the total number of clients. The results are available at ( https://anonymous.4open.science/r/ICML-Rebuttal-FedOne-649F/Varying_total_number_of_client.png ).
> > >
> > > >*Conflicting objectives or consensus among clients*
> > >
> > > We would like to clarify that the problem of "conflicting objectives and client consensus" is **beyond the scope of our study**.
> > >
> > > This problem typically arises in learning frameworks involving **multiple distinct objectives**, such as in personalized FL (client-specific objectives), multi-task learning (task-specific objectives), or in pretraining–finetuning conflicts (transferred objective). These settings primarily focus on enhancing generalization across different tasks or clients.
> > >
> > > In contrast, our work focuses on the convergence and query efficiency of the system. Accordingly, our formulation assumes a **shared global optimization objective** for the entire FL system (Eq. 1). Therefore, the study of conflicting objectives among clients is beyond the scope of our study.
> > >
> > > ## **2nd:**
> > >
> > > >*Experiments on RoBERTa-large are computationally heavy.*
> > >
> > > In that experiment, when using the **black-box baselines**, the "heavy computation" associated with RoBERTa-large is **offloaded to the cloud-based LLM server**. RoBERTa-large is treated as a frozen model hosted by the cloud-based LLM server, serving purely as a black-box oracle without exposing any model internals to the clients. This setup significantly reduces the computational burden on FL clients and aligns with the “Resource Constraints on FL Clients.”
> > >
> > > >*"The trainable prompts were placed at different positions in the model"*
> > >
> > > We apologize for the confusion, and we will modify this part to prevent ambiguity. This sentence mainly refers to the white-box baselines.
> > >
> > > In the white-box baselines, the LLM is stored and executed locally on each client. In this setting, trainable prompts are placed at different positions within the model, depending on the algorithm used (Prompt Tuning or Prefix-Tuning v2).
> > >
> > > In the black-box baselines, the LLM is hosted on the cloud-based server, and clients have no access to model internals. Instead, they optimize local parameters to generate prompts, which are evaluated by querying the cloud-based LLM, which fully adheres to the black-box learning paradigm.
> > >
> > > ## **3rd:**
> > >
> > > >*FedBPT*
> > >
> > > We will further clarify why FedBPT fails to address the core challenges of our work.
> > >
> > > FedBPT does not explicitly consider or optimize the query cost associated with cloud-based LLMs ("Inference API" in their paper). **Neither their theoretical analysis nor their experiments quantify or minimize the number of queries to the Inference API**. In contrast, our work explicitly targets query efficiency, providing both theoretical insights and a practical framework designed to minimize the number of LLM queries.

---

### Official Review · Reviewer_jBwD · 2025-03-13

**Overall Recommendation:** 4

**Summary:**

The paper introduces a federated learning (FL) framework designed to improve the query efficiency of Black-Box Discrete Prompt Learning (BDPL) when interacting with cloud-based Large Language Models (LLMs). Traditional federated black-box prompt tuning approaches incur high query costs due to multiple clients querying the cloud-based LLM in each training round. To address this, the authors propose FedOne, a specific case of FedAvg framework that activates only one client per round.

**Claims And Evidence:**

The convergence of Fed-BDPL (theorem 3.4, Coro 3.5). Activating one client in Fed-BDPL framework can achieve $\epsilon$-solution with least queries (corollary 3.6).

**Essential References Not Discussed:**

No

**Experimental Designs Or Analyses:**

The experiment looks pretty standard (benchmark, baselines). It uses the GLUE dataset and demonstrates results on the white-box RoBERTa and the black-box GPT-3.5, covering standard tuning methods such as prompt-tuning, prefix-tuning, and black-box tuning. Additionally, it includes an analysis of computational and communication costs. The toy experiment also aligns well with the theoretical findings.

**Methods And Evaluation Criteria:**

The proposed method is well-motivated and aligns with intuitive reasoning. The evaluation follows standard practices. However, further experiments on non-IID settings would strengthen the generalizability of the results.

**Other Comments Or Suggestions:**

1. #016, Large Language Model (LLM).
2. For table 1 you can also bold the highest, which makes a consistent format as table 3.
3. #220, the footnote 2 is not complete.
4. #325, in table 1. We observed...

**Other Strengths And Weaknesses:**

Strengths:
1. Though the idea of FedOne is quite simple, the author provides rigorous proof by deriving the explicit form of the number of queries required to achieve the $\epsilon$-solution (Corollary 3.6) effectively justifying the underlying principles of the approach. This finding is interesting and novel.
2. The paper identifies a previously overlooked research problem, the substantial query cost on cloud-based LLM using convention FedAvg.
3. The paper presents the first theoretical analysis of Fed-BDPL, providing a rigorous foundation for understanding its convergence and query efficiency. This analysis provides valuable insights in the optimization of Fed-BDPL.
4. The presentation is clear. The core idea is simple, with no unnecessary components added to the framework to artificially increase system complexity. Theoretical analysis and algorithmic design are seamlessly integrated, contributing to the paper's overall coherence.


Weaknesses:
1. While heterogeneity is not the research focus of this paper, a discussion on heterogeneity and its potential impact on the FedOne framework could be further explored in the theoretical analysis. This would provide deeper insights into how FedOne performs under varying degrees of client heterogeneity. I am very curious about this question.
2. The experiment is also solely based on the IID case. The theoretical analysis suggests that under bounded client heterogeneity, one client is most query-efficient. I suggest that the authors should also include additional experiments on heterogeneous data to evaluate whether the FedOne framework remains efficient under some level of heterogeneity, which make the experiment section more align with the theorem.

**Questions For Authors:**

How can heterogeneity affect the result of Corollary 3.6, FedOne? How does heterogeneity affect the results in the experiments?

**Relation To Broader Scientific Literature:**

The backbone of this paper is Fed-BDPL (Lin et al., 2023), which applies BDPL (Diao et al., 2022) to FedAvg (McMahan et al., 2017).
1. This paper identifies the previously overlooked query cost problem in cloud-based LLMs within Fed-BDPL.
2. This paper presents the first convergence analysis of Fed-BDPL (Theorem 3.4) and further explores query efficiency and convergence behavior (Corollary 3.6), providing a rigorous theoretical foundation for fed-BDPL.
3. Building on these theoretical results, the author demonstrate that activating only one client per round achieves optimal query efficiency in Fed-BDPL, leading to the proposed FedOne framework.

**Theoretical Claims:**

I checked the proof of Theorem 3.4. It makes sense to me.

---

> ### Author Rebuttal · Authors · 2025-04-01
>
> Thank you for your insightful comments and for recognizing the contribution of our work. Your feedback has been invaluable in enhancing the quality and clarity of our manuscript. We reply to the weaknesses and questions.
>
> >**W1&Q1**: Impact of heterogeneity on theoretical analysis
>
> Thanks for pointing that out. We will add more discussion on the heterogeneity in the theoretical analysis.
>
> Regarding the theoretical result under heterogeneity, the conclusion that activating one client per round yields the highest query efficiency in Fed-BDPL still holds. However, the trade-off is a slower convergence rate per round, as fewer clients contribute updates in each iteration.
>
> 1. Theorem 3.4, Theorem 3.5, and Corollary 3.6 explicitly consider bounded client heterogeneity, and the result that **$K_*=1$ achieves optimal query cost remains valid in this context**. This is because the core intuition behind FedOne continues to hold: the query cost to the LLM increases linearly with the number of activated clients (as each client issues one query), while increasing $K_*$ is not able to provide a linear speedup in convergence ([1,2]).
> 2. The trade-off is that fewer activated clients per round lead to a slower convergence rate (e.g., $O(\frac{1}{\sqrt{K T}})$ convergence in [1]), even though it improves overall query efficiency.
>
> >**W2&Q2**: Experiment on heterogeneity
>
> We have also included an additional experiment to demonstrate **FedOne’s performance under varying levels of client heterogeneity**, supporting the points discussed in W1&Q1. Following prior works [3, 4], we simulate heterogeneity using a Dirichlet distribution with concentration parameters $\alpha=0.5$ for medium heterogeneity and $\alpha = 0.1$ for high heterogeneity. Each experiment is run three times independently. The complete results, including three figures corresponding to different heterogeneity levels, are available at the following anonymous link ( https://anonymous.4open.science/r/ICML-Rebuttal-FedOne-649F/Converge2-Medium_Hetero.png ). We will include the above experiments in Appendix C.
>
> The experiment shows that across all levels of heterogeneity, when the trials converge to a similar stationary point (i.e., approaching the same validation accuracy), **activating a single client per round ($K_*=1$) consistently achieves the lowest query cost**.
>
> The trade-off is that:
> 1. Activating fewer clients increases variance in training, which can lead to greater fluctuations.
> 2. Activating fewer clients also lowers the convergence rate, and conversely, increasing K can speed up convergence at the cost of higher query overhead.
>
> Based on the above trade-off established in our work, practitioners can tune $K_*$ according to their specific system constraints and performance goals.
>
> Despite these trade-offs, the **primary objective of our work is to identify the optimal query-efficient strategy within the Fed-BDPL framework that still guarantees convergence**. Both our theoretical analysis and empirical results demonstrate that activating a single client per round achieves this goal, providing the best balance in settings constrained by the high cost and rate limits associated with querying cloud-based LLMs.
>
> >**Other comments**
>
> Thank you very much for pointing out these helpful corrections! We have revised them accordingly.
>
> ---
> References
>
> *[1] Haddadpour, Farzin, and Mehrdad Mahdavi. "On the convergence of local descent methods in federated learning." arXiv preprint arXiv:1910.14425 (2019)*
>
> *[2] Li, Xiang, et al. "On the convergence of fedavg on non-iid data." ICLR 2020.*
>
> *[3] Lin, Tao, et al. "Ensemble distillation for robust model fusion in federated learning." Advances in neural information processing systems 33 (2020): 2351-2363.*
>
> *[4] Yurochkin, Mikhail, et al. "Bayesian nonparametric federated learning of neural networks." International conference on machine learning. PMLR, 2019.*

---

### Official Review · Reviewer_BZU1 · 2025-03-17

**Overall Recommendation:** 4

**Summary:**

This paper introduces a federated learning framework for black-box discrete prompt learning (BDPL), specifically suitable for cloud-based LLMs. The core idea of FedOne is to optimize query efficiency by degrading the traditional FedAvg algorithm to activate only a single client per round. The authors claim to provide the first theoretical analysis of query efficiency in federated BDPL, demonstrating that FedOne achieves optimal query efficiency in this context. Empirical results from numerical experiments are presented to support these theoretical findings, showing significant improvements in query efficiency compared to existing federated black-box prompt tuning approaches.

## update after rebuttal

I feel my concerns are addressed and I have increased my score.

**Claims And Evidence:**

The paper's central claim about the optimality of $K^*=1$ for query efficiency.

E1: The theoretical analysis in Sec. 3 provides a foundation for the claim, the authors derive the convergence rate of Fed-BDPL (Corollary 3.5) and the query complexity function (Corollary 3.6), showing that it achieves minimum at $K^* < 1$.
E2: The empirical validation includes
- A toy experiment demonstrating that $K^*=1$ achieves better query efficiency on MNIST data
- Evaluations on GLUE benchmark tasks.

The claim about comparable performance despite reduced query cost is well-supported by the experimental results in Table 1 and Table 3, where FedOne-BDPL and FedOne-GS-BDPL achieve performance metrics similar to other methods while reducing queries by orders of magnitude.

**Essential References Not Discussed:**

I am not an expert in FL, and I am not able to identify clear missing references.

**Experimental Designs Or Analyses:**

The toy experiment (figure 2) appropriately demonstrates the relationship between query efficiency and $K^*$ in a controlled setting.

For the GLUE benchmark evaluation, the use of 100 clients with $k$-shot datasets simulates a FL scenario, and the hyper parameter tuning with grid search is thorough and follows standard practice.

I appreciate that the authors also provided computational efficiency experiment (table 2), including direct comparison of white-box vs. black-box approaches.

**Methods And Evaluation Criteria:**

The proposed method, FedOne, which is essentially FedAvg with client selection restricted to one client per round, is a sensible approach to potentially improve query efficiency. By activating only one client, the number of queries to the cloud-based LLM per round is directly reduced.

For evaluation, this paper focuses on query efficiency as a primary criterion. The authors also consider query cost for cloud-based LLM APIs.

For benchmark datasets, GLUE is a standard choice for evaluating LLM performance across diverse tasks.

**Other Comments Or Suggestions:**

N/A.

**Other Strengths And Weaknesses:**

I summarize the strengths and weaknesses I identified below. They may overlap with the previously discussed points:

### strengths

1. The paper introduces a counter-intuitive but well-substantiated insight that fewer activated clients leads to better query efficiency in federated black-box prompt learning.
2. The theoretical analysis establishes connections between convergence rates and query complexity
3. The experimental validation covers both synthetic data and real-world LLM APIs.
4. The topic being studied is timely and important.

### Weaknesses:
1. Limited analysis of the impact of heterogeneity across clients.
2. The analysis assumes uniform query costs across all clients, but in practice, query complexity might vary based on input length, client location, and other factors.
3. It appears that the paper does not explore adaptive strategies for determining $K^*$ based on system conditions.

**Questions For Authors:**

1. How does the FedOne approach perform when client data distributions are highly heterogeneous? Since only one client is activated per round, does this potentially slow convergence in highly non-IID settings compared to traditional FL approaches?
2. Have you explored adaptive strategies for setting K* based on observed convergence patterns or system conditions?
3. The paper assumes uniform query costs across clients. How would varying query costs (due to different prompt lengths, computation time, etc.) affect the theoretical analysis and the optimality of $K^*=1$?

**Relation To Broader Scientific Literature:**

This paper should be positioned within three interconnected areas: prompt tuning for LLMs, federated learning, and federated prompt tuning.

**Theoretical Claims:**

I did not verify the details in the proof. I do not find apparent flaws with high-level theoretical analysis.

---

> ### Author Rebuttal · Authors · 2025-04-01
>
> Thank you for your insightful comments and for recognizing the contribution of our work. Your feedback has been invaluable in enhancing the quality and clarity of our manuscript. We reply to the questions and weaknesses.
>
> >**Q1&W1**: Impact of heterogeneity
>
> When the client‘s data distribution is highly heterogeneous, **the theoretical result that activating one client per round yields the highest query efficiency in Fed-BDPL still holds**. This is because the core intuition behind FedOne remains valid under such conditions: the query cost to the LLM increases linearly with the number of activated clients (since each activated client issues one query), while increasing K is not able to provide a linear speedup in convergence ([1, 2]).
>
> We have also included an additional experiment to evaluate FedOne's performance under highly heterogeneous client distributions. Following prior works [3, 4], we simulated heterogeneity using a Dirichlet distribution with the concentration parameter $\alpha=0.1$, representing a highly non-IID setting. The result demonstrates that, **even under highly non-IID conditions, activating a single client per round ($K_*=1$) still achieves the highest query efficiency**, which is aligned with the theoretical result. The result figure can be found at this anonymous link ( https://anonymous.4open.science/r/ICML-Rebuttal-FedOne-649F/Converge3-High_Hetero.png )
>
> We will include this experiment in Appendix C.
>
> >**Q1**: Potentially slow convergence
>
> As demonstrated in [2], activating more clients per round under non-IID settings can improve the convergence rate, but it does not yield a linear speedup. By setting $K_*=1$ in FedOne, **our approach intentionally trades off this marginal gain in convergence rate in favor of query efficiency**.
>
> However, the primary goal of our work is not to maximize convergence rate per round, but to **identify an optimal query-efficient strategy within the Fed-BDPL framework that still guarantees convergence**. Both our theoretical analysis and empirical results confirm that activating a single client per round achieves this objective, offering the best trade-off when considering the high cost and rate limits associated with querying cloud-based LLMs.
>
> We will add a discussion of this trade-off in the Appendix to provide a more balanced view of the design choice.
>
> >**Q2&W3**: Adaptive strategy of $K_*$
>
> Regarding the adaptive strategy, this work provides **two key insights into the trade-off** between the number of activated clients $K_*$, convergence speed, and query efficiency:
> 1. Increasing $K_*$ leads to higher query costs (scaling linearly with $K_*$) while yielding only sublinear improvements in convergence rate.
> 2. Reducing the number of activated clients K improves query efficiency but may result in slower convergence. Notably, setting $K_*=1$ achieves optimal query efficiency.
>
> Our research provides the above principles for tuning $K_*$, enabling practitioners to adjust $K_*$ according to their specific system constraints and performance objectives.
>
>
> >**Q3&W2**: Uniform query cost assumption
>
> We acknowledge that query costs may vary in practice due to factors like input length or client-specific characteristics. However, the uniform cost assumption allows us to simplify the theoretical analysis without significantly compromising realism.
>
> Since clients are randomly selected at each round, **variations in individual query costs are expected to average out over time**. Even if we modeled client query costs as a distribution, the inherent randomness in selection would mitigate the overall impact. Therefore, the uniform cost assumption offers a practical balance between analytical tractability and real-world applicability.
>
> We will add the above discussion to the manuscript.
>
> ---
> References
>
> *[1] Haddadpour, Farzin, and Mehrdad Mahdavi. "On the convergence of local descent methods in federated learning." arXiv:1910.14425 (2019)*
>
> *[2] Li, Xiang, et al. "On the convergence of fedavg on non-iid data." ICLR 2020.*
>
> *[3] Lin, Tao, et al. "Ensemble distillation for robust model fusion in federated learning." NeurIPS 33 (2020): 2351-2363.*
>
> *[4] Yurochkin, Mikhail, et al. "Bayesian nonparametric federated learning of neural networks." ICML. PMLR, 2019.*

---

> > ### Comment · Reviewer_BZU1 · 2025-04-02
> >
> > I thank the authors for providing additional theoretical analysis and addressing my questions. I have no more concerns and think this is a good paper to be accepted. I have increased my score as well.

---

> > > ### Author Response · Authors · 2025-04-03
> > >
> > > **We are truly grateful for your thoughtful and in-depth discussions!**
> > >
> > > **Your support means so much to us!**

---

### Decision · Program_Chairs · 2025-05-01

**Decision:**

Accept (poster)

**Comment:**

This work addresses the overlooked query cost problem to cloud-based LLM-server in federated black-box prompt tuning. Targeting this problem, the author presents a counterintuitive yet well-justified finding (FedOne): activating only one client per round in Fed-BDPL leads to optimal query efficiency when interacting with cloud-based LLM servers. The paper identified an important but overlooked research problem in Federated black-box prompt tuning. The proposed solution has a solid theoretical foundation in optimizing the query cost. The core intuition is thoroughly discussed via theoretical result, intuition and empirical experiment.

A major concern raised by the reviewers was the impact of client heterogeneity. Specifically, three reviewers (BZU1, jBwD, kF7d) questioned how heterogeneity might affect the theoretical result. In response, the authors provided additional experiments and a detailed discussion on the intuition and trade-offs involved. Additional concerns were raised about the setting (vcwQ) and motivation (kF7d) of the research. The authors articulated the relevance of federated black-box prompt tuning and the importance of minimizing query cost when interacting with LLM APIs.

All reviewers raised their scores after the rebuttal and are in favor of acceptance. After carefully reviewing the paper and the authors' responses, I find the work to be solid, and the overall discussion to be thorough and convincing.